# Genetic regulation of *OAS1* nonsense-mediated decay underlies association with COVID-19 hospitalization in patients of European and African ancestries

A. Rouf Banday[1,46], Megan L. Stanifer[2,3,46], Oscar Florez-Vargas [1,46], Olusegun O. Onabajo [1], Brenen W. Papenberg[1], Muhammad A. Zahoor [4], Lisa Mirabello[5], Timothy J. Ring [1], Chia-Han Lee[1], Paul S. Albert[6], Evangelos Andreakos [7], Evgeny Arons[8], Greg Barsh [9], Leslie G. Biesecker [10], David L. Boyle[11], Mark S. Brahier[12], Andrea Burnett-Hartman[13], Mary Carrington [14,15,16], Euijin Chang [17], Pyoeng Gyun Choe [17], Rex L. Chisholm[18], Leandro M. Colli [19], Clifton L. Dalgard [20], Carolynn M. Dude[21], Jeff Edberg[22], Nathan Erdmann[23], Heather S. Feigelson [13], Benedito A. Fonseca [24], Gary S. Firestein[11], Adam J. Gehring [4,25], Cuncai Guo[26], Michelle Ho [1], Steven Holland[27], Amy A. Hutchinson[28], Hogune Im [29], Les'Shon Irby[21], Michael G. Ison[30], Naima T. Joseph[31], Hong Bin Kim [17,32], Robert J. Kreitman[8], Bruce R. Korf[33], Steven M. Lipkin [34], Siham M. Mahgoub[35], Iman Mohammed[36], Guilherme L. Paschoalini [19], Jennifer A. Pacheco [18], Michael J. Peluso[37], Daniel J. Rader [38], David T. Redden[39], Marylyn D. Ritchie [38], Brooke Rosenblum[10], M. Elizabeth Ross [36], Hanaisa P. Sant Anna[40], Sharon A. Savage [5], Sudha Sharma[41], Eleni Siouti [7], Alicia K. Smith [21], Vasiliki Triantafyllia [7], Joselin M. Vargas[1], Jose D. Vargas[42], Anurag Verma[38], Vibha Vij[43], Duane R. Wesemann [44], Meredith Yeager [28], Xu Yu[16], Yu Zhang[27], Steeve Boulant[3,26,45], Stephen J. Chanock [40], Jordan J. Feld[4,25] and Ludmila Prokunina-Olsson [1✉]

The chr12q24.13 locus encoding OAS1–OAS3 antiviral proteins has been associated with coronavirus disease 2019 (COVID-19) susceptibility. Here, we report genetic, functional and clinical insights into this locus in relation to COVID-19 severity. In our analysis of patients of European ($n = 2,249$) and African ($n = 835$) ancestries with hospitalized versus nonhospitalized COVID-19, the risk of hospitalized disease was associated with a common *OAS1* haplotype, which was also associated with reduced severe acute respiratory syndrome coronavirus 2 (SARS-CoV-2) clearance in a clinical trial with pegIFN-λ1. Bioinformatic analyses and in vitro studies reveal the functional contribution of two associated *OAS1* exonic variants comprising the risk haplotype. Derived human-specific alleles rs10774671-A and rs1131454-A decrease OAS1 protein abundance through allele-specific regulation of splicing and nonsense-mediated decay (NMD). We conclude that decreased *OAS1* expression due to a common haplotype contributes to COVID-19 severity. Our results provide insight into molecular mechanisms through which early treatment with interferons could accelerate SARS-CoV-2 clearance and mitigate against severe COVID-19.

The response to pathogens, such as SARS-CoV-2, which causes COVID-19, is determined by the interplay between host and pathogen factors. Variability in clinical outcomes of COVID-19 has prompted the search for host genetic factors to elucidate underlying disease mechanisms and guide optimal treatment options. Recent genome-wide association studies (GWASs) have reported a series of genetic variants in distinct loci associated with susceptibility to COVID-19 overall or severe disease, comparing patients with general population controls[1,2].

The sentinel variant for one of the identified loci is rs10774671 at 12q24.13 (refs. [1,2]). The locus harbors three genes encoding

antiviral 2′,5′-oligoadenylate synthetase (OAS) enzymes (OAS1, OAS2 and OAS3), interferon-inducible antiviral proteins activating the latent form of ribonuclease L (RNase L)[3,4]. Activation of the RNase L pathway leads to degradation of viral RNA, inhibition of virus replication and cell death[3]. The RNase L pathway is specifically important for the immune response to SARS-CoV-2, an RNA virus.

Here, we investigated whether the locus influencing COVID-19 susceptibility at 12q24.13 is associated with COVID-19 severity by comparing hospitalized versus nonhospitalized patients drawn from European and African ancestries. Our in silico and experimental

analyses provide functional and clinical insights into the basis of the genetic association.

## Results

**The 12q24.13 locus is associated with COVID-19 hospitalization.** In a case–case analysis, we evaluated 3,084 patients from COVNET, comparing hospitalized versus nonhospitalized patients with COVID-19, including 1,214 versus 1,035 patients of European ancestry and 511 versus 324 patients of African ancestry, respectively (Extended Data Fig. 1). The genomic inflation factor was $\lambda = 1.01$ for European ancestry and $\lambda = 1.0$ for African ancestry (Extended Data Fig. 2). Within the *OAS1-OAS2-OAS3* region (hg38, 113 kb, chr12: 112,904,114–113,017,173), we focused on a set of shared variants genotyped or imputed with high confidence ($r^2 > 0.8$) in both ancestries. In Europeans, this set includes 79 variants with significant associations (odds ratio (OR) = 1.19 to OR = 1.35, $P = 3.81 \times 10^{-2}$ to $P = 3.66 \times 10^{-4}$) (Fig. 1 and Supplementary Table 1).

Only six of the above variants, all within *OAS1*, were also associated in patients of African ancestry: rs1131454 (Gly162Ser, OR = 1.37, $P = 0.021$), rs1131476 (Ala352Thr, OR = 1.58, $P = 0.035$), rs2660 (3'UTR, OR = 1.58, $P = 0.035$), rs4766664 (intron/intergenic, OR = 1.76, $P = 0.011$), and two variants, rs1051042 (Arg361Thr, OR = 1.58, $P = 0.035$) and rs4767027 (intron/intergenic, OR = 1.87, $P = 0.0058$), imputed in the African dataset with lower confidence scores ($r^2 = 0.7$ and $r^2 = 0.67$) (Fig. 1 and Supplementary Table 1). The COVID-19 susceptibility GWAS sentinel single-nucleotide polymorphism (SNP)[2], rs10774671, was significantly associated in patients of European ancestry (OR = 1.33, $P = 6.45 \times 10^{-4}$) but did not reach significance in patients of African ancestry (OR = 1.23, $P = 0.079$). In a meta-analysis of patients of both ancestries, all seven variants were associated with hospitalized COVID-19 (OR = 1.29 to OR = 1.37 and $P = 1.91 \times 10^{-4}$ to $P = 6.48 \times 10^{-5}$; Supplementary Table 1). Mutual conditioning on the variants attenuated or eliminated the signal, suggesting they are not independent (Extended Data Fig. 3 and Supplementary Table 2). We applied the linkage disequilibrium (LD)-adjusted threshold method[5] to adjust for multiple testing considering four LD blocks in European and 11 LD blocks in African COVNET datasets (Extended Data Fig. 4 and Supplementary Table 1).

We checked the seven *OAS1* variants in the COVID-19 Host Genetics Initiative (HGI) release 6 (Supplementary Table 3). Despite differences in *P* values, likely due to unequal sample numbers, the effect sizes for all these variants were comparable both in B2 analysis (hospitalized COVID-19 patients versus general population) and B1 analysis (hospitalized versus nonhospitalized COVID-19). Thus, COVNET and HGI results agree on comparable effect sizes between these variants, although all COVNET effect sizes are larger than in HGI. Ancestry-specific results from this region were reported only for rs10774671, which showed comparable effect sizes in individuals of European (OR = 0.92, $P = 5.8 \times 10^{-10}$, HGI-r6) and African ancestry (OR = 0.93, $P = 0.02$ in HGI-r6 and OR = 0.94, $P = 0.03$ in an independent meta-analysis[6]; Supplementary Table 3).

Previously, a 185-kb haplotype spanning *OAS1*, *OAS2* and *OAS3* was reported as introgressed from the Neandertal lineage and enriched in the genomes of modern humans[7–10]. This enrichment was suggested[9] to be driven by rs10774671, which controls splicing and generation of OAS1-p46 (by G allele) and OAS1-p42 (by A allele) protein isoforms[11]. We explored haplotypes comprised of four *OAS1* variants that captured the Neandertal haplotype (Supplementary Table 1), namely, rs10774671 and the three variants associated with COVID-19 hospitalization. The associated rs4767027 and rs1051042 were excluded from haplotype analyses due to lower imputation quality in patients of African ancestry ($r^2 = 0.67$ and 0.70); these variants were represented by directly genotyped rs4766664 ($r^2 > 0.9$ with rs4767027 in both ancestries) and rs2660 ($r^2 = 1.0$ with rs1051042 and rs1131476 in both ancestries).

In patients of European ancestry, the four selected variants formed only three haplotypes, with rs10774671 alleles tagging the predominantly ancestral/Neandertal-type non-risk (G<u>G</u>GT) and the predominantly derived/Denisova-type risk (A<u>A</u>AG) haplotypes (Fig. 1, Supplementary Table 4 and Extended Data Fig. 5). The protective G<u>G</u>GT haplotype was common (37.5% in nonhospitalized versus 32.6% in hospitalized patients) and associated with OR = 0.76, $P = 1.00 \times 10^{-3}$ compared to the risk A<u>A</u>AG haplotype.

In patients of African ancestry, the same variants formed four haplotypes. As in Europeans, the ancestral/Neandertal G<u>G</u>GT haplotype was the main haplotype protective from hospitalized disease (OR = 0.46, $P = 1.39 \times 10^{-3}$), despite being less common than in Europeans (10.2% in nonhospitalized and 5.6% in hospitalized patients; Supplementary Table 4 and Extended Data Fig. 5). The rs10774671-G allele was also included in a common African-specific haplotype (G<u>G</u>AG, OR = 0.76, $P = 0.059$; Fig. 1c, Supplementary Table 4 and Extended Data Fig. 5); the ORs of these haplotypes (0.46 versus 0.76) were not significantly different ($P = 0.19$). Thus, in both ancestries, the risk haplotype included the derived human-specific alleles rs1131454-A and rs10774671-A, whereas non-risk haplotypes were more variable but always included rs1131454-G and rs10774671-G alleles (Extended Data Fig. 5).

**OAS1 isoforms have comparable anti-SARS-CoV-2 activity.** Next, we explored the functional properties of the *OAS1* haplotypes associated with COVID-19 severity. The well-studied functional variant in this region, rs10774671, creates distinct protein isoforms OAS1-p42 (A-allele) and OAS1-p46 (G-allele) by regulating *OAS1* splicing[11]. This haplotype also includes several associated *OAS1* missense variants (rs1131454 (Gly162Ser), rs1131476 (Ala352Thr) and rs1051042 (Arg361Thr)), which might affect OAS1 anti-SARS-CoV-2 activity.

We generated four *OAS1* expression plasmids with rs10774671 and the three missense variants (Fig. 2a) and transiently overexpressed the plasmids in A549-ACE2, a lung epithelial cell line A549 stably expressing the SARS-CoV-2 receptor, ACE2. To control for endogenous interferon response to the plasmids, we transfected cells with either a control *GFP* plasmid or individual *OAS1* plasmids and,

---

**Fig. 1 | Association analyses within the chr12q24.13 region for COVID-19 hospitalization in patients of European and African ancestries. a,** Genomic region and association results (ORs) for 79 genotyped or confidently imputed ($r^2 > 0.8$) markers associated (logistic regression, $P < 0.05$) with hospitalized compared to nonhospitalized (mild) COVID-19 in patients of European (blue dots) or African (red dots) ancestries. The COVID-19 susceptibility GWAS lead SNP (rs10774671) is included, although it is not significantly associated in patients of African ancestry ($P = 0.079$). A blue highlight indicates the *OAS1* region with markers significantly associated in both ancestries. **b,** LD ($r^2$) plots of the region in COVID-19 patients of European and African ancestries. Darker shading in the plots indicates stronger correlations between markers. **c,** Single-marker and haplotype association analyses in patients with hospitalized compared to nonhospitalized COVID-19 performed with logistic regression and omnibus haplotype tests, respectively, controlling for sex, age, squared mean-centered age and 20 principal components. The GGGT haplotype comprised of ancestral alleles of the corresponding markers is shared with the Neandertal lineage of archaic humans and is protective from hospitalized COVID-19 in COVNET patients of European and African ancestries. Regional LD plots ($r^2$, 14-kb region) are shown for the *OAS1* region associated with protection from hospitalized COVID-19. Full association results for individual variants and haplotypes are provided in Supplementary Tables 1–4.

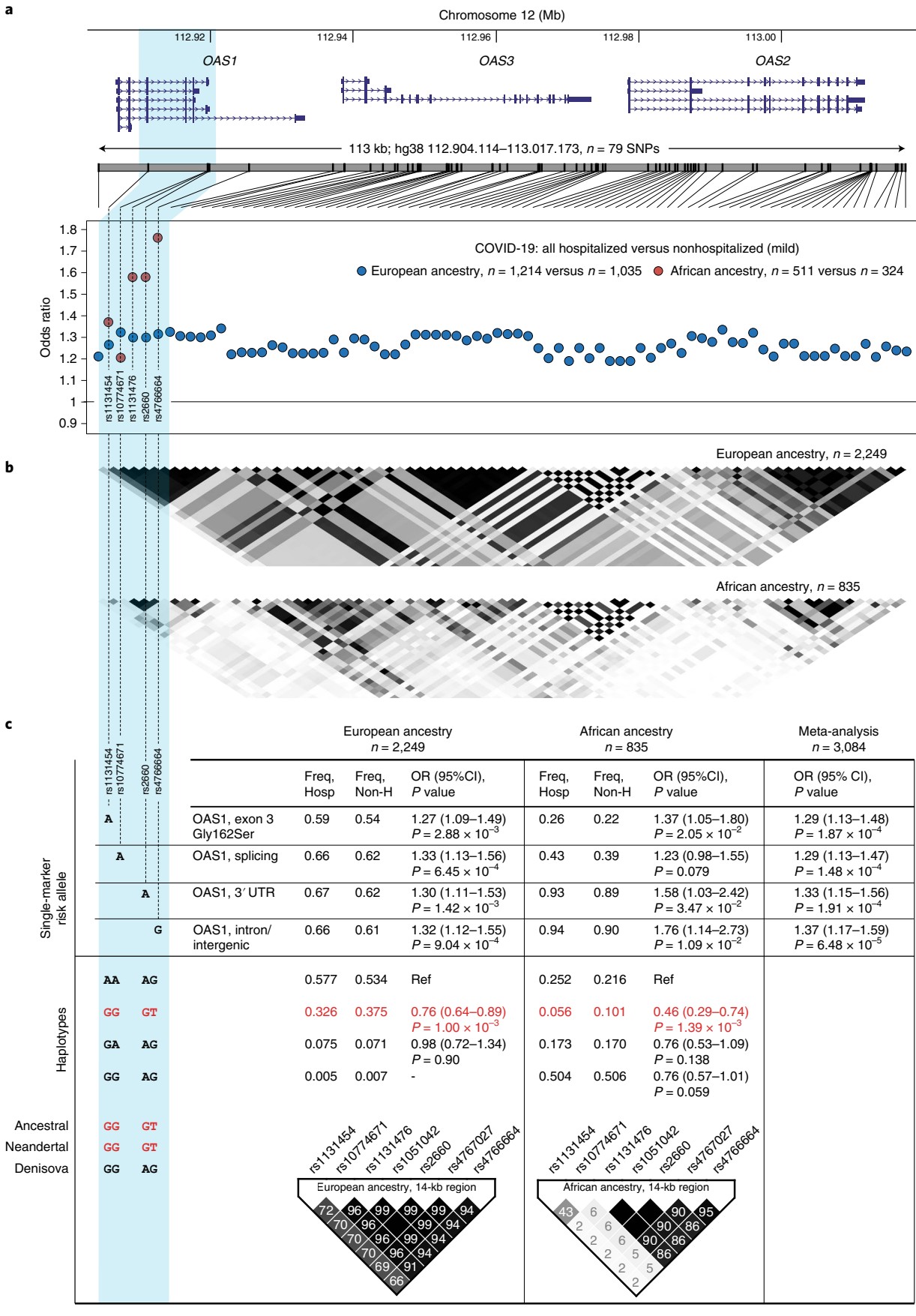

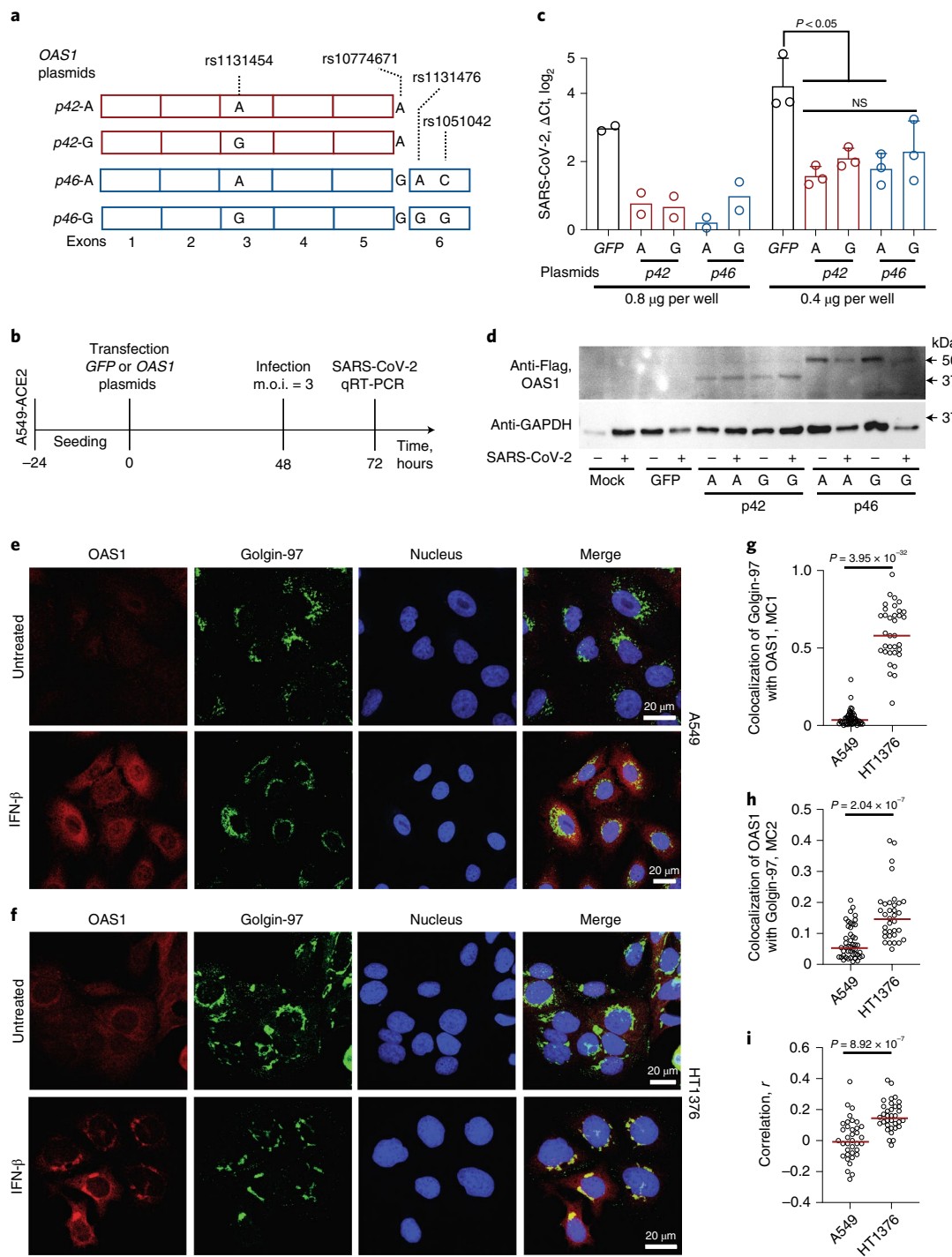

**Fig. 2 | Anti-SARS-CoV-2 activity and subcellular localization of OAS1-p42 and p46 isoforms. a**, Description of *OAS1-p42* and *OAS1-p46* plasmids with Flag-tags. **b**, Experimental outline: plasmids were transiently transfected in A549-ACE2 cells, followed by infection with SARS-CoV-2 and qRT-PCR for viral detection. **c**, SARS-CoV-2 load in A549-ACE2 cells transfected with *OAS1* or *GFP* plasmids in six-well plates with 0.4 μg per well ($n=2$, $P$ values and error bars are not applicable) or 0.8 μg per well ($n=3$, $P$ values are for unpaired, two-sided Student's $t$ tests, the data are presented as means and standard deviation (s.d.)). Expression of SARS-CoV-2 was detected by qRT-PCR and normalized to the expression of an endogenous control (*HPRT1*). Full results are presented in Supplementary Table 5. The experiment was independently repeated three times with comparable results; the results of one experiment are presented. **d**, A representative western blot showing similar expression of all Flag-tagged OAS1 protein isoforms in mock and SARS-CoV-2-infected A549-ACE2 cells, with GAPDH used as a loading control. **e,f**, Representative confocal images for endogenous OAS1 expression in untreated and interferon β (IFN-β)-treated A549 cells (rs10774671-AA, OAS1-p42, cytosolic expression) and HT1376 (rs10774671-GG, OAS1-p46, enrichment in trans-Golgi compartment); OAS1 (red), Golgin-97 (green) and nuclei (4,6-diamidino-2-phenylindole (DAPI), blue). Scale bars, 20 μm. **g**, Mander's coefficient 1 (MC1) for colocalization of Golgin-97 with OAS1 in confocal images. **h**, Mander's coefficient 2 (MC2) for colocalization of OAS1 with Golgin-97 in confocal images. **i**, Overall correlation (Pearson's $r$) between colocalization of Golgin-97 and OAS1 expression in confocal images. The results presented as individual points and group means are based on data collected from 5-7 fields of view from one of two comparable independent experiments. $P$ values are for nonparametric, two-sided Mann–Whitney $U$ tests. The full western blot is provided as Source Data.

after 48 h, infected these cells with SARS-CoV-2 for 24 h (Fig. 2b). The A549-ACE2 cells were highly infectable by SARS-CoV-2, but viral loads were similarly decreased by overexpression of all *OAS1* plasmids (Fig. 2c and Supplementary Table 5). Thus, in A549-ACE2 cells, proteins produced by similar amounts of *OAS1* plasmids (Fig. 2d) provided comparable anti-SARS-CoV-2 activity, without a detectable functional impact of the splicing (rs10774671) or missense (rs1131454, rs1131476 and rs1051042) variants (Fig. 2c,d). Additionally, all *OAS1* plasmids had similar effects on decreasing cell growth (Extended Data Fig. 6).

By confocal imaging in IFNβ-treated cells, we observed that endogenous OAS1-p46 produced in HT1376 cells (rs10774671-GG) was enriched in the trans-Golgi compartment, whereas endogenous OAS1-p42 produced in A549 cells (rs10774671-AA) appeared exclusively cytosolic (Fig. 2e–i), in line with previous reports[12,13]. Thus, despite differences in intracellular localization of OAS1 protein isoforms, determined by the splicing variant rs10774671, our results do not support significant differences in their anti-SARS-CoV-2 activity.

**rs1131454 regulates *OAS1* expression via a splicing enhancer.** One of the variants associated with COVID-19 severity in both ancestries, an *OAS1* intronic/intergenic variant rs4767027, has been reported as a protein quantitative trait locus (pQTL) for OAS1 blood levels in the European population[14]. The decrease in OAS1 levels was associated with rs4767027-C allele (linked with rs10774671-A risk allele in Europeans ($r^2 = 0.97$) but not Africans ($r^2 = 0.006$) in 1000 Genomes Project). Mendelian randomization analysis suggested that genetically regulated OAS1 deficiency could underlie the risk of severe COVID-19[14]. To explore whether the reported OAS1 protein deficiency[14,15] is affected by genetic regulation of messenger RNA (mRNA) expression, we quantified the expression of *OAS1* isoforms in various RNA-seq datasets. Expression of *OAS1-p46* (created by rs10774671-G allele) was higher than that of other *OAS1* isoforms in all datasets tested: adjacent normal tissues from The Cancer Genome Atlas (TCGA) (Extended Data Fig. 7), nasal epithelial cells infected with rhinovirus (Extended Data Fig. 8a), pulmonary alveolar type 1 cell-based organoids infected with SARS-CoV-2 (Extended Data Fig. 8b) and peripheral blood mononuclear cells (PBMCs) from COVID-19 patients (Extended Data Fig. 8c).

Higher expression of the *OAS1-p46* isoform could be due to its enhanced transcription regulated by rs10774671 or a linked variant. However, based on analyses of chromatin profiles in three cell lines by assay for transposase-accessible chromatin with high-throughput sequencing (ATAC-seq) and H3K27ac chromatin immunoprecipitation sequencing (ChIP-seq) or chromatin interactions by Hi-C, we did not observe evidence for regulatory elements within the associated region (Extended Data Fig. 9). Similarly, we did not detect chromatin interactions involving this region by Hi-C in a monocytic cell line THP-1 at baseline or after treatment with IFN-β (Extended Data Fig. 10). Therefore, we excluded transcriptional regulation of *OAS1* expression by rs10774671 or its linked variants and explored plausible posttranscriptional mechanisms.

As an indication of a potential imbalance in transcript production or stability, we evaluated the allele-specific expression of transcribed *OAS1* variants. Specifically, we analyzed heterozygous samples from nasal epithelial cells mock or in vitro infected with rhinovirus and PBMCs from COVID-19 and non-COVID-19 patients. Using allele-specific RNA sequencing (RNA-seq) reads, we evaluated two transcribed variants used in the COVNET haplotype analysis and linked with rs10774671: rs2660 in 3′ UTR ($r^2 = 0.96$ in Europeans, $r^2 = 0.016$ in Africans, included in *OAS1-p46* and *p48* transcripts) and rs1131454 in exon 3 ($r^2 = 0.72$ in Europeans, $r^2 = 0.30$ in Africans, included in all *OAS1* transcripts). For both variants, there were more RNA-seq reads with G than A alleles

(Fig. 3a). In most cases, rs1131454-G is linked with the rs10774671-G allele within the Neandertal haplotype, but rs1131454-G is also found in a less common haplotype with the risk rs10774671-A allele; the incomplete LD between these variants provides an opportunity to delineate their individual functional effects. We analyzed samples heterozygous for rs1131454 (AG) but homozygous for the risk alleles of rs10774671 (AA) and rs2660 (AA). Analysis of haplotypes of these three variants in different datasets showed higher expression in samples with the GAA compared to AAA haplotypes, indicative of allele-specific expression imbalance for rs1131454. The expression of both *OAS1-p46* and *OAS1-p42* transcripts was increased in the presence of exon 3 rs1131454-G allele (Fig. 3b–f).

By visual examination of RNA-seq plots, we noted different lengths of *OAS1* exon 3 (labeled as short and long; Fig. 4a), created due to alternative splicing through cryptic acceptor (~15–25% of reads) and donor (~5% of reads) splice sites. Splice quantitative trait locus analyses showed increased canonical splicing and reduced alternative splicing of exon 3 in the presence of the rs1131454-G allele (Fig. 4b–d). In silico analysis predicted that the rs1131454-G allele creates a putative exonic splicing enhancer/silencer (ESE/ESS; Fig. 5a). We generated allele-specific mini-genes with the A or G alleles of rs1131454 (Fig. 5b). By RT-PCR in two cell lines (A549 and T24), we detected increased splicing of long exon 3 in the mini-gene with the non-risk rs1131454-G allele (Fig. 5c,d). Based on these results, we conclude that the functional effect of the missense variant, rs1131454, could be related to creating an allele-specific ESE/ESS regulating splicing of *OAS1* short and long exon 3 isoforms.

***OAS1* transcripts are eliminated by NMD.** To explore the impact of exon 3 splicing on *OAS1* transcripts, we performed long-read Oxford Nanopore RNA-seq in A549 and HT1376 cells at baseline and after treatment with IFN-β (Supplementary Fig. 1). Reads with alternatively spliced exon 3 were included in all *OAS1* transcripts (Supplementary Fig. 1). The combination of *p42* and *p46* isoforms and short and long exon 3 created several transcripts; all of these transcripts, except for *p46*-long exon 3 isoform, had premature termination codons (Supplementary Fig. 2). Prematurely terminated transcripts might be targeted by NMD. To test this, we analyzed RNA-seq data[16] for HeLa cells (rs10774671-AA, *OAS1-p42*) after short interfering RNA (siRNA)-mediated knockdown (KD) of NMD-pathway genes *SMG6* and *SMG7*. The siRNA KD of *SMG6* and *SMG7* resulted in upregulation of both alternative (short) and canonical (long) isoforms of exon 3, confirming that all *OAS1-p42* transcripts are targeted by NMD (Fig. 6a,b).

To test whether increased expression of *OAS1-p46*-long could be due to protection from NMD, we performed siRNA KD of *SMG6*, *SMG7*, and *UPFI* (another component of the NMD pathway) in two cell lines, A549 (*OAS1-p42*) and HT1376 (*OAS1-p46*), and analyzed mRNA expression of long and short exon 3 as a readout (Fig. 6c). In A549 cells, the triple KD resulted in a significant increase in expression of both the short and long exon 3 versions of the *OAS1-p42* isoform (Fig. 6d,e). However, in HT1376 cells, we observed a modest significant increase of expression of *OAS1-p46*-short, but not *OAS1-p46*-long, isoform (Fig. 6f,g). We conclude that, unlike all other isoforms, the non-risk *OAS1-p46*-long isoform is NMD resistant. Because *OAS1* isoforms are created by haplotypes with rs1131454 and rs10774671 alleles, their combinations can contribute to the variation in NMD of *OAS1* transcripts.

**Interferons restore genetically impaired viral clearance.** Early interferon signaling is important for mounting an efficient antiviral response[17]. For some infections, such as with SARS-CoV-2, that either do not induce sufficient interferon response or use various ways to counteract it, this deficit in the magnitude or delayed timing of the response could be crucial[18–21]. Interferon treatment

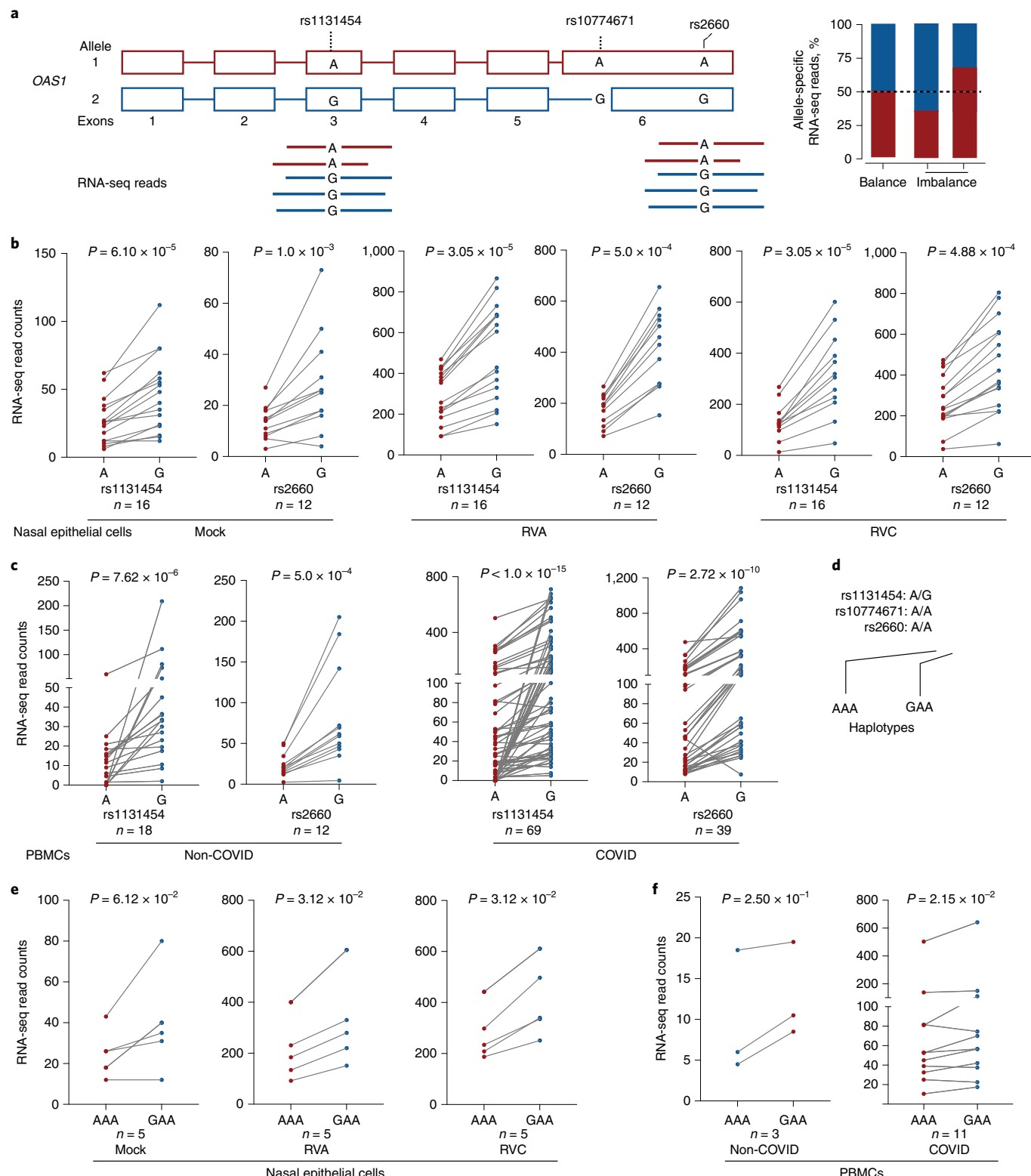

**Fig. 3 | Allelic expression imbalance of *OAS1* transcripts. a**, Analysis of allelic expression imbalance for transcribed *OAS1* variants based on RNA-seq reads. **b,c**, Counts of allele-specific RNA-seq reads in heterozygous samples for transcribed *OAS1* variants rs1131454 and rs2660 in nasal epithelial cells uninfected (Mock) and infected with rhinovirus (RV) strains A or C (RVA or RVC) (b) or in PBMCs from patients with COVID-19 and healthy controls (labeled as non-COVID) (c). **d**, Haplotypes of the three *OAS1* variants used for analysis. **e**, Haplotype-specific imbalance in *OAS1* expression contributed by rs1131454. All *P* values are for nonparametric, Wilcoxon matched-pairs two-sided signed-rank tests.

has been proposed for mitigating SARS-CoV-2 infection, especially at an early infection stage[22,23]. To explore whether interferon treatment could prevent or reduce infection in vitro, we treated

infection-permissive intestinal cells Caco2[24,25] (heterozygous for *OAS1* rs10774671 and rs1131454) with IFN-β or IFN-λ 4 h before or after SARS-CoV-2 infection.

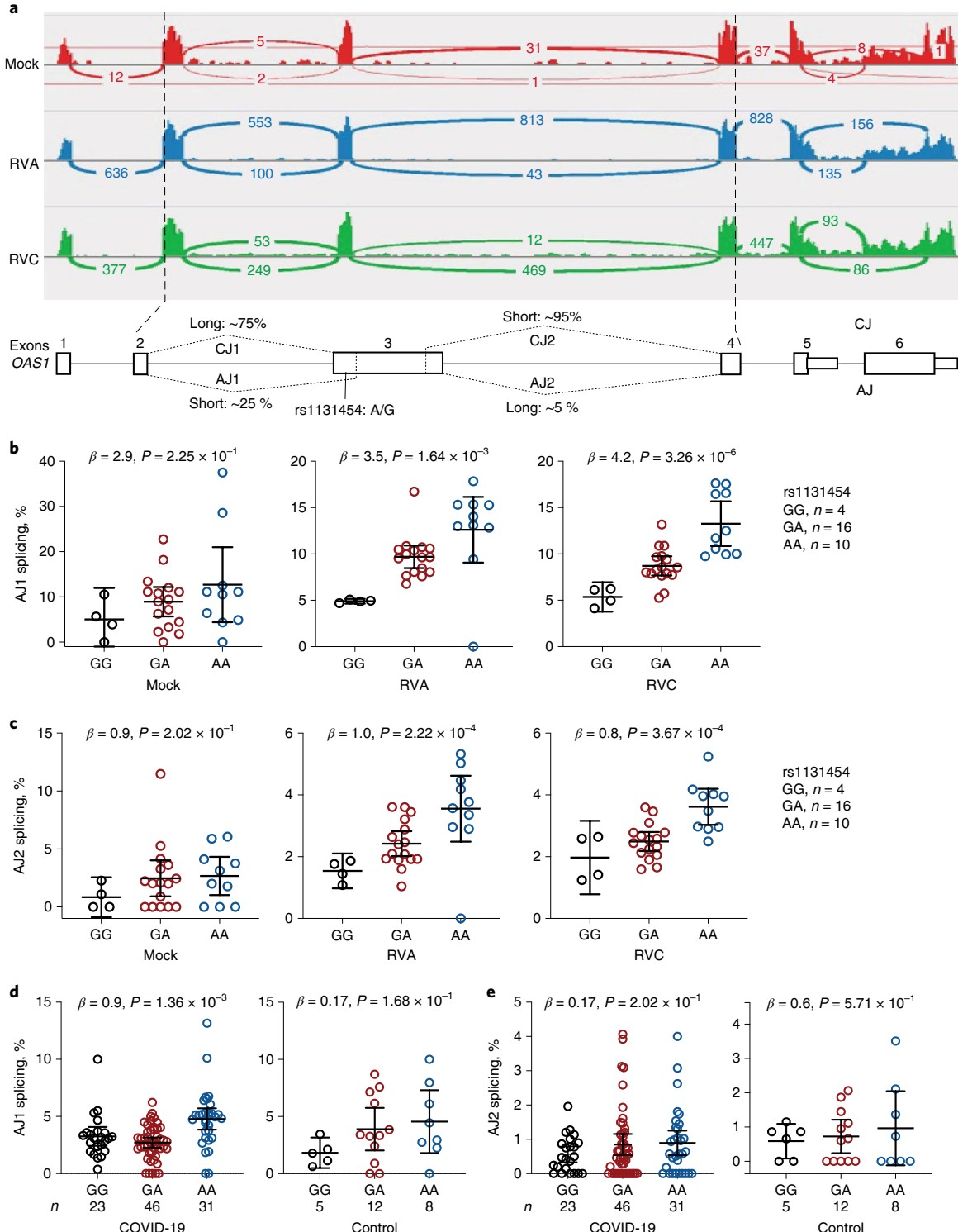

**Fig. 4 | Splicing of *OAS1* exon 3 is associated with rs1131454 alleles. a**, RNA-seq plots showing splicing patterns of *OAS1* exons in representative samples from nasal epithelial cells uninfected (Mock) and rhinovirus (RV)-infected with strains A or C (RVA or RVC). *OAS1* exon 3 shows alternative splicing at both 5′ acceptor and 3′ donor splice sites, resulting in four splicing junctions: two canonical junctions (CJ) and two alternative junctions (AJ) producing long and short versions of exon 3. Exon junctions AJ1 (major) and AJ2 (minor) account for approximately 25% and 5% of total RNA-seq reads, respectively. **b**–**e**, Splice quantitative trait locus (sQTL) analysis of AJ1 and AJ2 with rs1131454 in nasal epithelial cells uninfected (Mock) or infected with RVA or RVC (**b**,**c**) and in PBMCs from COVID-19 patients and healthy controls (labeled as non-COVID) (**d**,**e**). For b–e, *P* values are for linear regressions, adjusting for sex and age. All graphs show individual data points with means and 95% confidence intervals (CIs).

All treatments significantly decreased viral loads (Fig. 7a,b and Supplementary Table 6). In the same cells, *OAS1* expression was minimally induced by SARS-CoV-2 alone but was strongly induced by before and after treatment with both interferons (Fig. 7c and Supplementary Table 7). Interferon-induced expression of both OAS1-p42 and p46 isoforms was also detectable by

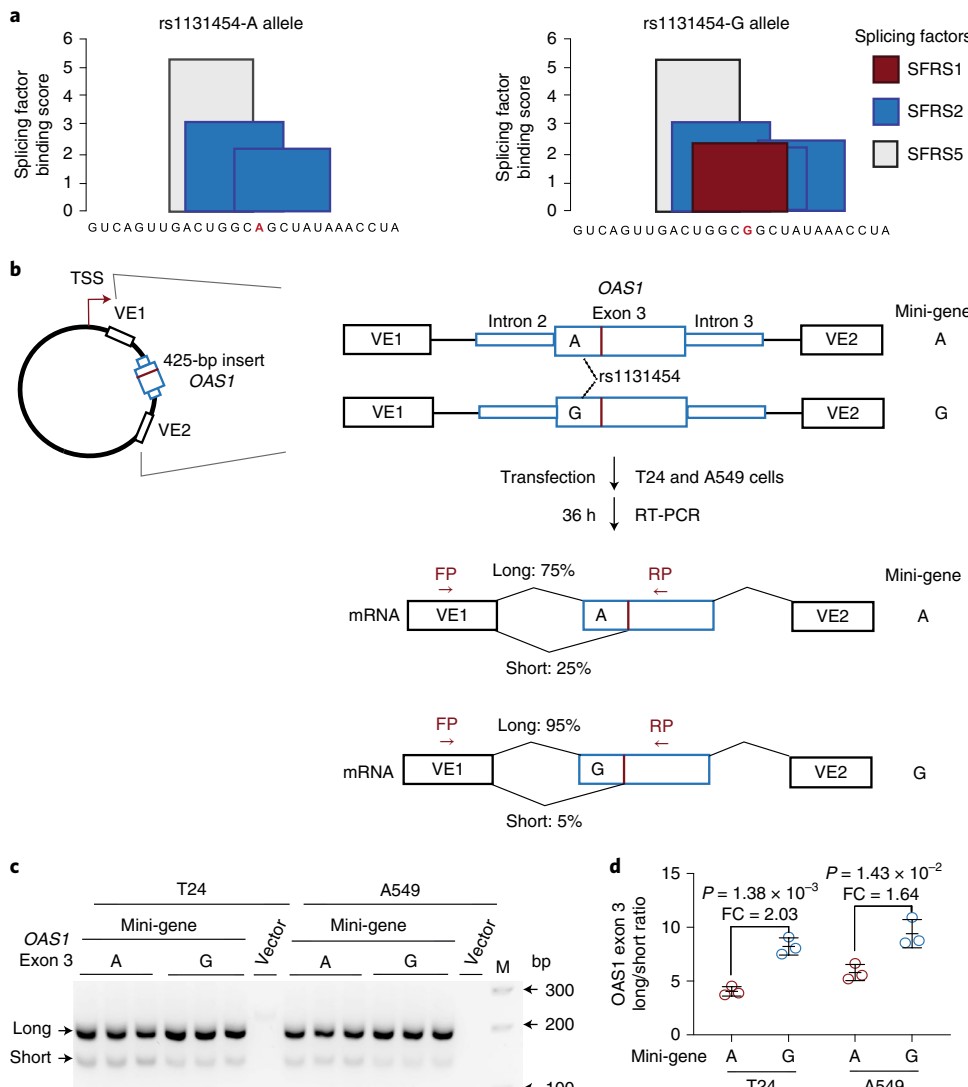

**Fig. 5 | Exontrap assays demonstrate the functional effect of rs1131454 on *OAS1* exon 3 splicing. a**, In silico prediction of allele-specific splicing factor binding sites within *OAS1* exon 3. Only rs1131454-G allele creates a binding site for SFRS1 splicing factor; binding sites for SFRS2 are created by both alleles, with three or two sites created in the presence of non-risk G or risk A alleles, respectively. **b**, Experimental outline: description of allele-specific mini-genes with *OAS1* exon 3 inserts, transfection in T24 and A549 cells, and splicing ratios of amplicons detected by RT-PCR with FP and RP primers. **c**, Representative agarose gel showing splicing events of mini-genes detected by RT-PCR in T24 and A549 cells. Vector corresponds to negative control, and M corresponds to 100-bp size marker. Upper and lower bands correspond to long and short exon 3 splicing events with vector exon 1 (VE1). No alternative splicing events were identified between exon 3 insert and VE2. Each mini-gene was analyzed in three biological replicates, and the results of one of two independent experiments are shown. **d**, The ratios of long/short *OAS1* exon 3 expression quantified by densitometry of agarose gel bands. Splicing of long exon 3 is significantly higher from the mini-gene with non-risk rs1131454-G allele compared to the mini-gene with risk rs1131454-A allele. Fold changes (FC) were calculated from the splicing ratios. The dot plots are presented with means and s.d.; *P* values are for unpaired, two-sided Student's *t* tests. The full agarose gel is provided as Source Data.

western blotting (Supplementary Fig. 3). These results suggested that interferons can generate robust immune response even in SARS-CoV-2-infected cells, inducing expression of both OAS1-p42 and OAS1-p46 isoforms and compensating for any loss of their expression due to NMD.

To investigate whether this can be relevant in clinical settings, we analyzed data from a clinical trial in which patients with nonhospitalized COVID-19 were treated with a single subcutaneous injection of pegIFN-λ1 or saline placebo[26]. In this preliminary study of 58 patients, we genotyped three variants that capture *OAS1* haplotypes associated with COVID-19 hospitalization in our case–case analyses: rs1131454, rs10774671, and rs2660. In multivariable analyses, SARS-CoV-2 viral load at

day 0 was not associated with age, sex, race, treatment group or individual *OAS1* variants or haplotypes (Fig. 7d and Supplementary Table 8). In analyses with linear mixed models, longitudinal viral load was significantly associated with *OAS1* variants but differentially in treatment groups (analysis of variance (ANOVA), *P* = 0.02; Supplementary Tables 9 and 10). The AAA haplotype comprised of risk alleles of these variants (*OAS1-p42*-A isoform) was associated with the least efficient viral clearance in the placebo (28 patients, haplotype frequency 0.46), but not the treatment group (30 patients, haplotype frequency 0.57) (Fig. 7d and Supplementary Table 8). Thus, pegIFN-λ1 treatment overcame the deficit in *OAS1* expression associated with the *OAS1*-AAA risk haplotype.

## Discussion

Interindividual variability in response to SARS-CoV-2 infection ranging from asymptomatic to fatal disease remains poorly understood. The chr12q24.13 region was reported as a susceptibility locus when patients with COVID-19 were compared to general population controls[1,2,6]. Here we focused on possible role of the chr12q24.13 region in outcomes of laboratory-confirmed SARS-CoV-2 infection. We demonstrated that a common haplotype comprised of derived human-specific risk alleles of two *OAS1* variants is associated with the risk of hospitalized COVID-19 in patients of European and African ancestries, compared to nonhospitalized patients. We provide evidence for the combined functional contribution of these variants — a splicing rs10774671 and a missense rs1131454 in exon 3 — on the expression of OAS1, an antiviral protein critical for SARS-CoV-2 clearance. Thus, genetically regulated *OAS1* expression contributes to association with SARS-CoV-2 clearance and risk of hospitalization for COVID-19. Our exploratory analyses suggest that the deficiency in *OAS1* expression in individuals with the *OAS1* risk haplotype could be compensated by early treatment with pegIFN-λ1.

Of the three proteins (OAS1, OAS2 and OAS3) encoded within the 12q24.13 region, only OAS1 was found to be functionally critical for anti-SARS-CoV-2 activity[12]. *OAS1*-rs10774671 was proposed as the primary functional candidate in this region. Specifically, the COVID-19 susceptibility rs10774671-A allele, as well as linked alleles of many other variants in this region, was associated with decreased basal OAS enzymatic activity in unstimulated PBMCs from healthy individuals[11], impaired clearance of West Nile virus infection[27], chronic hepatitis C virus infection[28], impaired SARS-CoV clearance[29], and increased susceptibility to several autoimmune conditions (multiple sclerosis[30,31], Sjögren's syndrome[32] and systemic lupus erythematosus[33,34]). However, due to high LD in this region in populations of European and Asian ancestries explored in these studies, more than 100 variants within the *OAS1-OAS2-OAS3* region would provide comparable association results, making it hard to pinpoint the causal variant(s).

Functionally, rs10774671 creates distinct OAS1 protein isoforms, mainly OAS1-p42 versus OAS1-p46 by the risk A and non-risk G allele, respectively[11,27]. The OAS1 protein isoforms do not differ by their enzymatic activities and similarly activate the antiviral RNase L pathway[12,13,35,36]. However, OAS1-p46 expression is enriched in trans-Golgi compartments, whereas OAS1-p42 is predominantly expressed in the cytosol[12,13]. Targeting OAS1 to endomembrane structures could benefit response to pathogens that hide from cytosolic pattern recognition receptors[37]. Virus-induced formation of complex membrane rearrangements originating from the endoplasmic reticulum was demonstrated as the mechanism used by plus-strand RNA flaviviruses (such as West Nile, hepatitis C, Dengue, Yellow Fever and Zika) to evade sensing and elimination[37].

It was hypothesized that coronaviruses, including SARS-CoV-2, might use the same mechanisms as flaviviruses to evade triggering the intracellular immune response[12,13]. However, there is considerable variability in reported in vitro anti-SARS-CoV-2 activity of OAS1-p42 and OAS1-p46 protein isoforms. In one study, both isoforms were active, but OAS1-p46 was 5-fold more efficient than OAS1-p42 in clearing SARS-CoV-2[13]. Another study reported no detectable anti-SARS-CoV-2 activity of OAS1-p42, attributing all the activity to OAS1-p46 alone[12]. In our experimental model of the A549-ACE2 lung epithelial cell line, similar amounts of all *OAS1* plasmids representing these isoforms with relevant missense variants provided similar anti-SARS-CoV-2 responses. Comprehensive analyses in comparable experimental models and conditions will be required to reconcile these differences in reported results.

Introgression of the 185-kb haplotype with the Neandertal versions of the *OAS1*, *OAS2* and *OAS3* genes[7–10] was hypothesized to protect modern humans from some deadly pathogens[9]. Several associations with immune-related phenotypes reported for individual variants comprising this haplotype supported this idea[11,27–34]. Based on the COVID-19 GWAS meta-analysis, a 97-kb block of linked variants ($r^2 > 0.8$) associated with the hospitalized disease compared to the general population was nominated as a potentially protective part of the Neandertal haplotype[10].

It is intriguing to evaluate the structure and origin of the *OAS1* haplotypes in European and African ancestries in light of the relationship between the ancestral and derived alleles of modern and archaic humans (Neandertal and Denisova lineages) (Extended Data Fig. 5). In our study, the risk of hospitalization for COVID-19 in patients of European and African ancestries was associated with a haplotype encompassing part of the *OAS1* region (~14 kb). This risk haplotype included the derived human-specific alleles rs10774671-A and rs1131454-A not found in Neandertal or Denisova lineages; these alleles might have created human-specific vulnerability to some pathogens. A separate part of the risk haplotype included a block of four derived and one ancestral allele that reached complete fixation in Africa. This part of the risk haplotype is also present in the Denisova lineage and might be a product of independent evolution or adaptive introgression.

All non-risk haplotypes included non-risk rs10774671-G and rs1131454-G alleles, which are ancestral and also shared with Neandertal and Denisova lineages. The main non-risk haplotype shared by Europeans and Africans also included the ancestral/Neandertal alleles (rs1131476-G, rs1051042-G, rs2660-G and rs4766664-T) and a derived/Neandertal allele (rs4767027-T). Surprisingly, the predominantly ancestral haplotype with three missense *OAS1* variants is absent in all African populations in 1000 Genomes Project but present in individuals of African American ancestry (AWS in 1000 Genomes Project and our COVID-19 patients of African ancestry; Supplementary Table 4), apparently due to admixture with non-African populations.

A separate non-risk haplotype that exists only in Africa includes a part of the risk/Denisova-type fragment, which narrows the shared non-risk part of this haplotype to rs10774671-G and rs1131454-G alleles. Although it has been suggested that the Neandertal haplotype protects from COVID-19 (refs. [7–10]), our study offers a different interpretation. Specifically, the emergence of derived human-specific alleles rs10774671-A and rs1131454-A might have decreased *OAS1* expression compared to all other versions of the gene (ancestral, Neandertal and Denisova), increasing the human-specific risk of severe COVID-19. Despite this detrimental effect on response

**Fig. 6 | NMD targets *OAS1* isoforms with short exon 3 upregulated by the risk rs1131454-A allele. a**, RNA-seq plots showing *OAS1* expression and splicing patterns in HeLa cells (rs10774671-AA, *OAS1-p42*) targeted by siRNA KD of NMD-pathway genes *SMG6* and *SMG7* or by scrambled siRNA (Neg Ctrl). **b**, Expression of both long and short *OAS1-p42* isoforms is increased in HeLa cells with siRNA KD of NMD genes. **c**, Schematics for characterizing effects of KD of NMD genes on the expression of long and short exon 3 in the context of *OAS1-p42* (in A549) and *OAS1-p46* (in HT1376). **d,e**, Downregulation of NMD-pathway genes (*SMG6*, *SMG7* and *UPF1*) targeted by siRNA KD in A549 cells. Expression of both long and short exon 3 *OAS1-p42* isoforms is increased in cells with siRNA KD of NMD genes. Tri-KD, triple KD. **f,g**, Downregulation of NMD-pathway genes (*SMG6*, *SMG7* and *UPF1*) targeted by siRNA-KD in HT1376 cells. Only expression of short-exon 3 *OAS1-p46* isoform is increased by siRNA-KD of NMD genes. Expression in three biological replicates was analyzed by qRT-PCR and normalized to an endogenous control (*HPRT1*). The dot plots are presented with means and s.d.; *P* values are for unpaired, two-sided Student's *t* tests.

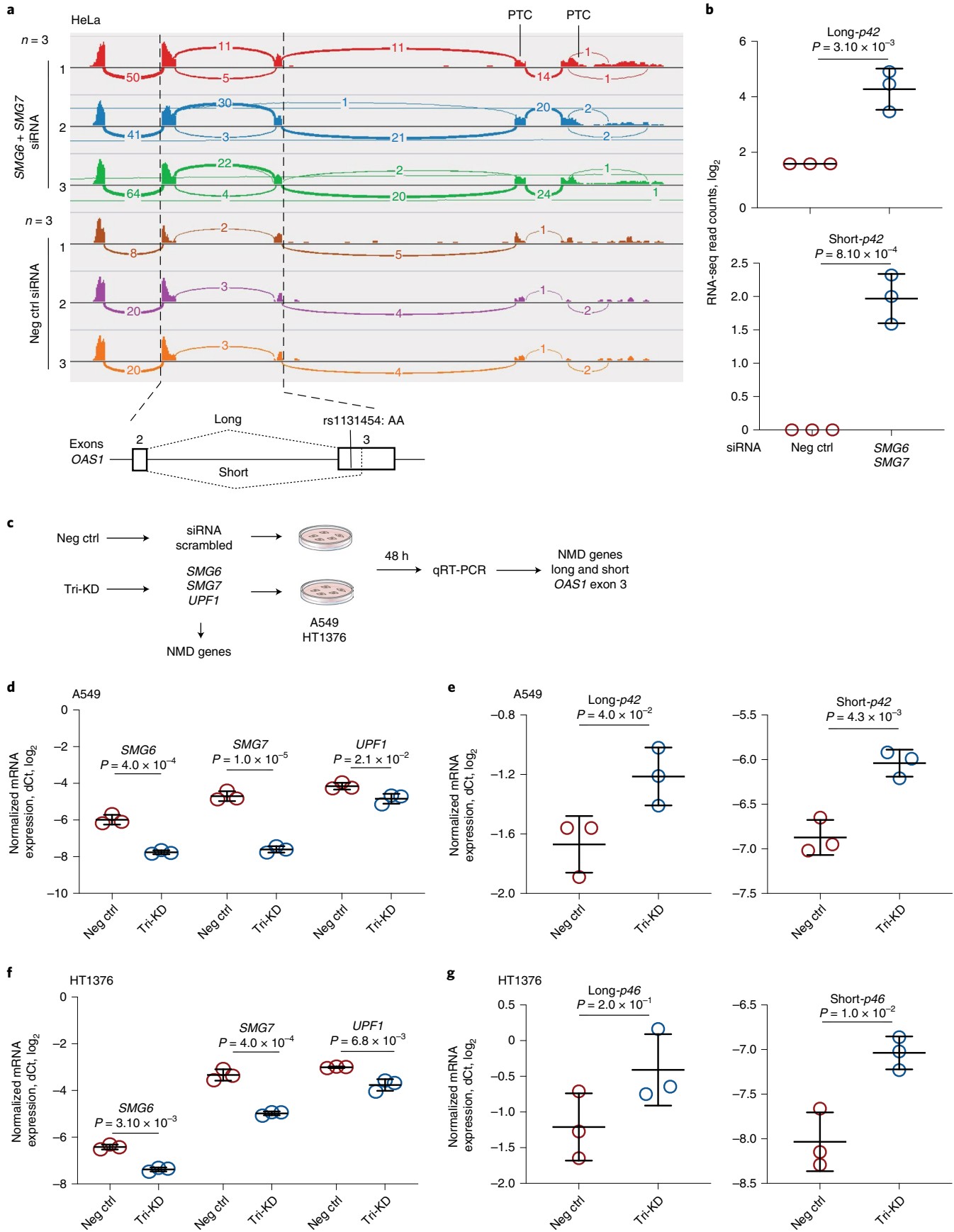

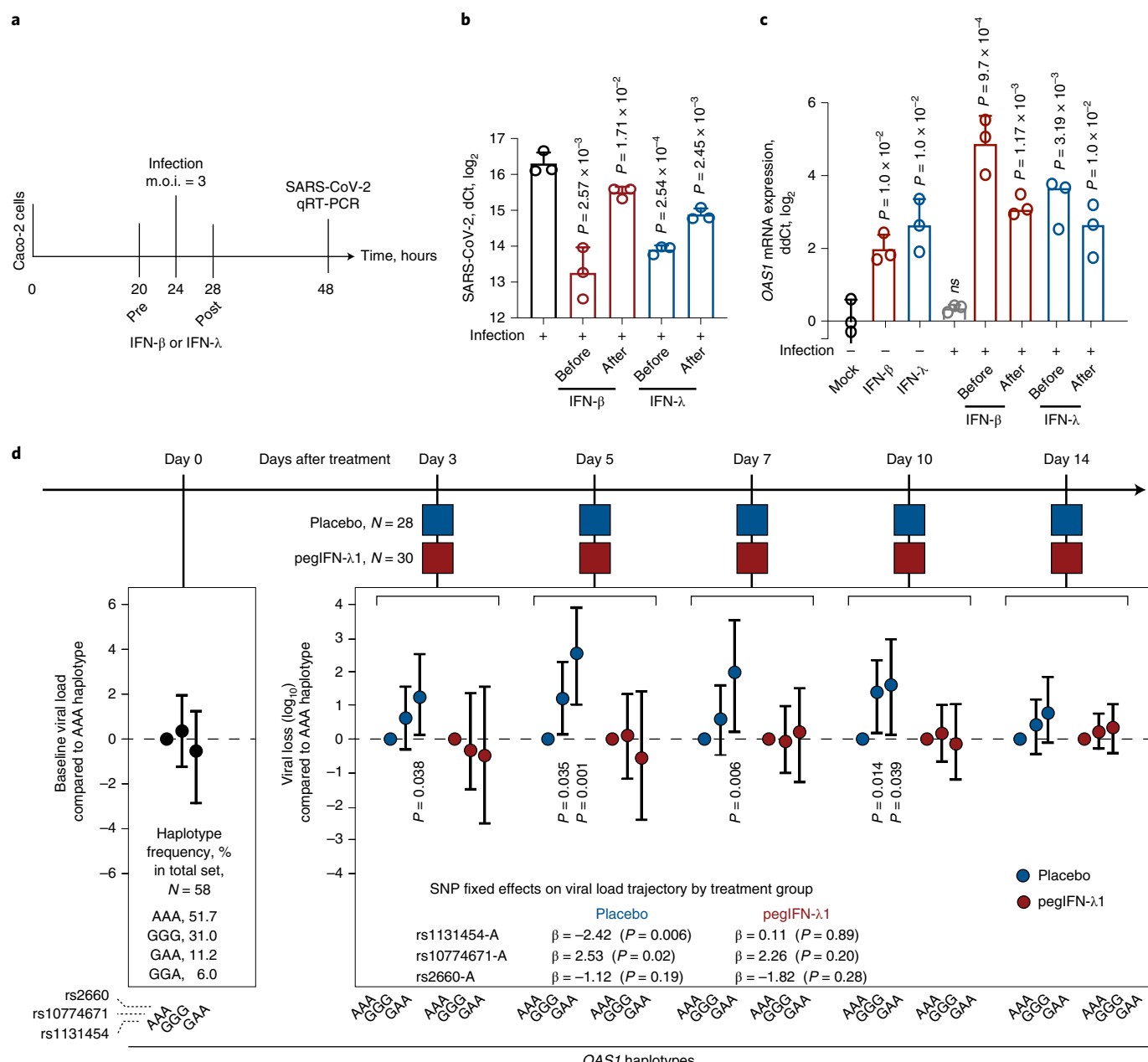

**Fig. 7 | Effects of interferons on SARS-CoV-2 viral loads *in vitro* and a clinical trial. a**, Outline of an experiment in Caco2 cell line. Cells were infected with SARS-CoV-2 and treated with IFN-β or IFN-λ 4 h before or after infection, and SARS-CoV-2 and *OAS1* expression was measured by qRT-PCR 24 h after infection. **b,c**, Expression of SARS-CoV-2 (b) and *OAS1* (c) normalized by the expression of endogenous control (*HPRT1*). *P* values are for comparison with infection alone or Mock, with three biological replicates for each condition. The dot plots are presented with means and s.d.; *P* values are for unpaired, two-sided Student's *t* tests. **d**, Outline of the clinical trial: a single subcutaneous injection of 180 μg pegIFN-λ1 (*n* = 30) or saline placebo (*n* = 28) was administered at day 0, and longitudinal trajectory of SARS-CoV-2 load (log₁₀ copies per ml of blood) was evaluated at indicated days compared to day 0 using linear mixed-effect models. The analysis included genotypes of *OAS1* variants rs1131454, rs10774671 and rs2660 used in haplotype analyses of COVID-19 severity. The model that included genotypes of these variants in interaction with treatment arms showed significantly better fit (two-way ANOVA *P* = 0.02; likelihood ratio test degrees of freedom (d.f.) = 3) compared to the base model, justifying analysis stratified by treatment arms. In the placebo group, the rs1131454-A risk allele was most significantly associated with less efficient viral loss (*P* = 0.006). Results are also presented as a post-hoc analysis for indicated haplotypes as viral loss (log₁₀) at specific days compared to day 0, using the risk AAA haplotype as a reference. The results are presented as point estimates (β, with 95% CI); *P* values are for omnibus haplotype test, adjusting for sex, age and viral load at day 0. Haplotypes are not associated with viral load at day 0. Full results are presented in Supplementary Tables 6–10.

to infection, the derived alleles rs10774671-A and rs1131454-A became major alleles in European and Asian populations, perhaps by providing benefit in noninfectious conditions. Activation of the innate immune response comes with the cost of decreased fitness, longevity and fecundity in noninfectious conditions[36]; thus,

mechanisms restricting overactivation of the immune response are likely to be under positive or balancing selection.

We propose that rs10774671 and rs1131454 functionally contribute to the association with COVID-19 severity by regulating the abundance of OAS1 protein. In all datasets explored, mRNA

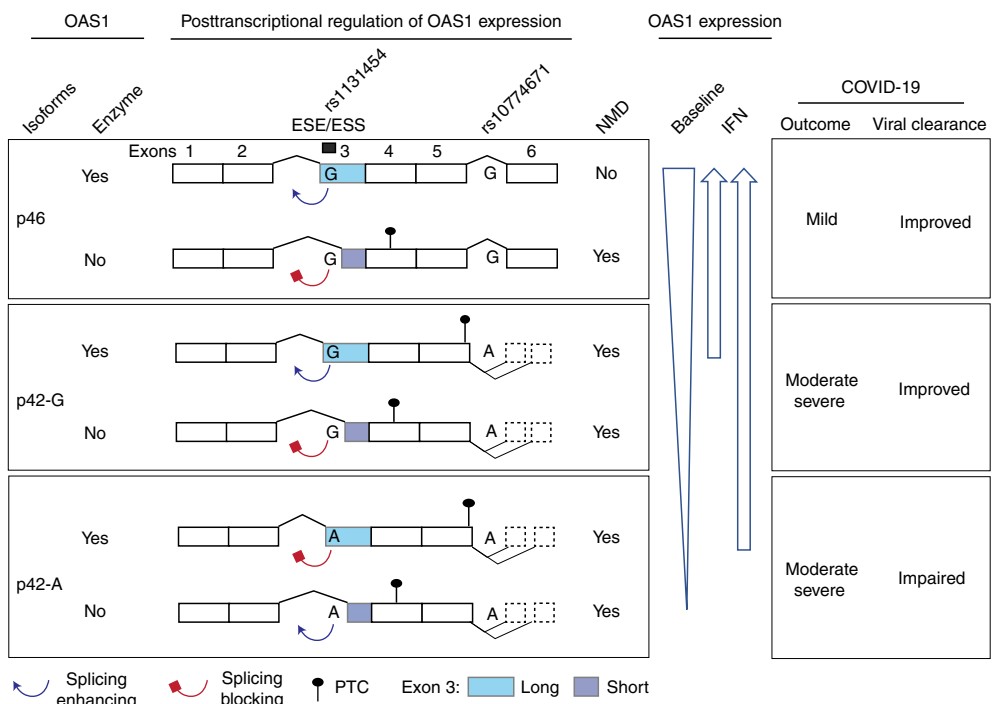

**Fig. 8 | Proposed model for mechanisms underlying association between *OAS1* genetic variants and COVID-19 outcomes.** Two *OAS1* variants, the splice site variant (rs10774671-A/G) and exon 3 missense variant (rs1131454-A/G, Gly162Ser), determine the structure and expression levels of *OAS1* isoforms. Alleles of the splicing variant rs10774671 define the *OAS1* isoforms *OAS1-p42* (A allele) and *OAS1-p46* (G allele). Alleles of rs1131454 create an ESE/ESS for splicing of the canonical/long versus alternative/short exon 3. Transcripts with short exon 3 are terminated by PTCs within exon 4 and efficiently targeted by NMD. The stop codon for *OAS1-p42* is located within exon 5, followed by several additional exons creating *OAS1-p44* and *OAS1-p48* isoforms, making this stop codon a PTC. Thus, *OAS1-p42* is also targeted by NMD, albeit less efficiently than transcripts with PTCs in exon 4. The combined splicing effects of rs10774671, which creates alternative *OAS1* isoforms, and rs1131454, which regulates the inclusion of short or long exon 3 and thus the introduction of additional PTCs, result in variable degradation of *OAS1* transcripts by NMD. At baseline, the expression is highest for *OAS1-p46* and the lowest for *OAS1-p42*-A. However, treatment with interferons may compensate for NMD, allowing OAS1-p42 protein to reach expression levels comparable to OAS1-p46. Thus, the effects of genetic variants on OAS1 expression can be compensated by IFN treatment to overcome impaired viral clearance and prevent progression to severe COVID-19 requiring hospitalization.

expression of *OAS1-p46* transcript was on average 3.9-fold higher than that of *OAS1-p42*. We did not find evidence for transcriptional regulation of *OAS1* expression but demonstrated that *OAS1* expression is regulated by NMD differentially affecting *OAS1* isoforms. By creating an NMD-resistant *OAS1*-p46 transcript, the non-risk rs10774671-G allele plays a major role in preserving *OAS1* expression. Additionally, we identified rs1131454 within exon 3 of *OAS1* as the second variant contributing to the regulation of *OAS1* NMD. Although rs1131454 is a missense variant in exon 3 (Gly162Ser), it did not appear to have a functional impact on the enzymatic activity of recombinant OAS1 proteins[38] and in our anti-SARS-CoV-2 assays. Instead, the non-risk rs1131454-G allele of this variant creates an ESE/ESS, which regulates the inclusion of the short versus long forms of exon 3. NMD targets all *OAS1-p42* transcripts; however, transcripts with rs1131454-G allele are partially rescued from NMD. Because the *OAS1-p46* transcripts always carry the rs1131454-G allele, they are the most NMD resistant.

We observed a decrease in SARS-CoV-2 expression after treating cells with interferons (either IFN-β or IFN-λ) before or after infection, suggesting that interferons can overcome insufficient viral clearance. Indeed, our exploratory analysis of a clinical trial with pegIFN-λ1 revealed that *OAS1* haplotypes were associated with the rate of SARS-CoV-2 clearance in the placebo group, but not the interferon treatment group. Thus, our results suggest that early treatment with interferons could compensate for SARS-CoV-2 clearance impaired due to *OAS1* variants. Although this treatment accelerated viral clearance in all patients, those with the risk *OAS1* haplotype

(AAA for rs1131454-A, rs10774671-A and rs2660-A) would benefit from this treatment the most because of their impaired ability to clear the virus without treatment. This haplotype is very common, with a 57% frequency in the general European population, 59% in East-Asian individuals and 15% in individuals of African ancestry. In our clinical trial, patients were treated with a single subcutaneous injection of pegIFN-λ1 (ref. [26]). Due to the restricted expression of its receptors, IFN-λ1, a type III interferon, is well tolerated, without causing systemic side effects or promoting inflammatory cytokine release, which are often associated with the administration of type I interferons[39]. Recently, preliminary results of a phase 3 clinical trial (TOGETHER trial, NCT04727424, 1,936 COVID-19 outpatients treated with pegIFN-λ1) showed a significant reduction of COVID-19-related hospitalization and death[40]. Inhaled nebulized type I interferons, IFNβ-1a[41] and IFNα2b[42,43], are also being tested as an early treatment for SARS-CoV-2 infection, with promising results.

The strengths of our study include genetic analyses evaluating outcomes in patients with laboratory-confirmed SARS-CoV-2 infection of European and African ancestries and integrated analyses of multiple genomic datasets (for example, RNA-seq, ATAC-seq, ChIP-seq and Hi-C) supported by experimental testing of our hypotheses. In addition, we analyzed *OAS1* haplotypes in association studies with COVID-19 severity and a clinical trial with pegIFN-λ1. The extensive multimethod investigation provides strong plausibility for our findings. Of multiple associated genetic variants, we identified rs10774671 and rs1131454 as the most functional within *OAS1* for COVID-19 severity and SARS-CoV-2

clearance. However, we cannot exclude additional functional variants, especially in non-European populations, in which we had lower statistical power for genetic analyses. We functionally annotated several additional variants within *OAS1* and *OAS3* that were significantly associated with COVID-19 hospitalization in patients of European ancestry but did not yet reach significance in patients of African ancestry (Supplementary Figs. 4 and 5). Our genetic analyses included patients recruited before vaccination and the emergence of the viral variants.

In conclusion, we propose that non-risk alleles of two variants (rs10774671-G and rs1131454-G) protect *OAS1* transcripts from NMD (Fig. 8). The primary functional effect is contributed by the rs10774671-G allele, which generates the *OAS1*-p46 isoform. At the same time, rs1131454-G additionally and independently contributes by creating an ESE that increases inclusion of long exon 3, thus protecting both *OAS1-p46* and *OAS1-p42* from elimination by NMD. The non-risk G alleles of both variants create the most abundant and NMD-resistant *OAS1* isoform (*OAS1-p46*-long). In contrast, the risk A alleles of both variants create the NMD-vulnerable and low-expressed isoform (*OAS1-p42*), whereas the haplotype with rs10774671-A but rs1131454-G allele creates the *OAS1-p42* isoform with an intermediate NMD resistance and expression levels (Fig. 8). Deficient viral clearance in individuals carrying the risk *OAS1* haplotype can be compensated by early treatment with interferons, which should be further explored in clinical trials.

## Online content

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

[1]Laboratory of Translational Genomics, Division of Cancer Epidemiology and Genetics, National Cancer Institute, Rockville, MD, USA. [2]Department of Infectious Diseases, Molecular Virology, University Hospital Heidelberg, Heidelberg, Germany. [3]Department of Molecular Genetics and Microbiology, College of Medicine, University of Florida, Gainesville, FL, USA. [4]Toronto Centre for Liver Disease, Toronto General Hospital Research Institute, University Health Network, Toronto, Ontario, Canada. [5]Clinical Genetics Branch, Division of Cancer Epidemiology and Genetics, National Cancer Institute, Rockville, MD, USA. [6]Biostatistics Branch, Division of Cancer Epidemiology and Genetics, National Cancer Institute, Rockville, MD, USA. [7]Laboratory of Immunobiology, Center for Clinical, Experimental Surgery and Translational Research, Biomedical Research Foundation of the Academy of Athens, Athens, Greece. [8]Laboratory of Molecular Biology, Center for Cancer Research, National Cancer Institute, Bethesda, MD, USA. [9]HudsonAlpha Institute for Biotechnology, Huntsville, AL, USA. [10]Center for Precision Health Research, National Human Genome Research Institute, Bethesda, MD, USA. [11]Altman Clinical & Translational Research Institute, UC San Diego Health Sciences, San Diego, CA, USA. [12]Georgetown University School of Medicine, Washington, DC, USA. [13]Institute for Health Research, Kaiser Permanente Colorado, Aurora, CO, USA. [14]Basic Science Program, Frederick National Laboratory for Cancer Research, National Cancer Institute, Frederick, MD, USA. [15]Laboratory of Integrative Cancer Immunology, Center for Cancer Research, National Cancer Institute, Bethesda, MD, USA. [16]Ragon Institute of MGH, MIT and Harvard, Cambridge, MA, USA. [17]Department of Internal Medicine, Seoul National University College of Medicine, Seoul, Republic of Korea. [18]Center for Genetic Medicine, Northwestern University Feinberg School of Medicine, Chicago, IL, USA. [19]Department of Medical Imaging, Hematology, and Oncology, Ribeirão Preto Medical School, University of São Paulo, Ribeirão Preto, Brazil. [20]Uniformed Services University of the Health Sciences, Bethesda, MD, USA. [21]Department of Gynecology and Obstetrics, Emory University School of Medicine, Atlanta, GA, USA. [22]Department of Medicine, Division of Clinical Immunology and Rheumatology, University of Alabama at Birmingham, Birmingham, AL, USA. [23]Department of Medicine, Division of Infectious Diseases, University of Alabama at Birmingham, Birmingham, AL, USA. [24]Department of Internal Medicine, Ribeirão Preto Medical School, University of São Paulo, Ribeirão Preto, Brazil. [25]Department of Immunology, University of Toronto, Toronto, Ontario, Canada. [26]Division of Cellular Polarity and Viral Infection, German Cancer Research Center (DKFZ), Heidelberg, Germany. [27]Laboratory of Clinical Immunology and Microbiology, National Institute of Allergy and Infectious Diseases, Bethesda, MD, USA. [28]Cancer Genomics Research Laboratory, Frederick National Laboratory for Cancer Research, Frederick, MD, USA. [29]Genome Opinion, Inc., Seoul, Republic of Korea. [30]Divisions of Infectious Diseases and Organ Transplantation, Northwestern University Feinberg School of Medicine, Chicago, IL, USA. [31]Department of Obstetrics & Gynecology, Beth Israel Deaconess Medical Center, Harvard Medical School, Boston, MA, USA. [32]Department of Internal Medicine, Seoul National University Bundang Hospital, Seongnam, Republic of Korea. [33]Department of Genetics, University of Alabama at Birmingham, Birmingham, AL, USA. [34]Department of Medicine and Program in Mendelian Genetics, Weill Cornell Medicine, New York, NY, USA. [35]Department of Medicine, Infectious Diseases Division, Howard University Hospital, Howard University College of Medicine, Washington, DC, USA. [36]Feil Family Brain and Mind Research Institute, Weill Cornell Medicine, New York, NY, USA. [37]Division of HIV, Infectious Diseases and Global Medicine, University of California, San Francisco, CA, USA. [38]Department of Genetics, Perelman School of Medicine, University of Pennsylvania, Philadelphia, PA, USA. [39]Department of Biostatistics, University of Alabama at Birmingham, Birmingham, AL, USA. [40]Laboratory of Genetic Susceptibility, Division of Cancer Epidemiology and Genetics, National Cancer Institute, Rockville, MD, USA. [41]Department of Biochemistry and Molecular Biology, National Human Genome Center, Howard University College of Medicine, Washington, DC, USA. [42]Veterans Affairs Medical Center, Washington, DC, USA. [43]Division of Cancer Epidemiology and Genetics, National Cancer Institute, Rockville, MD, USA. [44]Department of Medicine, Division of Allergy and Immunology, Division of Genetics, Brigham and Women's Hospital, Harvard Medical School, Boston, MA, USA. [45]Department of Infectious Diseases, Virology, University Hospital Heidelberg, Heidelberg, Germany. [46]These authors contributed equally: A. Rouf Banday, Megan L. Stanifer, Oscar Florez-Vargas. ✉e-mail: prokuninal@mail.nih.gov

## Methods

**Genetic analysis in COVNET.** Patients were recruited by studies participating in the Large-scale Genome-wide Association Study and Whole Genome Sequencing of COVID-19 Severity (COVNET, https://dceg.cancer.gov/research/how-we-study/genomic-studies/covnet). All institutions acquired ethical approvals based on informed consent provided by patients. COVID-19 diagnosis was confirmed based on positive viral testing or serology. Sample collection occurred pre-emergence of SARS-CoV-2 variants and vaccination. Detailed demographic and clinical records were provided by the participating studies and independently reviewed by the COVNET team. COVID-19 status was defined as nonhospitalized (mild), and hospitalized due to COVID-19.

DNA samples from patients were processed and analyzed as described in Extended Data Figure 1. Briefly, DNA samples were first analyzed by AmpFLSTR Identifiler (Thermo Fisher Scientific) for potential contamination and sex mismatch and then genotyped for 712,191 variants using the Global Screening Array version 2.0 (GSA2, Illumina) by the Cancer Genomics Research Laboratory, Division of Cancer Epidemiology and Genetics, National Cancer Institute (DCEG/NCI). Ancestry-specific genomic inflation factors (λ, Extended Data Fig. 2) were calculated using genome-wide genotyped data using PLINK version 1.9 (ref. [44]).

To evaluate imputation concordance, rs10774671 and rs1131454 were genotyped with TaqMan assays. In individuals of European ancestry, 65.5% of samples (1,520 of 2,249) were TaqMan-genotyped, with 95.4% concordance for rs10774671 and 91.6% concordance for rs1131454; in individuals of African ancestry, 99.6% of samples (832 of 835) were TaqMan-genotyped, with the concordance of 87.3% and 87.6%, respectively. The analyses for these markers were based on TaqMan genotype data supplemented by imputed genotypes.

Whole-genome sequencing data were available for 238 individuals of European ancestry; using whole-genome sequencing as a covariate did not affect the association results. LD plots were generated with Haploview version 4.2. A meta-analysis of results from patients of European and African ancestries was conducted using PLINK (v1.9). The LD-adjusted threshold method[5] was used to adjust for multiple testing; ancestry-specific LD blocks in COVNET samples were estimated based on the Solid LD spine method (Haploview version 4.2).

**Analysis of clinical trial.** We used data and samples from a phase 2 clinical trial (NCT04354259), in which patients with mild outpatient COVID-19 received a single subcutaneous injection of 180 mg pegIFN-λ1 ($n = 30$) or saline placebo ($n = 28$)[26]. The load of SARS-CoV-2 RNA (viral copies, log10) was measured at treatment days 0, 3, 5, 7, 10 and 14. Viral loss ($log_{10}$), calculated as the difference between viral copies at each posttreatment day and day 0, was used as the response variable. DNA was extracted from PBMCs of all participants and genotyped for three *OAS1* variants (rs1131454, rs10774671 and rs2660, all coded as 0, 1 or 2 based on the counts of risk alleles). These variants were selected to capture the main *OAS1* haplotypes associated with the risk of hospitalization for COVID-19 in COVNET (Fig. 1). Longitudinal trajectories of viral load in relation to genetic variants were explored using a linear mixed-effects model function from the R nlme package (v3.1–153) to build linear mixed models[45].

To explore whether associations between genetic variants and viral load varied by treatment, we built models to include genetic variants, treatment, viral load at day 0, sex and age as fixed effects, and patient IDs as a random effect (random intercept term). We used the maximum likelihood estimation procedure to conduct joint effects likelihood-ratio tests. Restricted maximum likelihood estimation was used for more precise estimates of the effect sizes.

A model for the mean longitudinal trajectory of the viral load that included interaction terms between the treatment arms and each genetic variant had a significantly better fit (ANOVA $P = 0.02$; likelihood ratio test degrees of freedom = 3) than a model with main effects only (Supplementary Table 9). This implied that the relationship between genetic variants and longitudinal viral load is different in the two treatment arms, justifying analyses of effects of genetic variants on viral load trajectory stratified by treatment group (Supplementary Table 10).

Haplotype analysis for viral load at baseline and after treatment was conducted using PLINK version 1.07 (ref. [44]). Omnibus haplotype association tests were calculated by haplotype replacement regression, controlling for sex, age, and viral load at day 0. The risk haplotype (rs1131454-A, rs10774671-A and rs2660-A) was used as a reference, and haplotypes with a combined frequency of less than 0.5% were excluded from the analysis. Analyses were based on additive genetic models, with haplotype-specific parameters representing the per-haplotype changes of viral load (log odds) compared to the reference haplotype.

**Genetic variants in archaic humans and chimpanzees.** Genetic variants of interest in three Neandertal individuals (Chagyrskaya, Altai and Vindija 33.19) were scored directly from BAM files retrieved from the Max Planck Institute for Evolutionary Anthropology website (http://cdna.eva.mpg.de/neandertal/). High-coverage sequence reads for Denisova genome and sequence alignments for 100 vertebrate species were accessed through the corresponding UCSC browser tracks (www.genome.ucsc.edu). Human variants within *OAS1* exons were analyzed in 29 chimpanzees (*Pan troglodytes*), including representatives of Central African subspecies, *P. t. troglodytes* ($n = 5$) and Western African subspecies, *P. t. verus* ($n = 24$). The analyses were based on previously generated sequences[46] available

from the European Nucleotide Archive (https://www.ebi.ac.uk (accession numbers FM163403.1–FM163432.1)).

**Cell lines.** Details for all cell lines used in this work are presented in Supplementary Table 11. Cell lines were either used within 6 months after purchase or periodically authenticated by microsatellite fingerprinting (AmpFLSTR Identifiler) by the Cancer Genomics Research Laboratory/DCEG/NCI. All cell lines were regularly tested for mycoplasma contamination using the MycoAlert Mycoplasma Detection kit (Lonza).

**TaqMan genotyping.** Genotyping of rs10774671, rs1131454 and rs2660 was done using TaqMan genotyping assays (Supplementary Table 12). Reactions (5 μl) were done with 2× TaqMan expression Master Mix (Qiagen) and 2–5 ng genomic DNA in 384-well plates on QuantStudio 7 (Thermo Fisher Scientific). Positive controls (HapMap samples with known genotypes) and negative controls (water) were included on each 384-well plate.

**Plasmids.** Plasmids with a Flag-tag for *OAS1-p46*-G (ID: OHu21619D, includes rs1131454-G, rs1131476-G, and rs1051042-G alleles) and *OAS1-p42*-A (ID: OHu29197D, includes rs1131454-A allele) were purchased from GenScript. The QuikChange II site-directed mutagenesis kit (Agilent) was used to generate plasmids *OAS1-p42*-G (rs1131454-G allele) and *OAS1-p46*-A (rs1131454-A, rs1131476-A and rs1051042-C alleles) using mutagenesis primers (Supplementary Table 12). Additionally, the Kozak sequence of Renilla (hRLuc) of psiCHECK-2 plasmid (Promega) was mutated to generate allele-specific plasmids for rs1859331-C and rs1859331-A within 5′ UTR of *OAS3*. The original and modified plasmids were confirmed by Sanger sequencing.

**Luciferase reporter assays with psiCHECK-2.** A549, HT1376 and T24 cells were seeded in 96-well plates ($2.0 \times 10^4$ cells per well). After 24 h, cells were transfected with allele-specific psiCHECK-2 plasmids for rs1859331 using Lipofectamine 3000 (Thermo Fisher Scientific). After 24 h, cells were lysed and assayed for Renilla and Firefly Luciferase (Promega) using GloMax Explorer (Promega).

**SARS-CoV-2 infections.** SARS-CoV-2 (strain BavPat1) was obtained from the European Virology Archive, amplified in Vero E6 cells, and used at passage 3. Media was removed from plated cells, and SARS-CoV-2 (MOI 3) was added to cells for 1 h at 37 °C; then, the virus was removed, cells were washed 1× with PBS, and fresh media was added back to the cells. RNA from harvested cells was extracted using RNeasy kit (Qiagen), cDNA was generated with iSCRIPT reverse transcriptase (Bio-Rad) from 250 ng of total RNA, and qRT-PCR was performed using SYBR Green assays (iTaq SYBR Green buffer, Bio-Rad) or TaqMan expression assays (Supplementary Table 12)[24,25].

A549-ACE2 cells were seeded in 12-well plates ($2.0 \times 10^5$ cells per well). After 24 h, cells were transfected with the indicated plasmids (*GFP* or *OAS1*) using Lipofectamine 2000. Media was replaced 6 h after transfection, and cells were infected with SARS-CoV-2 at an MOI = 3 for 1 h at 48 h after transfection. SARS-CoV-2 expression was evaluated in cells harvested 24 h after infection.

Caco2 cells were seeded in 48-well plates ($7.5 \times 10^4$ cells per well), then media was removed after 20 h, and interferons were added to the wells for 4 h. Media with interferons was collected and added back after infection with SARS-CoV-2 for 1 h. Interferon treatment: 2,000 IU ml$^{-1}$ IFN-β or 300 ng ml$^{-1}$ (a cocktail of 100 ng each of IFN-λ1, IFN-λ2 and IFN-λ3).

**Western blotting.** Cells (Caco2, HT1376, A549, and HBEC) were seeded in 6-well plates ($5 \times 10^5$ cells per well) and were untreated or treated with IFN-β (1 ng ml$^{-1}$), IFN-γ (2 ng ml$^{-1}$) or IFN-λ3 (100 ng ml$^{-1}$) for 24 h. Cells were lysed with RIPA buffer (Sigma-Aldrich) supplemented with protease inhibitor cocktail (Promega) and PhosSTOP (Roche) and placed on ice for 30 min, with vortexing every 10 min. Lysates were pulse-sonicated for 30 s, with 10-s burst-cooling cycles, at 4 °C, boiled in reducing sample buffer for 5 min and resolved on 4–12% Bis-Tris Bolt gels and transferred using an iBlot 2 (Thermo Fisher Scientific). Blots were blocked in 2.5% milk in 1% TBS-Tween before staining with rabbit anti-OAS1 antibody (1:200 dilution, Thermo Fisher Scientific, PA5-82113) and rabbit anti-GAPDH antibody (1:500 dilution, Abcam, ab9485). Signals were detected with HyGLO Quick Spray (Denville Scientific) or SuperSignal West Femto Maximum Sensitivity Substrate (Thermo Fisher Scientific) and viewed on a ChemiDoc Touch Imager with Image Lab 5.2 software (Bio-Rad).

For detection of OAS1-Flag protein isoforms in A549-ACE2 cells transfected with corresponding OAS1-Flag plasmids and infected with SARS-CoV-2, cells were rinsed with PBS and then lysed with 1× RIPA buffer supplemented with phosphatase and protease inhibitors (Sigma-Aldrich or Thermo Fisher Scientific) for 5 min. Samples were then collected and boiled at 95 °C for 5 mins. About 5 μg protein lysates was separated by 12% SDS-PAGE and then transferred onto a nitrocellulose membrane by wet blotting. Membranes were blocked with 5% non-fat milk in TBS-Tween for 1 h at room temperature with shaking. All antibodies were diluted in 5% BSA in TBS-Tween. Membranes were incubated with primary antibodies at 4 °C with shaking overnight, washed three times in TBS-Tween for 5 min at room temperature, incubated with secondary antibodies

for 1 h at room temperature with shaking and washed three times in TBS-Tween for 5 min at room temperature again. Horseradish peroxidase detection reagent was mixed 1:1 and incubated at room temperature for 5 min, and membranes were then visualized by chemiluminescence using the G:BOX Chemi gel doc Imaging System Instrument. Antibodies: rabbit anti-Flag (1:1,000 dilution, Sigma-Aldrich, F7425-2MG); anti-GAPDH (Cell Signaling Technology, 97166, mouse, 1:1,000 dilution or #ab9485, Abcam, rabbit, 1:500 dilution); secondary anti-rabbit (1:10,000 dilution, Abcam, ab97051), secondary anti-mouse (1:10,000 dilution, Abcam, ab6789) and ECL substrate (Thermo Fisher Scientific).

**Live-cell imaging analysis of cell growth.** A549 cells were seeded in 12-well plates at a density of $3.5 \times 10^4$ cells per well. After 24 h, cells were transfected in triplicate with plasmids (*GFP* or *OAS1*) using Lipofectamine 3000 transfection reagent (Thermo Fisher Scientific). Live-cell imaging was performed using the Lionheart FX Automated Microscope (BioTek) equipped with full temperature and $CO_2$ control to maintain 37 °C and 5% $CO_2$. Images were collected using a ×4 magnification right after transfection and then at 24, 48, 72 and 96 h after transfection. Data were processed with Gen5 Image+ software (BioTek) to determine cell counts and are presented normalized to cells transfected with GFP.

**Confocal microscopy.** HT1376 bladder cancer cell line (rs10774671-GG genotype, OAS1-p46 isoform) and A549 lung cancer cell line (rs10774671-AA genotype, OAS1-p42 isoform) were plated in 4-well chambered slides ($2 \times 10^4$ cells per well, LabTek) for 24 h. Cells were left untreated or treated with 2 ng ml$^{-1}$ IFN-β (R&D Systems) for 24 h. Cells were then washed twice with PBS and fixed with 4% paraformaldehyde (BD Biosciences) for 30 min. After rinsing twice in PBS and permeabilization buffer (BD Biosciences), cells were incubated with permeabilization buffer for 1 h. Fixed cells were incubated with mouse anti-Golgin-97 antibody (1:250 dilution, Thermo Fisher Scientific, A-21270) for 3 h at room temperature, washed and then stained with anti-rabbit Alexa Fluor 488 (1:500 dilution, Thermo Fisher Scientific, A21202). Cells were then incubated with rabbit anti-OAS1 antibody (1:100 dilution, Thermo Fisher Scientific, PA5-82113) overnight, washed and stained with anti-rabbit Alexa Fluor 680 (1:500 dilution, Thermo Fisher Scientific, A10043). Slides were mounted with antifade mounting media with 4,6-diamidino-2-phenylindole (Thermo Fisher Scientific) and imaged at ×63 magnification on an LSM700 confocal laser scanning microscope (Carl Zeiss) using an inverted oil lens. Colocalization and correlation coefficients between OAS1 and Golgin-97 expression were generated with LSM700 Zen software by analyzing randomly imaged fields of view (five to seven fields) containing at least seven cells from IFN-β-treated wells. The linear relationship between the expression of OAS1 and Golgin-97 at every pixel with protein expression was determined with Pearson's correlation coefficient[47]. Cells with less than 10 analyzed pixels were excluded due to very low expression of either protein, making the colocalization data unreliable. Mander's overlap coefficients were also calculated[47], which factor in the total number of pixels of either protein.

RNA-seq analysis of data from the National Center for Biotechnology Information (NCBI) Sequence Read Archive (SRA) and TCGA. RNA-seq datasets were accessed in the NCBI SRA with SRA command-line tools. SRA datasets analyzed in this study are listed in Supplementary Table 13. Briefly, the raw FASTQ files were compressed using GZIP (version 1.10) and aligned with STAR version 2.7.6a to the reference human genome assembly (hg38). Low-quality sequencing files with ≤80% of mappable reads were excluded from further analyses. BAM slices were indexed and sliced to include 117 kb of the *OAS1–OAS3* genomic region (chr12:112,901,893–113,019,729, hg38). For TCGA, BAM slices for the *OAS* locus were generated through the NCI Genomics Data Commons portal accessed on 25 November 2020 using standard workflow (https://docs.gdc.cancer.gov/API/Users_Guide/BAM_Slicing/).

**Estimation of RNA-seq read counts specific to *OAS1* isoforms.** Expression of *OAS1* isoforms *p42*, *p44*, *p46* and *p48* was quantified based on unique RNA-seq reads. Specifically, RNA-seq BAM slices were processed using the R package ASpli version 1.5.1 with default settings. Specific exon and exon–exon junction reads were quantified and exported in a tab file format. For *OAS1* isoforms *p44*, *p46* and *p48*, RNA-seq reads specific to their unique last exon–exon junctions were used for quantification. For the *p42* isoform, which does not have a unique exon–exon junction, sequencing reads corresponding to its unique 3′ UTR (extension of exon 5) were used as a proxy for quantification. For normalizing expression, junction reads were divided by 50 (average length of an RNA-seq read), and *p42* 3′ UTR exon reads were divided by 317 bp, corresponding to its length. The mean expression of each isoform was calculated from samples with three or more RNA-seq reads supporting the unique splice junction or exon.

**Analysis of ATAC-seq, ChIP-seq and Hi-C data in cell lines.** Raw data for ATAC-seq, H3K27ac ChIP-seq, Hi-C and RNA-seq for SW780, HT1376 and SCABER bladder cancer cell lines were downloaded from NCBI SRA (ID: PRJNA623018) using the SRA tools. For ATAC-seq and H3K27ac ChIP-seq analysis, the FASTQ files were aligned to hg19 using ENCODE-DCC ATAC-seq-pipeline version 1.9.1 (https://github.com/ENCODE-DCC/atac-seq-pipeline) and ChIP-seq-pipeline2 version 1.6.1 (https://github.com/ENCODE-DCC/chip-seq-pipeline2) with default settings. The output bigwig files were then uploaded to the UCSC genome browser for visualization. For RNA-seq analysis, the FASTQ files were mapped to hg19 using STAR version 2.7.6a aligner (https://github.com/alexdobin/STAR) with default settings. The output sorted BAM files were indexed using SAM tools (https://github.com/samtools/). For Hi-C, FASTQ files were processed using Juicer version 1.6 (https://github.com/aidenlab/juicer) by selecting relevant restriction cutting sites such as MboI/DpnII and aligned to hg19. The chromatin loops in Hi-C data were detected using Hiccups in Juicer version 1.6 with default settings. The same procedure was applied to analyze Hi-C data for THP-1 monocytic cell line untreated or treated with INF-β for 6 h. The Hi-C and chromatin interactions were visualized in the UCSC genome browser (https://genome.ucsc.edu). Integrative data analysis was performed to identify open chromatin marks and chromatin interactions between associated genetic variants co-localizing with enhancers and promoters of *OAS1*, *OAS2* and *OAS3*.

**Allele-specific analyses in RNA-seq datasets.** RNA-seq BAM slices were genotyped for *OAS1* exonic variants with an Integrative Genome Viewer (version 2.8.9) command-line tool using a 21-bp sequence centered on each variant. A 10% threshold of allele-specific reads was used for genotype calling of each variant.

**Analysis of exonic splicing enhancer activity for rs1131454.** The allele-specific sequence (5′-GUCAGUUGACUGGC[A/G]GCUAUAAACUA-3′) centered on rs1131454 was used for the prediction of exonic splicing enhancer (ESE)/silencer (ESS) motifs using the Human Splicing Finder (www.umd.be/HSF3/). The binding sites for alternative splicing factors were depicted with a bar graph. Exontrap mini-genes were generated for alleles of rs1131454. Specifically, allele-specific sequences of exon 3 with 100 bp of flanking intronic sequences and overhangs for restriction sites (XhoI and NotI) were custom-synthesized as gene fragments (IDT, Supplementary Table 14). These fragments were cloned in sense orientation in Exontrap vector pET01 (MoBiTec) using XhoI and NotI restriction sites and validated by Sanger sequencing. The A549 and T24 cells were seeded in a 12-well plate at a cell density of $2 \times 10^5$ and transfected after 24 h with 200 ng allele-specific mini-genes using Lipofectamine 3000 transfection reagent (Invitrogen) in three biological replicates. At 48 h after transfection, cells were harvested, and total RNA was extracted with QiaCube using RNeasy kit with on-column DNase I treatment (Qiagen). cDNA was prepared for each sample with 500 ng total RNA using SuperScript III reverse transcriptase (Invitrogen) and a vector-specific primer (5′-AGGGGTGGACAGGGTAGTG-3′). cDNA corresponding to 5 ng RNA input was used for each RT-PCR reaction. Two common primer pairs were used for characterizing splicing products of allele-specific mini-genes (Supplementary Table 14). Only primer pair 1 (FP vector exon 1: 5′-GGA GGA CCC ACA AGG TCA GTT-3′; and RP exon 3: 5′-GCTG CTT CAG GAA GTC TCT CTG-3′) identified alternative splicing events corresponding to endogenous exon 3 splicing between vector exon 1 and insert after PCR-amplified products were resolved by agarose gel electrophoresis. The specific bands were cut out from the gel, purified and validated by Sanger sequencing. The ratio of alternative splicing products was calculated based on band intensity using densitometry, and fold changes were calculated between two allele-specific mini-genes.

**RNA-seq analysis with Oxford Nanopore.** A549 or HT1376 cells ($2 \times 10^6$ per sample) were seeded in T25 flasks overnight. The next day, media was replaced with either media containing 2 ng ml$^{-1}$ IFN-β (treatment) or normal media without IFN-β (mock). Total RNA was prepared from cells 24 h after treatment using the RNeasy Mini Kit (Qiagen). Poly(A)$^+$ RNA was enriched from total RNA using the Dynabeads mRNA Purification Kit (Invitrogen). cDNA libraries were prepared from 200 ng poly(A)$^+$ RNA using the Direct cDNA Sequencing Kit (Oxford Nanopore), according to the PCR-free 1D read protocol for full-length cDNA (Oxford Nanopore, SQK-DCS109), with some modifications. Specifically, RNase Cocktail Enzyme Mix (Thermo Fisher Scientific) was used during the RNA digestion step after the first-strand synthesis; all reaction amounts for reverse transcription reactions up to the second-strand synthesis step were doubled; from second-strand synthesis up to adapter ligation, reactions were 1.5× of original amounts; during adapter ligation, 35 μl Blunt/TA Ligase Master Mix was used instead of the recommended 50 μl, and nuclease-free water was excluded. Final libraries were loaded into MinION Fluidics Module flow cells (Oxford Nanopore, FLO-MIN106D), and sequencing was carried out on GridION MK1 and MinION MK1C instruments (Oxford Nanopore) for 3 days, using default parameters.

The FASTQ files generated by Nanopore GridION long-read sequencer were trimmed using Porechop version 0.2.4 (https://github.com/rrwick/Porechop) and aligned to the hg19 genome using Minimap2 version 2.18 (https://github.com/lh3/minimap2) with -ax splice command. The output SAM files were then converted to indexed, sorted BAM files using SAM tools version 1.11 (https://github.com/samtools/). The BAM files were visualized with the UCSC genome browser.

**Analysis of NMD of *OAS1* isoforms.** RNA-seq data for HeLa cells (OAS1-p42 expressing) with and without siRNA-mediated KD of NMD genes *SMG6* and *SMG7* were downloaded from SRA (PRJNA340370). The FASTQ files were aligned with STAR aligner (https://github.com/alexdobin/STAR) with default settings

followed by quantification of isoforms-specific reads for alternative splicing junctions of *OAS1* exon 3 with adjacent exons. The data were also visualized as RNA-seq plots using the Integrative Genome Viewer. We also generated a triple KD by transfecting A549 (OAS1-p42 expressing) and HT1376 (OAS1-p46 expressing) cells with siRNAs scrambled (negative control) and targeting genes for the NMD pathway (*SMG6*, *SMG7*, and *UPF1*). After 48 h, cells were harvested, and total RNA was isolated using an RNeasy kit with on-column DNase I treatment (Qiagen). Subsequently, cDNA for each sample was prepared from equal amounts of RNA using the RT² First-Strand cDNA kit (Qiagen). TaqMan assays were used to confirm the KD of each gene using the expression of *HPRT1* as an endogenous control (Supplementary Table 12). *OAS1* exon 3 splicing events were detected with custom expression assays (Supplementary Table 12). Experiments were performed in biological triplicates for each condition, and expression was quantified in four technical replicates on QuantStudio 7 (Life Technologies) using TaqMan Gene Expression buffer (Thermo Fisher Scientific). Genomic DNA and water were used as negative controls for all assays. Expression was measured as $C_t$ values (PCR cycle at detection threshold) and calculated as $\Delta C_t$ values normalized by endogenous control and $\Delta\Delta C_t$ values normalized by a reference group of samples.

**Tools for statistical analyses and graphics.** We utilized the NIH Biowulf supercomputing cluster (http://hpc.nih.gov) and specific packages in R versions 3.6.0 to 4.0.4 for data processing and statistical analyses. Data were plotted using ggplot2 version 3.3.3 in R or GraphPad Prism version 8.

**Reporting summary.** Further information on research design is available in the Nature Research Reporting Summary linked to this article.

## Data availability
Summary statistics for primary and conditional association analyses for 79 genetic variants within the *OAS1*–*OAS3* region in individuals of European and African ancestries are provided in Supplementary Tables 1 and 2. The dataset for Oxford Nanopore RNA-seq was deposited to NCI as SRA: PRJNA743928. Full-length sequence data for *OAS1-p42* transcript with short exon 3 were deposited to NCBI GenBank with accession number MZ491787. Requests for any additional data or reagents should be addressed to L.P.-O. (prokuninal@mail.nih.gov). Source data are provided with this paper.

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

## Acknowledgements
We are grateful to all the patients who donated their samples. We are also indebted to all the doctors and nurses who contributed to this study. Additionally, we thank L. McReynolds (CGB/DCEG/NCI) for clinical review of the samples; M. Machiela (ITEB/ DCEG/NCI) for contributing to the data analysis plan, M. Yan (LTG/DCEG/NCI) for help with analyses, N. Cole (CGR/DCEG/NCI) for help with the acquisition of TCGA sliced BAM files, M. Dean (LTG/DCEG/NCI) for help with Oxford Nanopore sequencing; G. Vatsellas and A.G. Korovesi (Biomedical Research Foundation of the Academy of Athens) for technical assistance on DNA extraction and preparation; J.C. Sapp (NHGRI) for support and guidance in patient recruitment; M. Martin (BSP, Frederick National Laboratory for Cancer Research) for coordinating sample and data transfers; J. Li (Harvard Medical School) for cohort organization; L. Jin, J. Wu and R. Shevin in the UAB Center for Clinical and Translational Science for preparation of genomic DNA; and B. Antônio Lopes da Fonseca and F. Crivelenti Vilar (University of São Paulo, Brazil) for recruiting patients. The results are partially based on data generated by the TCGA Research Network. We used data provided by the Kaiser Permanente Research Bank (KPRB) from the KPRB collection, which includes the Kaiser Permanente Research Program on Genes, Environment, and Health (RPGEH), funded by the National Institutes of Health (NIH; RC2 AG036607), the Robert Wood Johnson Foundation, the Wayne and Gladys Valley Foundation, The Ellison Medical Foundation and the Kaiser Permanente Community Benefits Program. Access to data used in this study may be obtained by application to the KPRB at kp.org/researchbank/ researchers. Research reported in this publication was supported by the Intramural Research Programs of the National Cancer Institute, Division of Cancer Epidemiology and Genetics and Center for Cancer Research, Frederick National Laboratory for Cancer Research, contract number HHSN261200800001E; National Center for Advancing Translational Science of the NIH under award number UL1TR003096 associated with the University of Alabama at Birmingham COVID-19 Enterprise Study IRB-20005127, NHGRI Intramural Research Program grants HG200388-07 and HG200359-12. Additional funding was provided by the state of Alabama through the Alabama Genomic Health Initiative (IRB F170303004). S.B. was supported by grants from Deutsche Forschungsgemeinschaft project numbers 415089553 (Heisenberg program), 240245660 (SFB1129) and 272983813 (TRR179) and from the state of Baden Wuerttemberg (AZ: 33.7533.-6-21/5/1) and the Bundesministerium Bildung und Forschung (01KI20198A). M.L.S. was supported by grants from the Bundesministerium Bildung und Forschung (01KI20239B) and Deutsche Forschungsgemeinschaft project 416072091. E.A. was supported by grants from the European Commission (IMMUNAID, 779295 and CURE, 767015) and the Hellenic Foundation for Research and Innovation INTERFLU (1574). H.B.K. was supported by grant no 02-2020-012 from the SNUBH Research Fund. Additional support was provided by the Emory Department of Gynecology and Obstetrics EmPOWR Initiative (IRB00101931) and a grant from the Emory Medical Care Foundation Grant (IRB00000312). M.E.R. and S.M.L. were supported by NIH grant NS105477 and the NYPH-Weill Cornell Medicine COVID-19 Biobank Fund (IRB 20-04021808). D.R.W. was supported by the Massachusetts Consortium on Pathogen Readiness and NIH grants AI165072 and AI139538. The pegIFN-λ1 clinical trial was supported by the Toronto COVID-19 Action Initiative (72059280), the Ontario Together COVID-19 Research Application (C-224-2428560-FELD) and the Canadian Institutes for Health Research (VR3-172648). Support for title page creation and format was provided by AuthorArranger (https://authorarranger.nci.nih.gov/), a tool developed at the NCI. The content of this publication is solely the responsibility of the authors and does not necessarily represent the official views and policies of the NIH or Department of Health and Human Services, nor does mention of trade names, commercial products or organizations imply endorsement by the US Government.

## Author contributions
A.R.B. and L.P.-O. conceived and designed the study. L.P.-O. supervised the study. A.R.B., M.L.S., O.O.O., S.B., J.J.F. and L.P.-O. designed the experiments and analyses. A.R.B., M.L.S., O.O.O., M.A.Z., B.W.P., T.J.R., C.G., M.H. and J.M.V. performed the experiments and analyzed the data. O.F.-V. analyzed genetic data. H.P.S.A. and M.Y. contributed to the analysis of genetic data. A.R.B. and C.-H.L. performed computational analyses of genomic data. O.F.-V. and P.S.A. analyzed clinical trial data. S.J.C. supervised the COVNET study. A.A.H. supervised COVNET sample processing and genotyping. L.M., S.S., and V.V. contributed to COVNET sample and data management. A.R.B., M.L.S., O.F.-V., O.O.O., M.A.Z., E. Andreakos, E. Arons, G.B., L.G.B., D.L.B., M.S.B., A.B.-H., M.C., E.C., P.G.C., R.L.C., L.M.C., C.M.D., C.L.D., J.E., N.E., H.S.F., B.A.F., G.S.F., A.J.G., S.H., A.A.H., H.I., L.I., M.G.I., N.T.J., H.B.K., R.J.K., B.R.K., S.M.L., S.M.M., I.M., G.L.P., J.A.P., M.J.P., D.J.R., D.T.R., M.D.R., B.R., M.E.R., S.A.S., S.S., E.S., A.K.S., V.T., J.D.V., A.V., D.R.W., M.Y., X.Y., Y.Z., S.B., S.J.C., J.J.F. and L.P.-O. contributed reagents/materials/ analysis tools. A.R.B. and L.P.-O. wrote the manuscript. All authors discussed the results and commented on the manuscript.

## Competing interests
J.J.F. reports receiving research support unrelated to this work from Eiger BioPharmaceuticals. All other authors declare no competing interests.

## Additional information
**Extended data** is available for this paper at https://doi.org/10.1038/s41588-022-01113-z.

**Correspondence and requests for materials** should be addressed to Ludmila Prokunina-Olsson.

## Analyses in COVNET

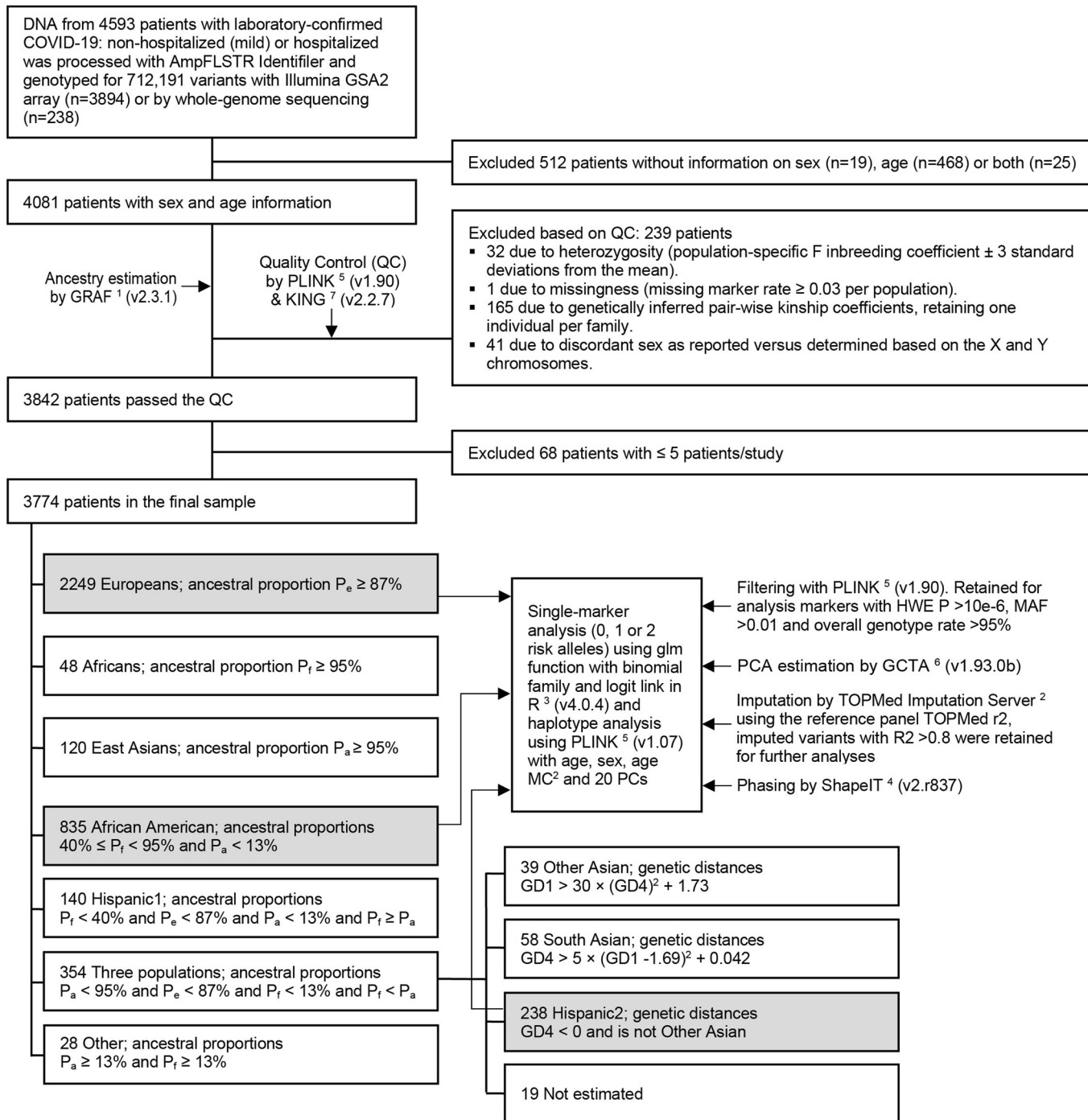

**Extended Data Fig. 1 | Outline of genetic analyses in COVNET.** COVNET: Large-scale Genome-wide Association Study and Whole Genome Sequencing of COVID-19 Severity (https://dceg.cancer.gov/research/how-we-study/genomic-studies/covnet). Analyses were done in sets shaded in gray, which have more than 100 patients per ancestry and outcome group (hospitalized versus nonhospitalized); this included patients of European (n = 2,249: 1,214 versus 1,035) and African (n = 835: 511 versus 324) ancestries. For rs10774671 and rs1131454, genotyped data based on TaqMan assays were used for all non-European patients and a subset of European patients (see Methods).

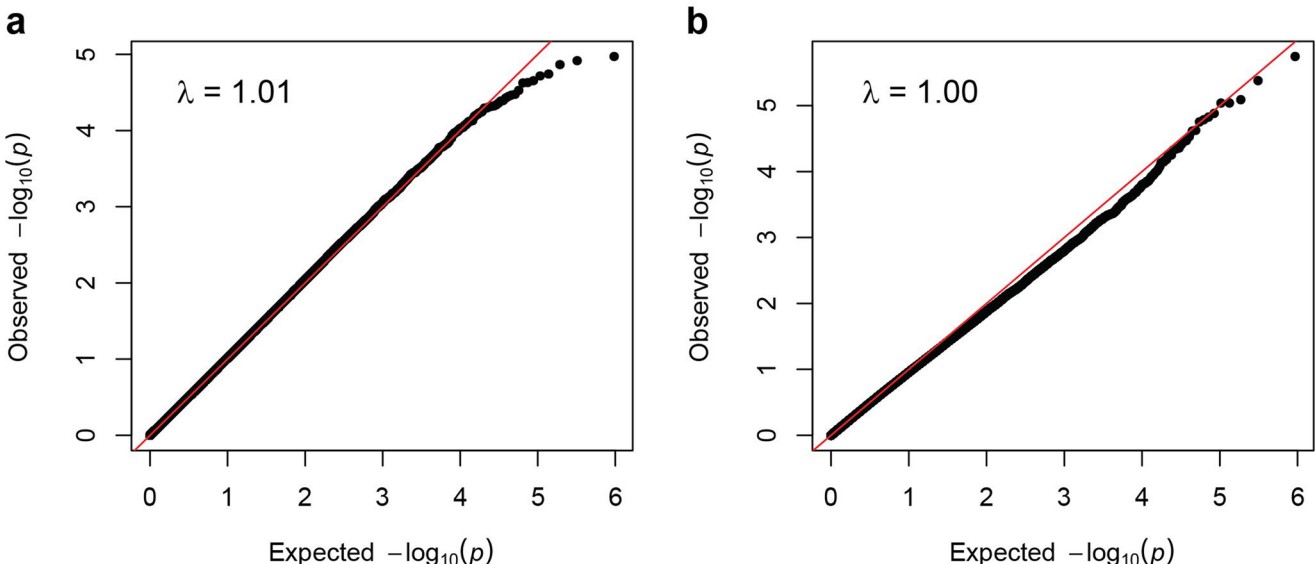

**Extended Data Fig. 2 | Quantile-Quantile plots and genomic inflation factors (λ) for COVNET GWAS analyses of COVID-19 severity. a,b**, The analyses are based on 511,229 genome-wide array-genotyped markers in hospitalized versus nonhospitalized COVID-19 patients of European ancestry ($n=2,249$: 1,214 versus 1,035) (**a**) and African ancestry ($n=835$: 511 versus 324) (**b**). $P$-values are for two-sided Fisher's exact test adjusting for age, sex, squared mean-centered age, and 20 PCs.

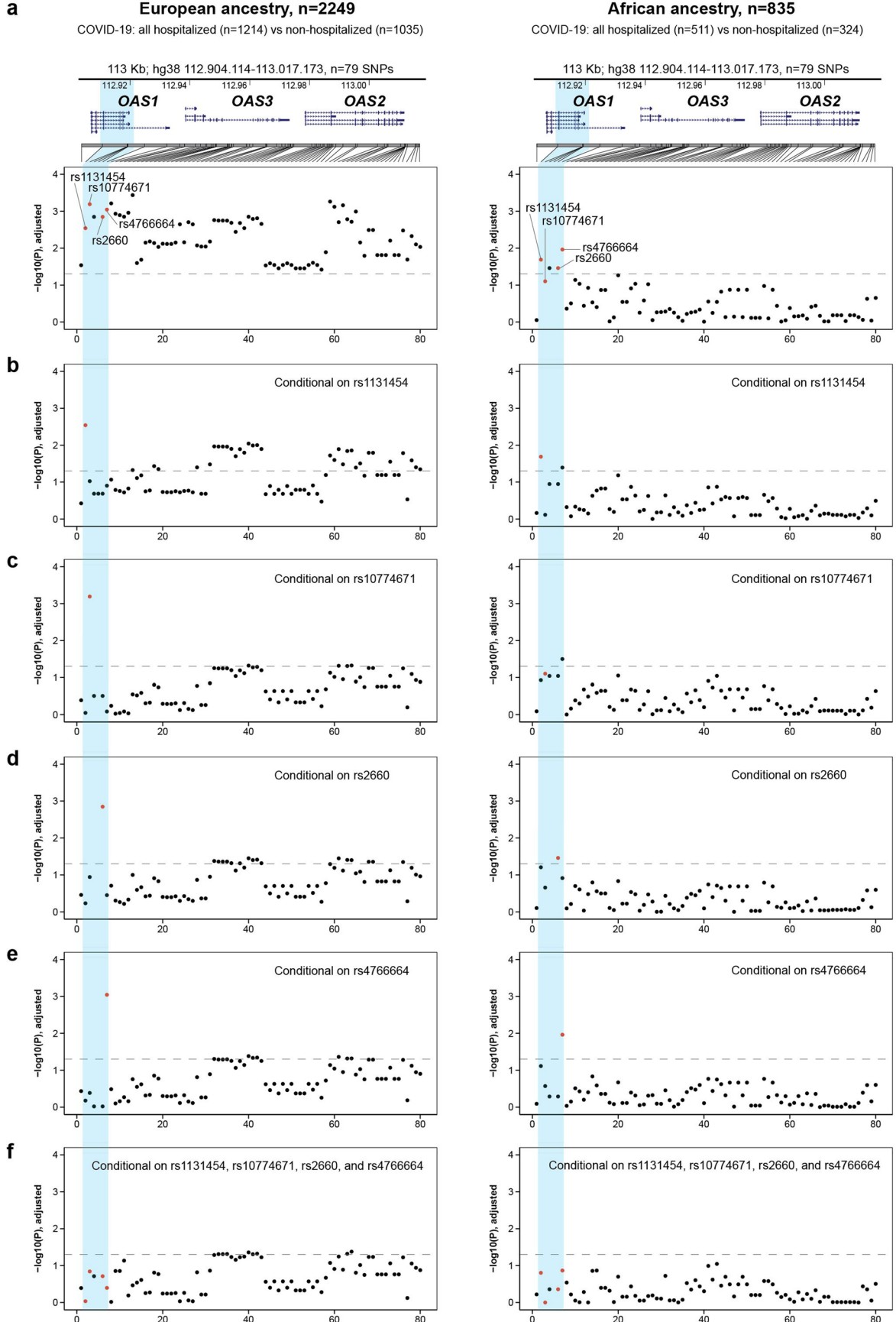

**Extended Data Fig. 3 | See next page for caption.**

**Extended Data Fig. 3 | Conditional association analyses for COVID-19 severity within the chr12q24.13 region in COVNET patients of European and African ancestries. a**, Genomic region and association results (*P*-values) for 79 genotyped or confidently imputed markers for hospitalized compared to nonhospitalized COVID-19 in patients of European and African ancestries. Presented results are for logistic regression analyses conditioning on markers indicated by red dots. **b**, rs1131454. **c**, rs10774671. **d**, rs2660. **e**, rs4766664. **f**, rs1131454, rs10774671, rs2660, and rs4766664 combined. The horizontal dotted line represents the *P* = 0.05 significance threshold. All association analyses were performed using logistic regression models, adjusting for sex, age, squared mean-centered age, and 20 PCs. Full conditional association results for individual variants are provided in Supplementary Table 2.

**a**  COVID-19 patients of European ancestry: all hospitalized (n=1214) and non-hospitalized (n=1035)

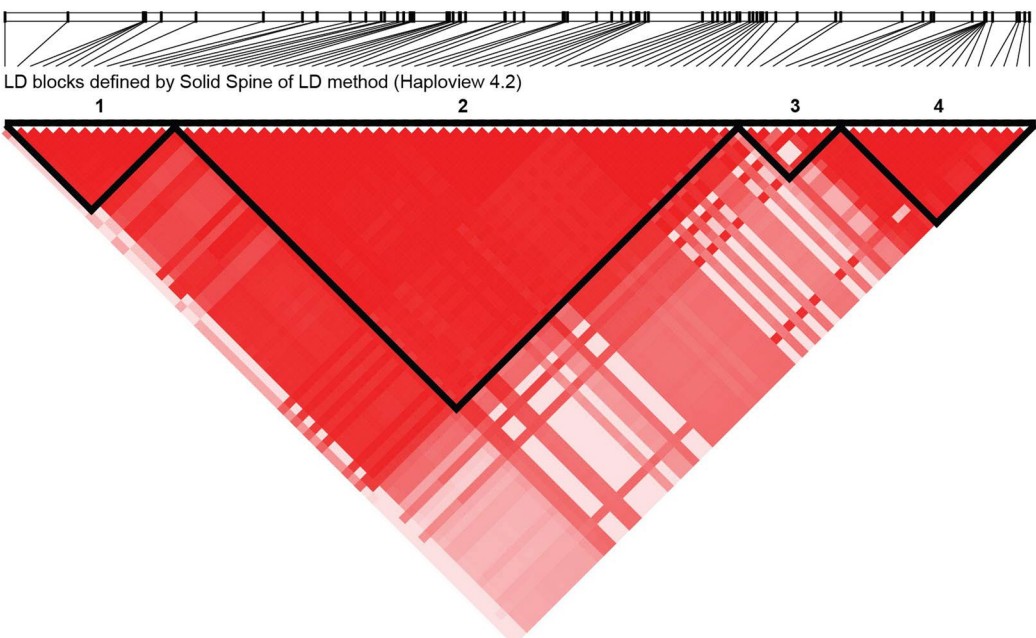

**b**  COVID-19 patients of African ancestry: all hospitalized (n=511) and non-hospitalized (n=324)

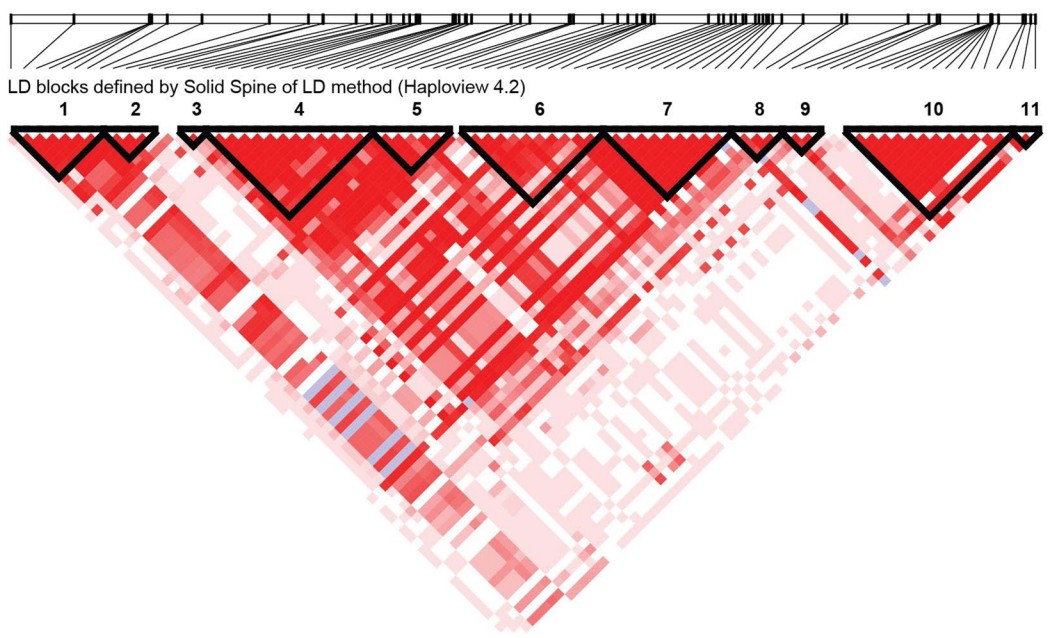

**Extended Data Fig. 4 | Linkage disequilibrium blocks within the chr12q24.13 region in COVNET patients of European and African ancestries.**
**a,b,** Analysis of the 113-kb region (hg38:112,904,114-113,017,173, $n = 79$ SNPs) at chr12q24.13 with Solid Spine method (Haploview version 4.2) identified 4 linkage disequilibrium (LD, D') blocks in COVNET COVID-19 patients of European ancestry (**a**) and 11 LD blocks in patients of African ancestry (**b**). Dark red shading denotes $D' > 0.80$.

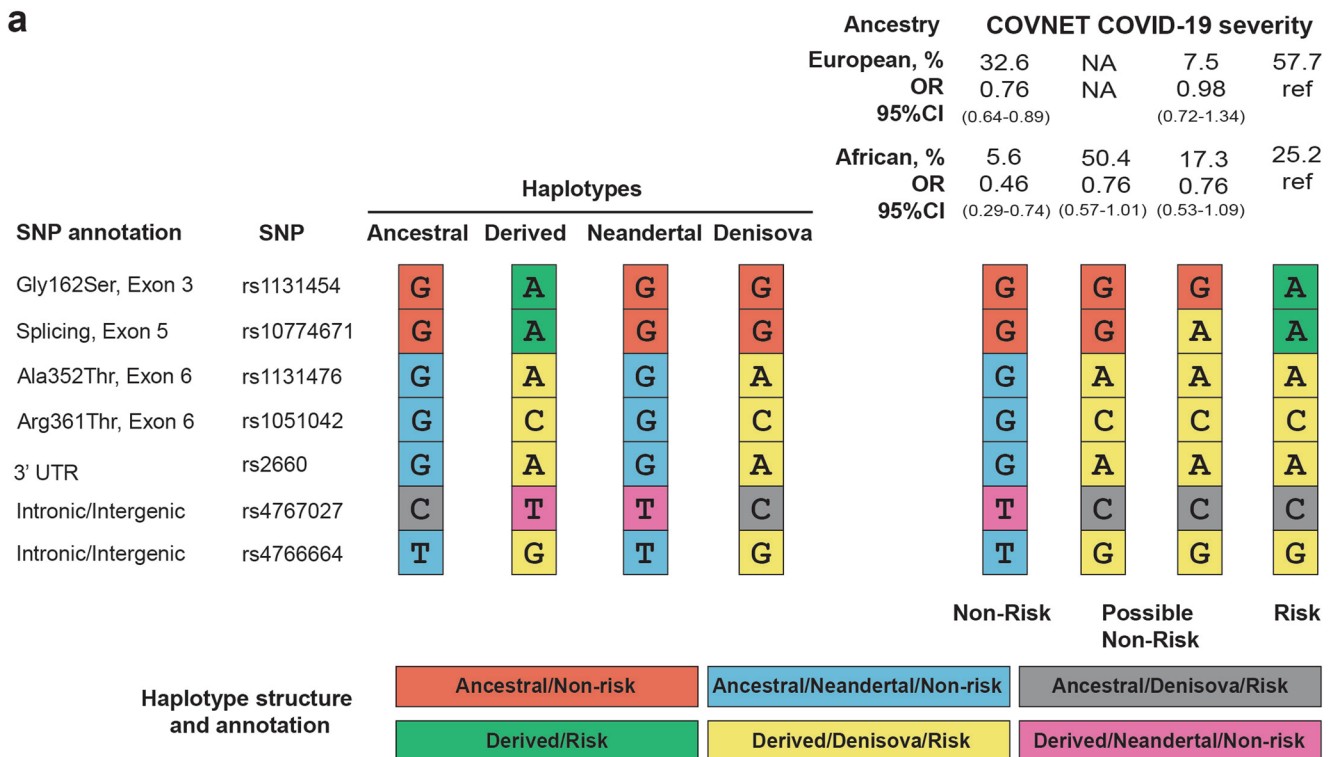

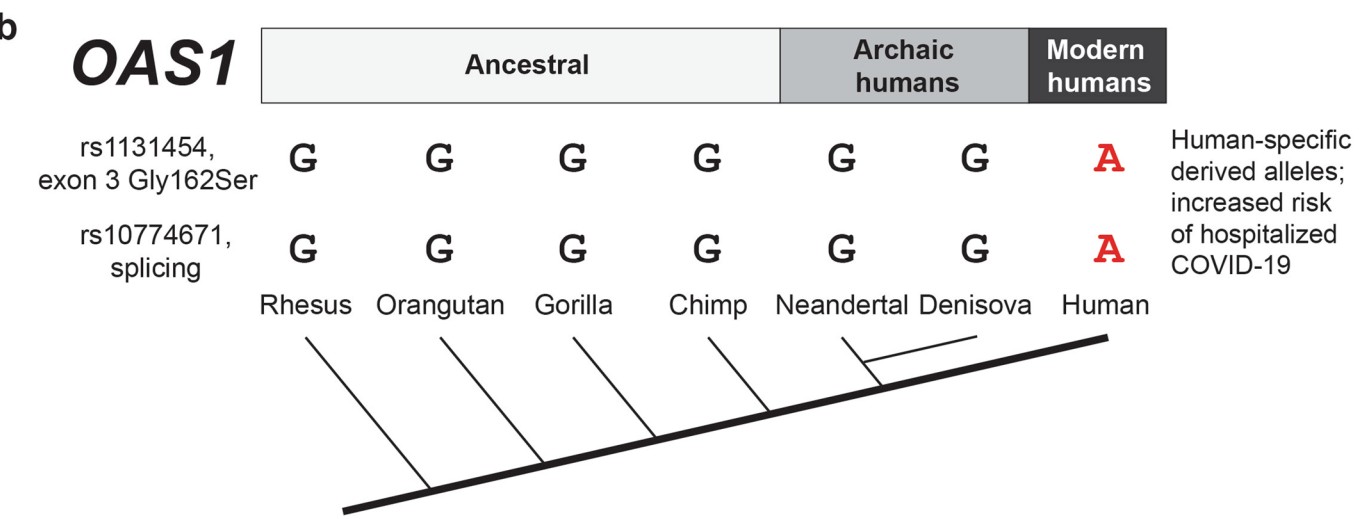

**Extended Data Fig. 5 | Structure of *OAS1* haplotypes in relation to ancestral status and association with COVID-19 severity. a**, Analysis of the *OAS1* haplotypes (14.13 kb, hg38:112,911,065-112,925,192) comprised of 7 markers. The color-coding indicates the ancestral status of specific alleles – human (ancestral or derived), archaic humans (Neandertal or Denisova lineages), and COVID-19 severity status (Non-risk/Risk). Haplotype frequencies are shown for hospitalized patients with COVID-19 of European and African ancestry from COVNET. NA, haplotype is not detected. Odds ratios (ORs) and 95% confidence intervals (95%CIs) are for comparison with the common Risk haplotype (also marked as ref); full results can be found in Supplementary Table 4. The Non-risk haplotypes differ from the Risk haplotype by the alleles of rs1131454 and rs10774671. The COVID-19 risk is associated with human-specific derived alleles rs1131454-A and rs10774671-A. Additionally, the Risk haplotype includes a Denisova-type fragment of derived alleles spanning rs1131476 to rs4766664. Human polymorphisms rs1131454, rs1131476, rs1051042, and rs2660 were also explored and found monomorphic in genomic sequences of 29 chimpanzees: *Pan troglodytes verus* (*n* = 24) and *Pan troglodytes troglodytes* (*n* = 5). Source[8]: European Nucleotide Archive https://www.ebi.ac.uk, accession numbers FM163403.1–FM163432.1. **b**, *OAS1* haplotypes and phylogenetic tree. COVID-19 risk alleles rs1131454-A and rs10774671-A are human-specific and derived.

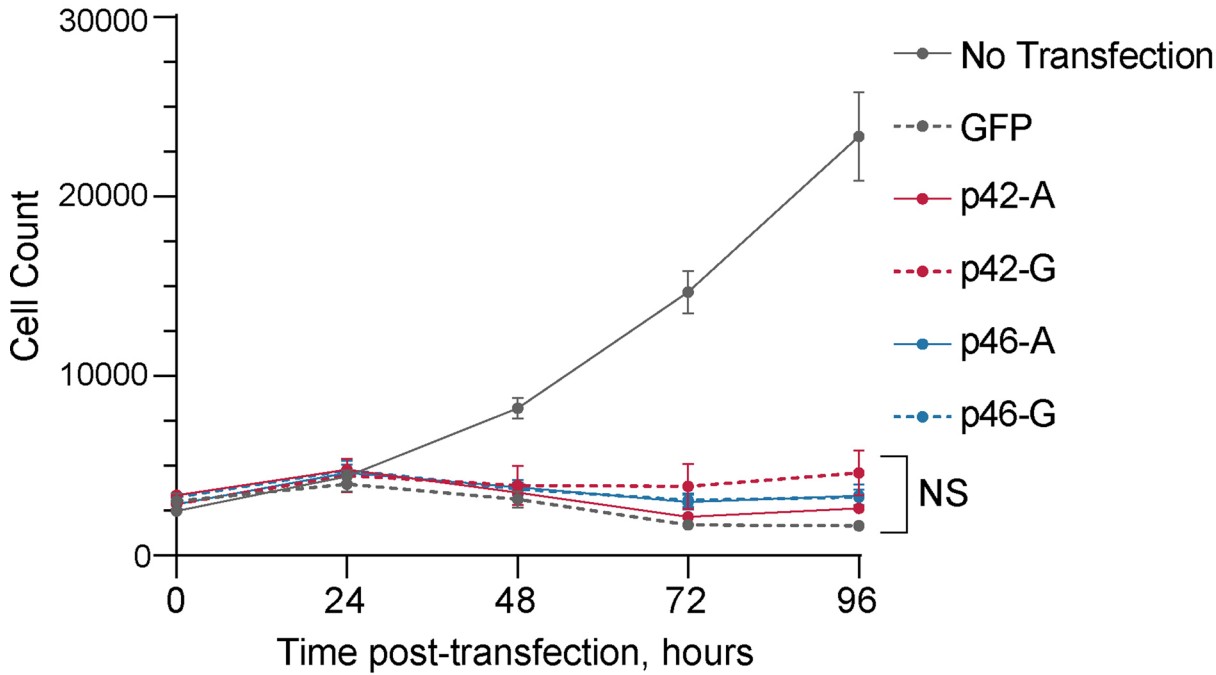

**Extended Data Fig. 6 | Allele-specific *OAS1* plasmids show similar effects on cell growth.** A549 cells were untransfected or transiently transfected in four biological replicates with plasmids: GFP, OAS1-p42-A, OAS1-p42-G, OAS1-p46-A, or OAS1-p46-G (see Fig. 2a for plasmid details). Cells were counted by automated live-cell imaging using Lionheart microscopy right after transfection (0 h) and then every 24 h for 96 h. The plots are presented with means and s.d. *P*-value is for comparison between all plasmids using one-way ANOVA with Tukey's multiple comparison test. NS, not significant.

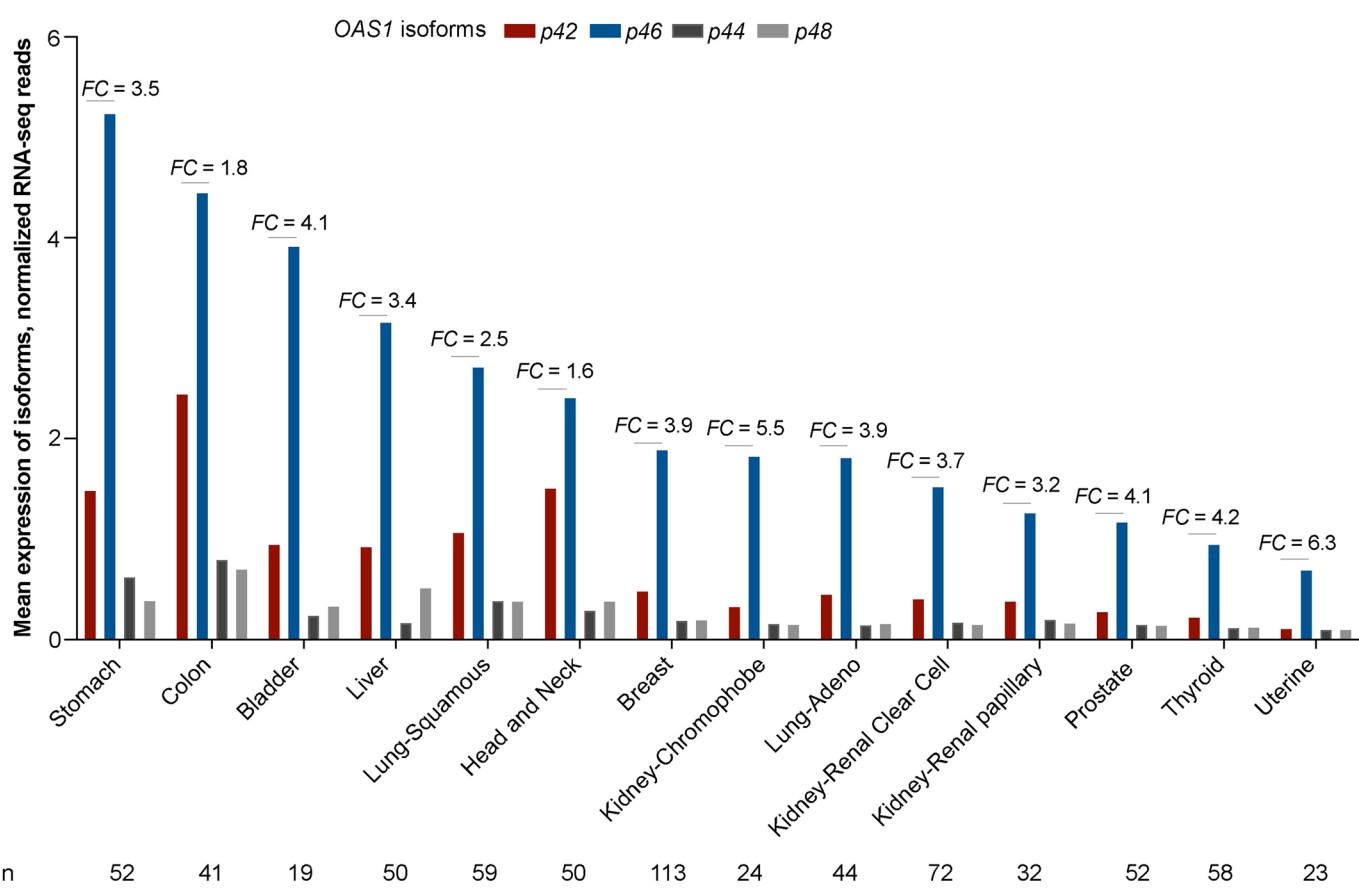

**Extended Data Fig. 7 | Mean expression levels of *OAS1* isoforms in tumor-adjacent normal tissues in TCGA.** Expression of *OAS1* isoforms was analyzed in RNA-seq data in The Cancer Genome Atlas (TCGA), using tumor-adjacent normal tissues with RNA-seq data in ≥15 samples. RNA-seq reads for each *OAS1* isoform were calculated based on unique splice junctions (*OAS1-p44, p46,* and *p48*) or unique 3'UTR (*p42*). For normalization, the total counts of RNA-seq reads for exon–exon junctions for *p44, p46* and *p48* were divided by 50 (length of an RNA-seq read), and unique RNA-seq reads for *p42* exon 5 by its length (317 bp). The mean expression levels of each isoform were calculated from all the samples with ≥3 RNA-seq reads. Overall, the mean expression of *OAS1-p46* is higher than *OAS1-p42*, with an average fold change (*FC*) of 3.9 across tissues.

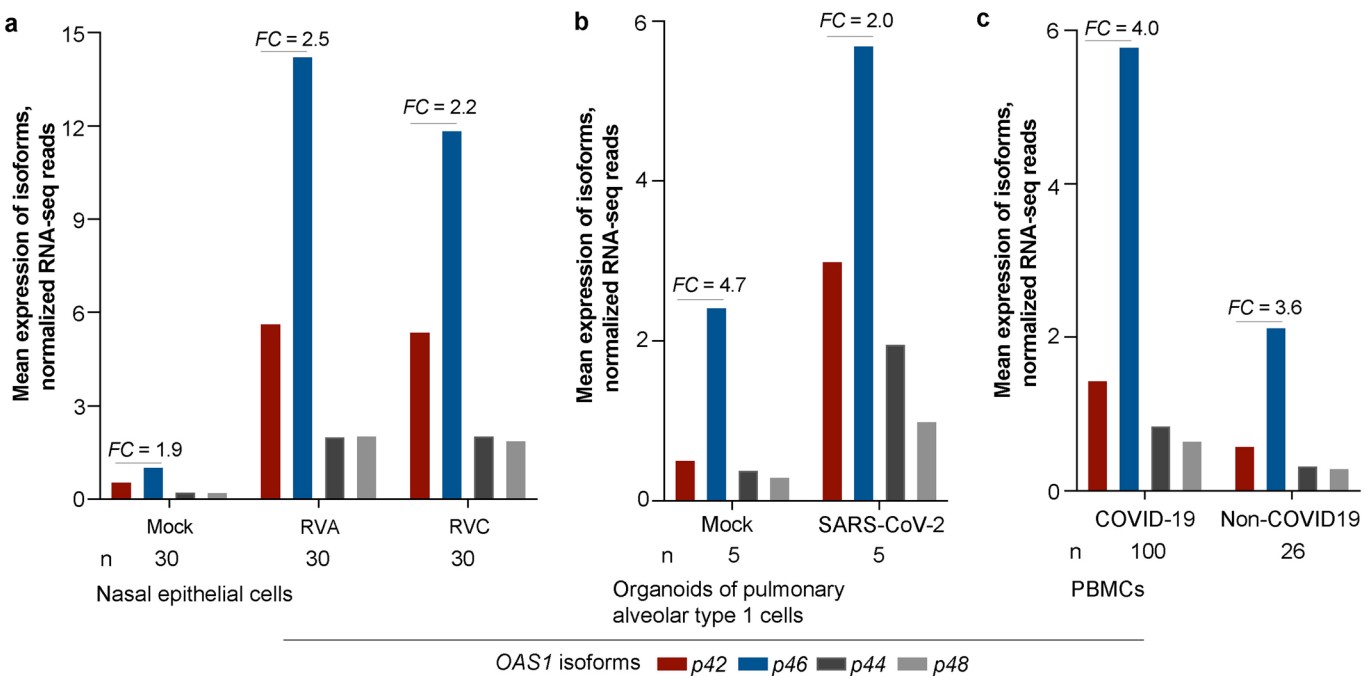

**Extended Data Fig. 8 | Mean expression levels of *OAS1* isoforms in nasal epithelial cells, pulmonary alveolar cells and PBMCs. a-c,** Expression of *OAS1* isoforms was analyzed in RNA-seq data of nasal epithelial cells (SRA: PRJNA627860) (**a**), organoids of pulmonary alveolar type 1 cells (SRA: PRJNA673197) (**b**), and PBMCs (SRA: PRJNA660067) (**c**). The bar graphs show the mean expression levels of each *OAS1* isoform normalized as described in Extended Data Figure 7. The expression of *OAS1-p46* is higher than *OAS1-p42* in all cell types and conditions tested.

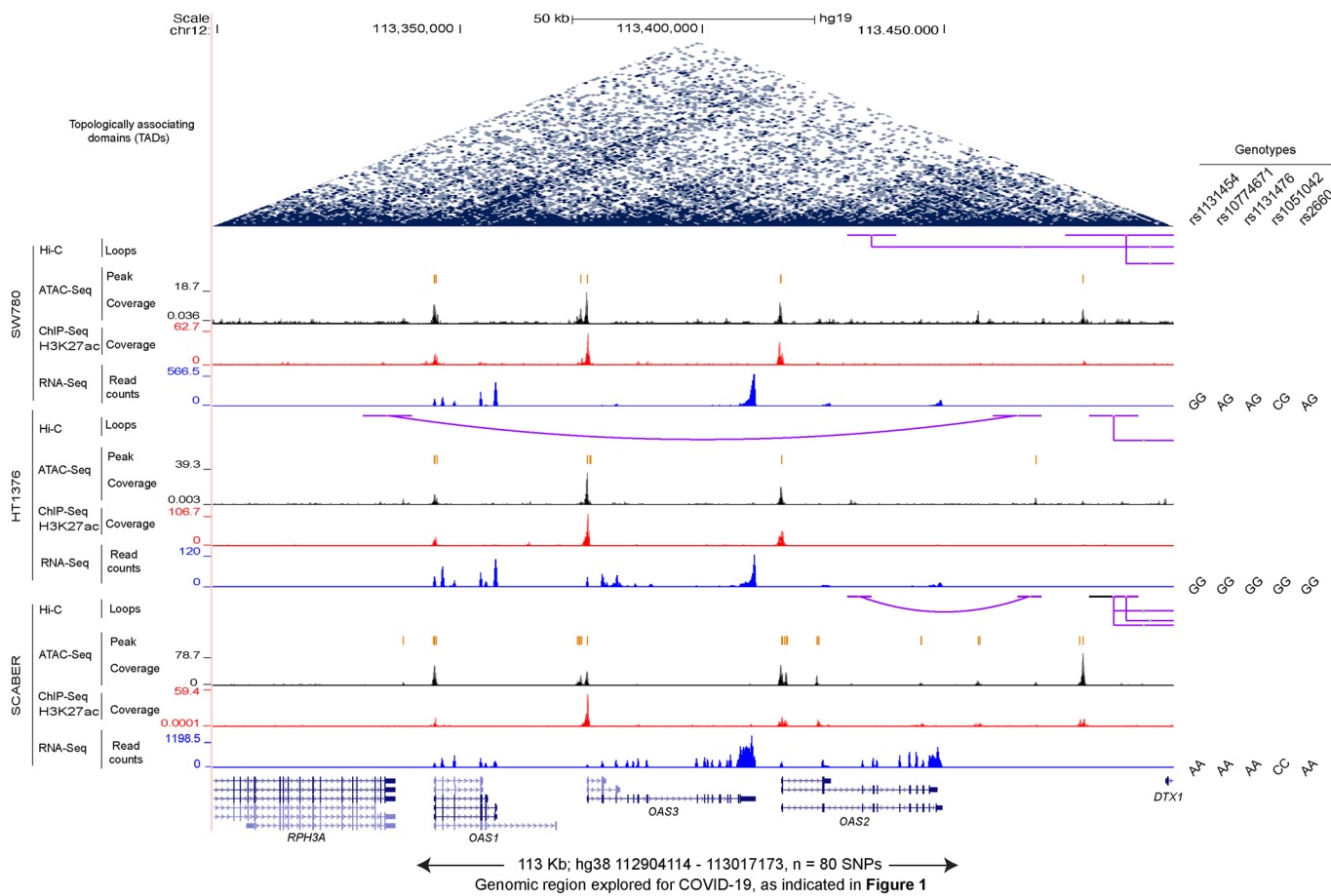

**Extended Data Fig. 9 | Multi-omics profile of the *OAS1-OAS3-OAS2* genomic region.** Multi-omics data for genome-wide Hi-C, ATAC-seq, H3K27ac ChIP-seq, and RNA-seq of three bladder cancer cell lines (SRA: PRJNA623018) were analyzed and visualized using UCSC genome browser. *OAS1*, *OAS2*, and *OAS3* expression was observed in all samples, but there is no evidence of open chromatin, enhancer activity, or chromatin interactions within the region that includes 79 variants associated with hospitalized versus nonhospitalized COVNET COVID-19 in patients of European ancestry (Fig. 1).

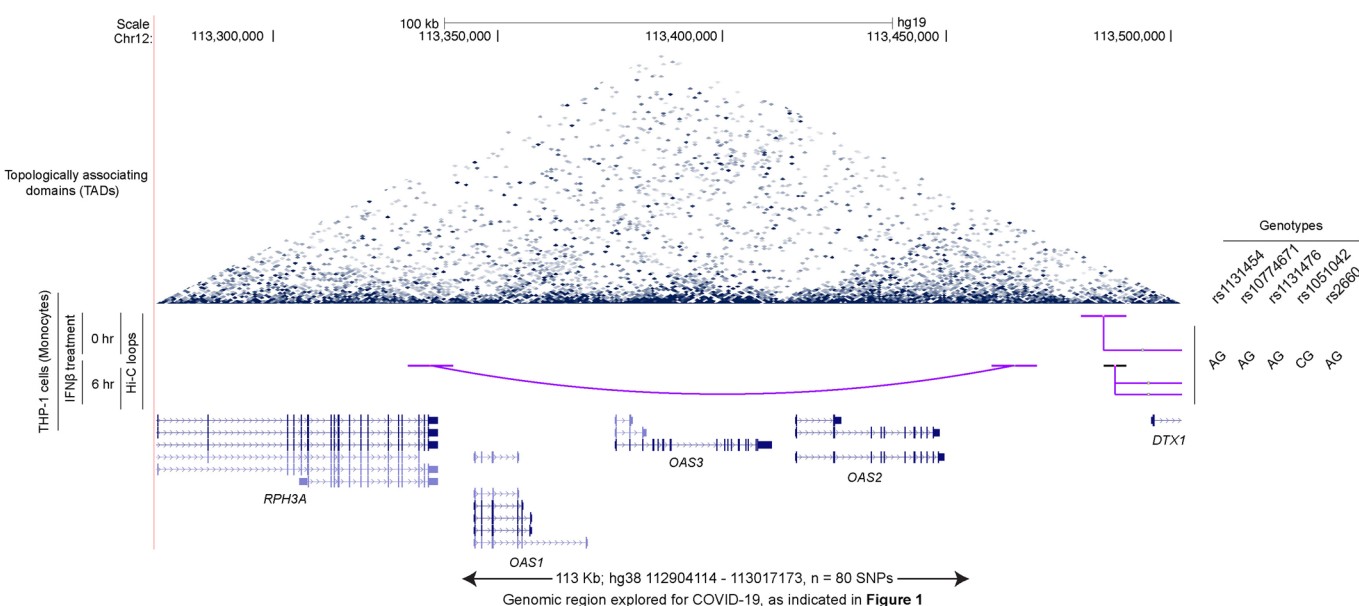

**Extended Data Fig. 10 | Hi-C chromatin interaction analysis in the *OAS1-OAS3-OAS2* genomic region.** Chromatin interaction (Hi-C) data for the THP-1 monocytic cell line untreated and treated with IFNβ for 6 h (SRA: PRJNA401748) were analyzed and visualized using UCSC genome browser. No chromatin interactions were detected with the region that includes 79 variants associated with hospitalized versus nonhospitalized COVNET COVID-19 in patients of European ancestry (Fig. 1).

# nature research

# Reporting Summary

Nature Research wishes to improve the reproducibility of the work that we publish. This form provides structure for consistency and transparency in reporting. For further information on Nature Research policies, see our Editorial Policies and the Editorial Policy Checklist.

## Statistics

For all statistical analyses, confirm that the following items are present in the figure legend, table legend, main text, or Methods section.

| n/a | Confirmed | |
|---|---|---|
| ☐ | ☒ | The exact sample size ($n$) for each experimental group/condition, given as a discrete number and unit of measurement |
| ☐ | ☒ | A statement on whether measurements were taken from distinct samples or whether the same sample was measured repeatedly |
| ☐ | ☒ | The statistical test(s) used AND whether they are one- or two-sided<br>*Only common tests should be described solely by name; describe more complex techniques in the Methods section.* |
| ☐ | ☒ | A description of all covariates tested |
| ☐ | ☒ | A description of any assumptions or corrections, such as tests of normality and adjustment for multiple comparisons |
| ☐ | ☒ | A full description of the statistical parameters including central tendency (e.g. means) or other basic estimates (e.g. regression coefficient) AND variation (e.g. standard deviation) or associated estimates of uncertainty (e.g. confidence intervals) |
| ☐ | ☒ | For null hypothesis testing, the test statistic (e.g. $F$, $t$, $r$) with confidence intervals, effect sizes, degrees of freedom and $P$ value noted<br>*Give P values as exact values whenever suitable.* |
| ☒ | ☐ | For Bayesian analysis, information on the choice of priors and Markov chain Monte Carlo settings |
| ☐ | ☒ | For hierarchical and complex designs, identification of the appropriate level for tests and full reporting of outcomes |
| ☐ | ☒ | Estimates of effect sizes (e.g. Cohen's $d$, Pearson's $r$), indicating how they were calculated |

*Our web collection on statistics for biologists contains articles on many of the points above.*

## Software and code

Policy information about availability of computer code

| Data collection | 1. Patients were recruited by several studies participating in COVNET: Large-scale Genome-wide Association Study and Whole Genome Sequencing of COVID-19 Severity (https://dceg.cancer.gov/research/how-we-study/genomic-studies/covnet).<br>2. COVNET DNA samples were analyzed by AmpFLSTR Identifiler and then genotyped for 712,191 variants using the Global Screening Array version 2.0 (GSA2, Illumina)<br>3. RNA-seq, ATAC-seq, H3K27ac ChIP-seq, and Hi-C datasets used in this study are listed in Table S15. These data were downloaded from NCBI SRA using SRA toolkit version 2.3.2. |
|---|---|
| Data analysis | 1. CONVET genetic data quality control was performed with PLINK (v1.9) and KING (v2.2.7). Ancestry estimation was done with GRAF (v2.3.1). Principal components analysis (PCA) of genetic data was performed with GCTA (v1.93.0 beta) to generate the first 20 eigenvectors, separately for each ancestry.<br>2. Imputation variants was performed using the TOPMed imputation server (https://imputation.biodatacatalyst.nhlbi.nih.gov/#!). Logistic regression analysis of genetic data with phenotypes were done with using glm function in R (v4.0.4). For haplotype analyses, ShapeIT (v2.r837) software was used to phase the selected variants and PLINK (v1.07) was used to perform associations. LD plots were generated with Haploview (v4.2).<br>3. The FASTQ files were compressed using GZIP (version 1.10) and aligned with STAR version 2.7.6a to the GRChg38/hg38 or hg19 genome assembly.<br>4. Bam files were indexed and sliced for specific genomic regions using SAM tools version 1.11<br>5. Exon quantification from BAM files was performed using ASpli package (version 1.5.1: https://bioconductor.org/packages/release/bioc/vignettes/ASpli/inst/doc/ASpli.pdf) on R platform (versions 3.6.0 to 4.0.4).<br>6. For ATAC-seq and H3K27ac ChIP-seq analysis, the FASTQ files were aligned to hg19 using ENCODE-DCC ATAC-seq-pipeline version 1.9.1 (https://github.com/ENCODE-DCC/atac-seq-pipeline) and ChiP-seq-pipeline2 version 1.6.1 (https://github.com/ENCODE-DCC/chip-seq-pipeline2) with default settings. The output bigwig files were then uploaded to the UCSC genome browser for visualization.<br>7. Hi-C, FASTQ files were processed using Juicer 1.6 (https://github.com/aidenlab/juicer) by selecting relevant restriction cutting sites such as Mbo I/Dpn II and aligned to hg19. The chromatin loops in Hi-C data were detected using Hiccups in Juicer tools with default settings. |

8. Integrative Genomics Viewer version 2.8.9 (http://www.broadinstitute.org/igv) was used for RNA-seq visualization
9. For identification of splicing factor binding motifs in exons, we used: Human Splicing Finder (HSF, www.umd.be/HSF3/)
10. Statistical analyses and plotting was performed using packages in R versions 3.6.0 to 4.0.4 and Prism - GraphPad (version 8)
11. Longitudinal trajectories of viral load in relation to genetic variants for clinical trail data were explored using a linear mixed-effects model function from the R nlme package (v3.1-153) to build linear mixed models
12. Nanopore GridION generated FASTQ files of long-read sequences were trimmed using Porechop version 0.2.4 and aligned to the hg19 genome using Minimap2 version 2.18

For manuscripts utilizing custom algorithms or software that are central to the research but not yet described in published literature, software must be made available to editors and reviewers. We strongly encourage code deposition in a community repository (e.g. GitHub). See the Nature Research guidelines for submitting code & software for further information.

## Data

Policy information about availability of data

All manuscripts must include a data availability statement. This statement should provide the following information, where applicable:
- Accession codes, unique identifiers, or web links for publicly available datasets
- A list of figures that have associated raw data
- A description of any restrictions on data availability

Summary statistics for all genetic analyses is provided in Supplementary Tables. Dataset for Oxford Nanopore RNA-seq was deposited as SRA: PRJNA743928. Full-length sequence data for OAS1-p42 transcript with Short exon 3 was deposited to NCBI GenBank with accession number MZ491787. Requests for any additional data or reagents should be addressed to L.P.-O. (prokuninal@mail.nih.gov).

# Field-specific reporting

Please select the one below that is the best fit for your research. If you are not sure, read the appropriate sections before making your selection.

☒ Life sciences          ☐ Behavioural & social sciences          ☐ Ecological, evolutionary & environmental sciences

For a reference copy of the document with all sections, see nature.com/documents/nr-reporting-summary-flat.pdf

# Life sciences study design

All studies must disclose on these points even when the disclosure is negative.

| | |
|---|---|
| Sample size | Sample size information for all types of data used in this study in provided in supplementary Tables. For omics data sets accessed from SRA, none of the samples were excluded from analysis based on 80% mappable sequence read cut-off. All samples with available data were used based on specific criteria such as genotypes. |
| Data exclusions | Except duplicates no samples were excluded from the analysis. |
| Replication | Yes, all experimental findings were reliably reproduced in multiple biological and technical replicates. Results were verified with minimum of three independent experiments. |
| Randomization | Not applicable for laboratory-based experimental data or genetic association studies |
| Blinding | All samples were processed blindly to their phenotype status |

# Reporting for specific materials, systems and methods

We require information from authors about some types of materials, experimental systems and methods used in many studies. Here, indicate whether each material, system or method listed is relevant to your study. If you are not sure if a list item applies to your research, read the appropriate section before selecting a response.

## Materials & experimental systems

| n/a | Involved in the study |
|---|---|
| ☐ | ☒ Antibodies |
| ☐ | ☒ Eukaryotic cell lines |
| ☒ | ☐ Palaeontology and archaeology |
| ☒ | ☐ Animals and other organisms |
| ☐ | ☒ Human research participants |
| ☐ | ☒ Clinical data |
| ☒ | ☐ Dual use research of concern |

## Methods

| n/a | Involved in the study |
|---|---|
| ☒ | ☐ ChIP-seq |
| ☒ | ☐ Flow cytometry |
| ☒ | ☐ MRI-based neuroimaging |

# Antibodies

| Antibodies used | Target gene | Cat. No. | Source | Target species | Host | Tag | Dilution |
|---|---|

Target gene  |  Cat. No. | Source | Target species | Host | Tag | Dilution
1. OAS1 PA5-82113 f Human Rabbit  1:200
2. GAPDH Ab9485 Abcam Human Rabbit  1:500
3. Golgin-97  A-21270 Thermo Fisher Tag Mouse  1:250
4. Anti-rabbit Alexa Flour 488  A21202 ThermoFisher 1:500
5. Anti-rabbit Alexa Fluor 680 A10043 ThermoFisher 1:500
6. Anti-Flag F7425-2MG Sigma Human Rabbit 1:1000
7. GAPDH 97166 Cell Signaling Technology Human mouse 1:1000
8. Anti-rabbit ab97051 Abcam 1:10,000
9. Anti-mouse ab6789 Abcam 1:10,000

**Validation**

We used antibodies that were validated as evidenced by Western Blot or Immunofluorescence images provided by vendors. Additionally, we used positive and negative controls to further assess the validity of all antibodies.
Vendor specific web-pages describing validation of antibodies are as:
1. OAS1 PA5-82113 | https://www.thermofisher.com/antibody/product/OAS1-Antibody-Polyclonal/PA5-82113
2. GAPDH Ab9485 |  https://www.abcam.com/GAPDH-antibody-Loading-Control-ab9485.html?gclsrc=aw.ds| aw.ds&gclid=CjwKCAjwve2TBhByEiwAaktM1BwiNq_fLMM0PVql4Jvkzz3hieNwi4BJp1UPN53zcBRvKbDAwADpTxoC2yEQAvD_BwE
3.  Golgin-97  A-21270 | https://www.thermofisher.com/antibody/product/Golgin-97-Antibody-clone-CDF4-Monoclonal/A-21270
4. Anti-rabbit Alexa Flour 488  A21202 | https://www.thermofisher.com/order/genome-database/generatePdf?productName=Mouse%20IgG%20(H+L)&assayType=PRANT&productId=A-21202&detailed=true
5. Anti-rabbit Alexa Fluor 680 A10043 | https://www.thermofisher.com/antibody/product/Donkey-anti-Rabbit-IgG-H-L-Highly-Cross-Adsorbed-Secondary-Antibody-Polyclonal/A10043
6. Anti-Flag F7425-2MG | https://www.sigmaaldrich.com/US/en/product/sigma/f7425
7. GAPDH 97166 | https://www.cellsignal.com/products/primary-antibodies/gapdh-d4c6r-mouse-mab/97166
8. Anti-rabbit ab97051 | https://www.abcam.com/goat-rabbit-igg-hl-hrp-ab97051.html
9. Anti-mouse ab6789 Abcam | https://www.abcam.com/goat-mouse-igg-hl-hrp-ab6789.html

# Eukaryotic cell lines

Policy information about cell lines

| Cell line source(s) | Most of the cell lines used were purchased from the American Type Culture Collection (ATCC). Detailed information is provided in Table S4. |
|---|---|
| Authentication | Cells lines were freshly purchased from ATCC or if used longer than 6 months, authenticated by genotyping of a panel of microsatellite markers  - Identifiler, performed by the Cancer Genomics Research Laboratory, NCI<br>Cells Source<br>HT-1376 -  ATCC ,STR profiling<br>Caco-2, ATCC ,used within 6 months of purchase<br>A549 ATCC STR profiling<br>ACE2-A549 stable - gift from Dr. Ralf Bartenschlager STR profiling<br>THP-1 - ATCC<br>T24-ATCC<br>SCaBER - ATCC<br>SW780- ATCC<br>HEBC - ATCC<br>Vero E6 - ATCC |
| Mycoplasma contamination | All cell lines in the laboratory are regularly tested for mycoplasma contamination using the MycoAlert Mycoplasma Detection kit (Lonza). Cell lines tested negative when compared to positive control |
| Commonly misidentified lines<br>(See ICLAC register) | No commonly misidentified cell lines were used. |

# Human research participants

Policy information about studies involving human research participants

| Population characteristics | *Describe the covariate-relevant population characteristics of the human research participants (e.g. age, gender, genotypic information, past and current diagnosis and treatment categories). If you filled out the behavioural & social sciences study design questions and have nothing to add here, write "See above."* |
|---|---|
| Recruitment | *Describe how participants were recruited. Outline any potential self-selection bias or other biases that may be present and how these are likely to impact results.* |

Ethics oversight

*Identify the organization(s) that approved the study protocol.*

Note that full information on the approval of the study protocol must also be provided in the manuscript.

# Clinical data

Policy information about clinical studies

All manuscripts should comply with the ICMJE guidelines for publication of clinical research and a completed CONSORT checklist must be included with all submissions.

Clinical trial registration | We used published data and samples from a clinical trial NCT04354259

Study protocol | *Note where the full trial protocol can be accessed OR if not available, explain why.*

Data collection | *Describe the settings and locales of data collection, noting the time periods of recruitment and data collection.*

Outcomes | *Describe how you pre-defined primary and secondary outcome measures and how you assessed these measures.*

