## [Peer Review File · Nature Genetics]

Peer Review Information

Manuscript Title: Genetic regulation of OAS1 nonsense-mediated decay underlies association with COVID-19 hospitalization in patients of European and African ancestries

Corresponding author name(s): Dr Ludmila Prokunina-Olsson

Reviewer Comments & Decisions:

Decision Letter, initial version:

25th August 2021

Dear Mila,

Your Article entitled "Genetic regulation of OAS1 nonsense-mediated decay underlies association with risk of severe COVID-19" has been seen by three referees, whose comments are copied below. While they find your work of potential interest, they have raised overlapping concerns that in our view are sufficiently important that they preclude publication of the work in Nature Genetics, at least in its present form.

Should further experiments and analyses allow you to fully address these criticisms, we would be willing to consider an appeal of our decision (unless, of course, something similar has by then been accepted at Nature Genetics or appeared elsewhere). This includes submission or publication of a portion of this work someplace else.

We hope you understand that until we have read the revised manuscript in its entirety we cannot promise that it will be sent back for peer review.

If you are interested in attempting to revise this manuscript for submission to Nature Genetics in the future, please contact me to discuss a potential appeal. Otherwise, we hope that you find our referees' comments helpful when preparing your manuscript for resubmission elsewhere.

Sincerely,
Kyle

Kyle Vogan, PhD
Senior Editor
Nature Genetics
<https://orcid.org/0000-0001-9565-9665>

Referee expertise:

Referee #1: Genetics, infectious diseases

Referee #2: Genetics, inflammatory diseases

Referee #3: Innate immunity, viral infections

Reviewers' Comments:

Reviewer #1:
Remarks to the Author:

The authors investigate the role of OAS1-3 in the severity and risk of COVID-19. This region was previously identified in prior literature, but the authors conduct fine mapping and functional follow-up to further elucidate its role in disease. There is a tremendous amount of work bridging statistical methods, functional experiments, and clinical trial data to answer this question. However, there were several points that were concerning and may not necessarily support the authors conclusions. Overall, I found that much of the manuscript was not related specifically to infection with SARS-CoV-2, but rather with identifying the genetic regulation of OAS1 expression unrelated to this specific virus. It is also unclear what spontaneous clearance refers to, considering most people clear detectable virus within a certain time frame, unlike hepatitis C or HIV. Specific comments are found below.

1. The previously-identified association of the OAS1 region with COVID-19 was only found in analyses comparing infection to the general population, but not when looking at severity (hospitalized versus non-hospitalized). However, the authors conduct their analyses with the exact same phenotypic definition (hospitalized versus non-hospitalized patients) and find strong signals. It is unclear why there is such a difference in results between the two analyses. Comparing various levels of disease severity to a general population which is likely uninfected is a comparison of infection risk, not disease severity. Comparing hospitalized to non-hospitalized within cases is a measure of severity. The authors should clarify if the effect sizes and allele frequencies were similar and expand on why there is inconsistent results between the two analyses. Page 14 lines 307-309 claim that they confirm prior studies when they do the exact opposite, finding a signal for severity where there was none in prior analyses.
2. The authors first fine-map the previously identified locus covering OAS1-3 using SuSiE, a method to finemap highly correlated data. This region is introgressed and in high LD, therefore it is not surprising that the credible set is unable to significantly narrow down the sites, resulting in one credible set of 76 variants and another of 12. This could be just due to sample size as the original analysis had a much larger sample size, therefore resulting in many more P-values above a certain threshold. This analysis is much smaller and therefore finds a smaller credible set because of relatively deflated p-values in the first place. Despite decreasing, it is still a large credible set and a large region including the original top variant.
3. What is the ancestry of the study participants? Neanderthal introgression is known to have different geographical frequencies which can be recapitulated in population substructure. It would be helpful to know if there are differences in your study population compared to the earlier manuscripts to explain the difference in results.
4. Throughout the manuscript, the focus is on comparison of p-values between analyses. Given the small sample sizes, much more focus should be on the effect sizes and whether they are consistent across analyses, such as the all hospitalized versus non-hospitalized and the severe hospitalized versus non-hospitalized.
5. I did not entirely follow the jump from the 76 variant credible set to the 4 missense variants in OAS1. Were there no eQTLs for surrounding genes in this credible set? Did these four variants have the highest posterior probabilities in the credible set?
6. It is unclear how the two sections starting on page 9 line 205 to page 12 line 246 are related to SARS-CoV-2 infection at all. This is all focused on splicing variants with OAS1 isoforms, but none of this is related back to this specific virus.
7. In general I do not understand the framework of impaired spontaneous clearance of SARS-CoV-2, considering that it does not seem to exhibit chronic infections like other viruses (such as HCV and HIV). What is impaired spontaneous clearance? This is especially unclear as these are evaluated in mild cases where there seems to be no consequences for possibly taking longer to clear the virus.
8. I would hesitate to come to any conclusions based on the trial data from 58 participants, which are further split into haplotypes and treatment arms. While there was marginal significance using a $P < 0.05$ threshold at days 5 and 10 in the placebo arm, there was no difference on days 7 and 10, which seems odd. It would also be more appropriate to conduct a longitudinal analysis looking for linear trends, instead of multiple comparisons at irregular periods of time.
9. The last sentence of the manuscript is not supported by the evidence within the manuscript. It is unclear what deficient spontaneous clearance is with SARS-CoV-2 as this is never defined and all

patients had a mild disease course. If individuals with these “risk” OAS1 haplotypes have a mild disease course, then why do they need treatment? How is the effect of interferon treatment related to these specific haplotypes? And why should these be further explored in clinical trials when their relevance has not been established by the evidence base in this manuscript. These conclusions would be much better supported by showing these effects in severe cases, since the associations in the first section (where the finemapping occurred) were comparing severe to mild cases.

Reviewer #2:
Remarks to the Author:

The key findings for me are:

- Splice variant rs10774671 - lead variant in the analysis of severe disease reported by the HGI - is associated with risk of hospitalization among infected individuals (all hospitalized [n=954] vs. not-hospitalized [n=601]). Figure 1a
- When a lung cell line was infected with SARS-CoV-2, there were no differences in attenuation of viral load between OAS1 proteins carrying different missense variants that are in high LD with rs10774671.

The authors then present results from an additional set of functional experiments and analysis of a small clinical trial, but I find these less convincing/relevant to help understand the association between the OAS1 locus and severe COVID-19.

Line 143, please indicate here how association analyses were run (e.g. logistic regression in R, with age, sex, etc. included as covariates), and provide metrics of genomic inflation (i.e. lambda and LD-score regression intercept; based on genome-wide results).

Line 146, please add sample size to each group (hospitalized and non-hospitalized).

Posterior probabilities for individual variants from SuSIE are very low (Figure 1b), with a max of 0.2. For example, rs10774671 has a PIP very close to zero. So how confident are you that your fine mapping analysis is adequately identifying the likely causal variants at this locus?

If the signal (i.e. lead variant) at this locus is the same between your analysis of hospitalized (n=954) vs. not-hospitalized (n=601) and the A2 analysis of the HGI, then is it not more accurate to use the HGI results to fine-map this region, given the much larger sample size? Have you tried to fine map the HGI results? Does your analysis really improve (i.e. narrow) the credible set from the HGI?

Related to this, it is noteworthy that rs10774671 is not associated with hosp vs. not-hosp (B1 phenotype) in the HGI ($P = 0.8$), with almost 6K cases and 15K controls. Why do you think that this is the case?

Are there any missense variants in high LD with rs10774671? If so, please indicate in text. [I see now that this is mentioned in line 180 – I would move this up a few paragraphs.]

Analysis of the severe hospitalized (n=566) vs. not-hospitalized (n=601) phenotype identified an additional set of 24 variants with $0.005 < P < 0.05$. As these variants are in modest LD with rs10774671, is it not possible that the weaker signal with these additional variants is fully explained by rs10774671? Comparing Figure 1a with 1c suggests that the signals in the region are very similar. The authors should repeat the association analysis of severe hosp vs. not hospitalized with

rs10774671 included as a covariate (and make an additional locuszoom plot with those results). Looking at Figure 1e, it's possible that rs1131454 is driving the two significant associations shown in the bottom part of this figure, but both might disappear when conditioning on rs10774671. Please also add frequency of each haplotype to this figure.

Line 164, "we selected four directly genotyped variants spanning a 4.2-kb region (rs10774671, rs1131476, rs2660, and rs4766664)". If there are only two common haplotypes underlying this signal (which makes sense given that all the associated variants are in very high LD), why do we need multiple variants (i.e. a haplotype) to define the signal in this region? Why is this useful? And why did you select four variants, not more or not less? Why not just use rs10774671?

Line 169, three (of the 24) variants identified in the severe hospitalized vs. not-hospitalized analysis were then added to the four variants included in the common risk haplotype. If the association with these 3 variants is explained by LD with rs10774671, I don't think this analysis is particularly informative.

Line 173 to 174, "In smaller sets of patients of non-European ancestries (483 African, 166 admixed-Hispanic, and 103 East-Asian), none of these variants or their haplotypes showed significant associations". I suggest focusing the text on results for rs10774671, explicitly reporting the allele frequency and association results in the three different groups. Also, it would be helpful to indicate power to detect a significant association with the European odds ratio in each of these smaller groups.

Line 180, "Our analyses suggested that in Europeans at least one effect variant for the risk of hospitalized disease is included in the 95% credible set of 76 OAS1 variants." This could be stated more clearly. I think what you mean is that the fine mapping analysis of your hosp vs. not hosp results identified a credible set of 76 variants, which is likely to include the (or at least one) causal variant underlying the association between this locus and the A2 phenotype reported by the HGI.

Line 182, "included four missense variants", helpful to indicate that these were missense variants for OAS1.

Line 221 to 246. This section focuses predominantly on missense variant rs1131454, but this variant is not part of the credible set for the common risk signal in this region. Instead, it has weak associations with the hosp severe vs. not-hosp phenotype, which could potentially be explained by the modest LD with rs10774671. So not clear how relevant these functional data are to understand the common risk signal in this region.

Line 288 to 303. Very small sample size for the genetic analysis in the clinical trial data (n=28 and n=30, respectively in placebo and treatment group). The weak association between the rs10774671/rs1131454 haplotype and viral load in the placebo group is not very convincing, it would not remain significant after correcting for multiple testing. I also think it would make more sense to just analyze rs10774671 per se, given my reservations with the association with rs1131454, as mentioned above. Not clear why rs2660 is included in this analysis (Figure 7d), as it does seem informative (high LD with rs10774671).

Line 702. "Additionally, whole-genome sequencing data generated by contributing studies were used for a subset of samples." What proportion of samples had available whole-genome sequencing data and how were these data used?

Line 703. "Sex, relatedness and population groups (European, African, admixed-Hispanic and East-Asian) were determined based on genetic analyses for all patients." Please provide more detail. How exactly were samples assigned to continental ancestries?

Lines 708 to 714. Did you only impute chr 12? I think this is highly unlikely. Very important to have association results for the full genome, to be able to assess if associations statistics are well behaved (based on lambdas, etc.). In the unlikely event that imputed data is not available for the whole genome, then this should be performed with the array data.

Reviewer #3:

Remarks to the Author:

The human genomic locus chr12q24.13 encoding 2',5'-oligoadenylate synthetases (OASs) 1-to-3 has been previously associated with severity of COVID-19. In this study of 1555 Covid-19 cases of European ancestry, a single haplotype of 76 OAS1 variants, included in a genomic fragment introgressed from Neandertals, was associated with risk of hospitalization of COVID-19 patients, and with spontaneous but not treatment induced SARS-CoV-2 clearance with pegIFN-lambda1. Two exonic variants examined affected splicing and nonsense-mediated decay (NMD) of OAS1 mRNAs. Data support that it is not transcriptional control that is responsible for higher levels of OAS1-p46, but rather control of mRNA stability through NMD. OAS1-p46-long exon 3 lacked premature termination codons, in contrast to other OAS1 transcripts. SiRNA knockdowns of NMD components convincingly demonstrate that the OAS1-p46 long isoform was resistant to NMD. Further results suggest that the OAS1 risk haplotypes with impaired spontaneous clearance of SARS-CoV-2 could be compensated by early treatment with IFN-lambda. In other words, pegIFN-lambda1 overcame the deficiency of OAS1 in the high-risk group.

This paper advances knowledge of how alternative splicing controls levels of OAS1 proteins. NMD mRNA decay reduces expression of the p42 OAS1 isoform, which is probably responsible for the relatively higher levels of the alternative spliced p46 OAS1 isoform. It is not transcription control, or differences in specific activities of p42 vs. p46 OAS1, but rather the higher levels of mRNA encoding p46 that sets protein levels, since the mRNA for this isoform is relatively stable.

The following point should be addressed in a revised manuscript. In Figure 2c, the effects of expressing the risk (OAS1-p42) and non-risk (OAS1-p46) isoforms on SARS-CoV-2 yields in A549-ACE2 cells were determined. The two OAS1 isoforms showed similar antiviral activity. In contrast, a recent study reported that the prenylated and endomembrane targeted OAS1-p46 isoform had greater antiviral activity than the p42 OAS1 isoform (ref. 21, this reference should be updated as published in eLife: PMID: 34342578). However, the experiment in Figure 2c requires Western blots for p42 and p46 to monitor relative expression the different OAS1 isoforms (with antibody to the Flag epitope-tag), controlled for beta-actin levels. Are expressed levels of p42 and p46 similar, or, if different, do levels of these proteins correlate with the antiviral effects?

**Although we cannot publish your paper at this time, it may be appropriate for another journal in the Nature Portfolio. If you wish to explore the journals and transfer your manuscript please use our [Redacted] manuscript transfer portal. If you transfer to Nature journals or the Communications journals, you will not have to re-supply manuscript metadata and files. This link can only be used once and remains active until used.

All Nature Portfolio journals are editorially independent, and the decision on your manuscript will be taken by their editors. For more information, please see our manuscript transfer FAQ page.

Note that any decision to opt in to In Review at the original journal is not sent to the receiving journal on transfer. You can opt in to [In Review](https://www.nature.com/nature-research/for-authors/in-review) at receiving journals that support this service by choosing to modify your manuscript on transfer. In Review is available for primary research manuscript types only.

Author Rebuttal to Initial comments

Dear Kyle,

Happy New Year and I hope you are doing well!

I would like to inquire about an opportunity to resubmit our paper, which is now titled “**Genetic regulation of *OAS1* nonsense-mediated decay underlies association with COVID-19 hospitalization in patients of European and African ancestries**”.

We are aware of the Brief Communication recently published in NG "Multi-ancestry fine mapping implicates *OAS1* splicing in risk of severe COVID-19", but believe our paper is non-redundant to this publication.

Specifically, our paper is the only one so far (published or in preprint) that is based on case-case analyses, while other papers compare COVID-19 patients to the general population; our analyses are not limited by one or a few variants of presumed functional significance; we identified a commonality in genetic signals in patients of European and African ancestries based on the analyses of the whole region, and we provide mechanistic and clinical insights into these associations.

Below is the summary of the specific updates included in the paper:

1. We expanded the sample sets of COVID-19 patients of European ancestry from 1555 to 2249 and patients of African ancestry from 483 to 835.
2. We addressed the issue of cryptic population substructure and provide genomic inflation factor values based on genome-wide genotyped markers: $\lambda=1.01$ in patients of European ancestry and $\lambda=1.0$ in patients of African ancestry.
3. We performed haplotype analyses using four *OAS1* markers. Three of those (rs1131454, rs2660 and rs4766664) are significantly associated with hospitalized vs. non-hospitalized COVID-19 both in patients of European and African ancestries; rs2660 represents two additional variants in complete LD (missense variants rs1051042 and rs1131476). We added rs10774671 as a previous COVID-19 susceptibility GWAS lead (HGI, PMID: 34237774, although this SNP was significantly associated in patients of European ($p=6.5E-04$), but not of African ancestry ($p=0.079$).

4. These four selected markers captured the previously described introgressed Neandertal haplotype and directly genotyped these markers in BAM files of three Neandertals and evaluated their ancestral and Denisova status as well.
5. We demonstrate that the specific (ancestral/Neandertal) haplotype, rather than individual markers comprising it, is associated with protection from hospitalized COVID-19 both in patients of European (OR=0.76, $p=1.0E-03$) and African ancestry (OR=0.46, $p=1.39E-03$). However, the rs10774671-G allele was not significantly protective when included in another, common African-specific haplotype (OR=0.76, $p=0.059$). Although this haplotype might eventually be associated in larger sample sets, its effect size (OR=0.76, $p=0.059$) is not as strong as of the Neandertal haplotype (OR=0.46, $p=1.39E-03$).
6. We reanalyzed the data from the clinical trial with pegIFN- λ 1 using mixed models for longitudinal repeated measures to confirm the effect of specific variants and their haplotypes on SARS-CoV-2 clearance.
7. We provided HGI-r6 results for OAS1 variants identified by our analyses. In B2 analysis (hospitalized patients with the general population), rs10774671, which affects OAS1 splicing, shows a stronger association ($p=7.99E-12$) than rs1131454 ($p=7.41E-07$). However, in the B1 analysis (hospitalized vs. non-hospitalized COVID-19), the association for these markers is weaker but comparable, with $p=9.93E-04$ for rs10774671 and $p=7.14E-04$ for rs1131454.
8. We included the Western blot analysis data showing that all OAS1-Flag plasmids produced a similar amount of corresponding proteins.
9. We changed the title from "Genetic regulation of OAS1 nonsense-mediated decay underlies association with risk of severe COVID-19" to "Genetic regulation of *OAS1* nonsense-mediated decay underlies association with COVID-19 hospitalization in patients of European and African ancestries".

Abstract:

The chr12q24.13 locus encoding OAS1-3 antiviral proteins has been associated with COVID-19 susceptibility. Here, we report genetic, functional, and clinical insights into this locus in relation to COVID-19 severity. We analyzed patients of European ($n=2249$) and African ($n=835$) ancestries with hospitalized vs. non-hospitalized COVID-19. In both ancestries, protection from hospitalized disease was consistently associated with one ancestral/introgressed Neandertal haplotype comprising several *OAS1* variants. The *OAS1* haplotypes were also associated with SARS-CoV-2 clearance in a clinical trial with pegIFN- λ 1. We demonstrate the combined functional effect of two exonic variants in *OAS1* included in the protective haplotype. The associated splicing rs10774671 and missense rs1131454 variants increased OAS1 protein abundance through the regulation of splicing and nonsense-mediated decay. Thus, *OAS1* expression is elevated in the presence of genetic variants or due to treatment with interferons, improving SARS-CoV-2 clearance and decreasing the risk of hospitalization for COVID-19 in patients of European and African ancestries.

Thank you for your time and consideration,
Kind regards,
Mila

Decision Letter, first revision:

IMPORTANT: Please note the reference number: NG-A57954R-Z Prokunina-Olsson. This number must be quoted whenever you communicate with us regarding this paper.

28th January 2022

Dear Mila,

Thank you for asking us to consider a resubmission of your revised manuscript entitled "Genetic regulation of OAS1 nonsense-mediated decay underlies association with COVID-19 hospitalization in patients of European and African ancestries". I have discussed your proposed resubmission with my editorial colleagues, and we invite you to upload your revised manuscript and point-by-point response for further editorial assessment and peer review.

When preparing a revision, please ensure that it fully complies with our editorial requirements for format and style; details can be found in the Guide to Authors on our website (<http://www.nature.com/ng/>).

Please be sure that your manuscript is accompanied by a separate letter detailing the changes you have made and your response to the points raised. At this stage we will need you to upload:

- 1) A copy of the manuscript in MS Word .docx format.
- 2) The Editorial Policy Checklist:
<https://www.nature.com/documents/nr-editorial-policy-checklist.pdf>
- 3) The Reporting Summary:
<https://www.nature.com/documents/nr-reporting-summary.pdf>

(Here you can read about the role of the Reporting Summary in reproducible science:
<https://www.nature.com/news/announcement-towards-greater-reproducibility-for-life-sciences-research-in-nature-1.22062>)

Please use the link below to be taken directly to the site and view and revise your manuscript:

[Redacted]

With kind wishes,
Kyle

Kyle Vogan, PhD
Senior Editor
Nature Genetics
<https://orcid.org/0000-0001-9565-9665>

Author Rebuttal, first revision:

Dear Dr. Vogan,

We appreciate the constructive comments and suggestions of the reviewers. Below, we provide point-by-point responses also reflected in our updated manuscript.

The main updates in this manuscript:

1. We expanded the sample sets of COVID-19 patients of European ancestry from 1555 to 2249 and patients of African ancestry from 483 to 835.
2. We addressed the issue of cryptic population substructure and provide genomic inflation factor values based on genome-wide genotyped markers: $\lambda=1.01$ in patients of European ancestry and $\lambda=1.0$ in patients of African ancestry.
3. We performed haplotype analyses using four OAS1 markers. Three of those (rs1131454, rs2660 and rs4766664) were significantly associated with hospitalized vs. non-hospitalized COVID-19 both in patients of European and African ancestries; rs2660 represents two additional variants in complete LD (missense variants rs1051042 and rs1131476). We added rs10774671 as a previous COVID-19 susceptibility GWAS lead (HGI, PMID: 34237774), although this SNP was significantly associated in patients of European ($p=6.5E-04$), but not of African ancestry ($p=0.079$).
4. These four selected markers captured the previously described introgressed Neandertal haplotype. We directly genotyped these markers in BAM files of three Neandertals and evaluated their ancestral and Denisova status as well.
5. We demonstrate that the specific (ancestral/Neandertal) haplotype, rather than individual markers comprising it, is associated with protection from hospitalized COVID-19 both in patients of European (OR=0.76, $p=1.0E-03$) and African ancestry (OR=0.46, $p=1.39E-03$). However, the rs10774671-G allele was not significantly protective when included in another, common African-specific haplotype (OR=0.76, $p=0.059$). Although this haplotype might eventually be associated in larger sample sets, its effect size (OR=0.76, $p=0.059$) is not as strong as of the Neandertal haplotype (OR=0.46, $p=1.39E-03$).
6. We provided HGI-r6 results for OAS1 variants identified by our analyses. In B2 analysis (hospitalized patients vs. the general population), rs10774671, which affects OAS1 splicing, shows a stronger association ($p=7.99E-12$) than rs1131454 ($p=7.41E-07$). However, in the B1 analysis (hospitalized vs. non-hospitalized COVID-19), the association for these markers is comparable albeit weaker, with $p=9.93E-04$ for rs10774671 and $p=7.14E-04$ for rs1131454. This is in line with our results in hospitalized vs. non-hospitalized patients of European and African ancestries.
7. We reanalyzed the data from the clinical trial with pegIFN- $\lambda 1$ using mixed models for longitudinal repeated measures to confirm the effect of specific variants and their haplotypes on SARS-CoV-2 clearance.
8. We included the Western blot analysis data showing that all OAS1-Flag plasmids produced similar amounts of corresponding proteins.
9. We changed the title from "Genetic regulation of OAS1 nonsense-mediated decay underlies association with risk of severe COVID-19" to "Genetic regulation of OAS1 nonsense-mediated decay underlies association with COVID-19 hospitalization in patients of European and African ancestries".

10. We included additional functional data to show that all four OAS1 plasmids had a similar effect on cell growth (Figure S4). We also added data on a candidate variant, rs1859331 within 5'UTR of OAS3 (Figure S12, S13). This variant is associated with patients of European ancestry but didn't reach significance in patients of African ancestry. We demonstrated a potential cell-type specific effect of this variant on protein translation initiation efficiency that would affect OAS3 protein expression.

Reviewers' Comments:

Reviewer #1:

Remarks to the Author:

The authors investigate the role of OAS1-3 in the severity and risk of COVID-19. This region was previously identified in prior literature, but the authors conduct fine mapping and functional follow-up to further elucidate its role in disease. There is a tremendous amount of work bridging statistical methods, functional experiments, and clinical trial data to answer this question. However, there were several points that were concerning and may not necessarily support the authors conclusions. Overall, I found that much of the manuscript was not related specifically to infection with SARS-CoV-2, but rather with identifying the genetic regulation of OAS1 expression unrelated to this specific virus. It is also unclear what spontaneous clearance refers to, considering most people clear detectable virus within a certain time frame, unlike hepatitis C or HIV. Specific comments are found below.

1. The previously-identified association of the OAS1 region with COVID-19 was only found in analyses comparing infection to the general population, but not when looking at severity (hospitalized versus non-hospitalized). However, the authors conduct their analyses with the exact same phenotypic definition (hospitalized versus non-hospitalized patients) and find strong signals. It is unclear why there is such a difference in results between the two analyses. Comparing various levels of disease severity to a general population which is likely uninfected is a comparison of infection risk, not disease severity. Comparing hospitalized to non-hospitalized within cases is a measure of severity. The authors should clarify if the effect sizes and allele frequencies were similar and expand on why there is inconsistent results between the two analyses. Page 14 lines 307-309 claim that they confirm prior studies when they do the exact opposite, finding a signal for severity where there was none in prior analyses.

Response:

We thank the reviewer for acknowledging our "tremendous amount of work bridging statistical methods, functional experiments, and clinical trial data" to explore the genetic association within the chr 12 locus with COVID-19 outcomes.

Some specific comments:

"Much of the manuscript was not related specifically to infection with SARS-CoV-2 but rather to identifying the genetic regulation of OAS1 expression unrelated to this specific virus".

Response: This region represents a very interesting opportunity to explore the genetically-regulated innate immune responses to SARS-CoV-2. The introgression of an extended haplotype known as "the Neandertal haplotype" and its retention at a high frequency in non-African populations suggested its

beneficial functional role but through unknown mechanisms. Previously, only rs10774671 was considered as the functional variant underlying this haplotype's retention after introgression. We agnostically explored this region in relation to anti-SARS-CoV-2 immunity/COVID-19 severity in individuals of European and African ancestries. Our various functional studies were essential for uncovering molecular mechanisms of the association in this region. We show that although rs10774671 is an important functional part of this protective ancestral/Neandertal haplotype, the rs10774671-G allele is most protective only when included in the specific haplotype but not in a common African-specific haplotype. We demonstrated that several variants synergistically contribute to the protective effect of the ancestral/Neandertal haplotype. This study is important for understanding the biology of SARS-CoV-2 infection and the genetic regulation of the immune response, in general.

"It is also unclear what spontaneous clearance refers to".

Response: Although we used the term "spontaneous clearance" to differentiate natural from treatment-induced clearance, we removed it in the current version.

"The authors claim that they confirm prior studies when they do the exact opposite, finding a signal for severity where there was none in prior analyses".

Response:

Our study is agnostic to previous results. Our goal was to perform a detailed genotype and haplotype analyses of COVID-19 severity in independent samples. We analyzed outcomes of laboratory-confirmed SARS-CoV-2 before vaccination and the emergence of viral variants. In our study, medical data for all the patients were reviewed by the same clinical team to assign the status (non-hospitalized or hospitalized due to COVID-19) and further stratify into moderate and severe hospitalized disease based on death or mechanical ventilation. All the samples used in our study (except for a small subset of WGS samples) were genotyped in one center, and 65.5% of European and 99.6% of African samples were also TaqMan-genotyped for two main variants, rs10774671 and rs1131454; several other variants are array-genotyped. We used 0.8 imputation threshold and implemented rigorous quality analyses and assignment into ancestral groups. All our analyses were done in exactly the same sets of patients (100% genotyping/imputation rate), making these results directly comparable between markers. With all that, we believe our results are sound, technically robust and supported by a wide range of biological data.

The previous version of the HGI results (r5, mentioned in our initial manuscript) did not report an association with COVID-19 severity for any markers in this region. However, the results are different in the current version (HGI-r6). In B2 analysis (hospitalized patients vs general population) rs10774671, which affects OAS1 splicing, shows a much stronger association than rs1131454 (see below and Table S3). However, in B1 analysis (hospitalized vs. non-hospitalized COVID-19), both these variants show weaker but comparable associations among the top in this region. These results might reflect some biological mechanisms of the OAS1 variants manifesting only after infection.

rsID	HGI ID, hg38, NR-R allele	annotation	HGI-r6, B2 analysis		HGI-r6, B1 analysis		COVNET meta-analysis, n=3084, European and African ancestry				
			p-value	beta	p-value	beta	P-value	OR	L95	U95	I
rs1131454	12-112911065-G-A	OAS1, Gly162Ser, ex 3	7.41E-07	0.056	7.14E-04	0.054	1.87E-04	1.296	1.132	1.483	0
rs10774671	12-112919388-G-A	splicing	7.99E-12	0.075	9.93E-04	0.052	1.48E-04	1.294	1.134	1.476	0
rs1131476	12-112919404-G-A	OAS1, Ala352Thr, ex 6	3.37E-10	0.072	2.91E-03	0.05	1.91E-04	1.335	1.148	1.553	0
rs1051042	12-112919432-G-C	OAS1, Arg361Thr, ex 6	3.46E-06	0.057	5.02E-03	0.048	1.91E-04	1.335	1.148	1.553	0
rs2660	12-112919637-G-A	OAS1, 3'UTR	2.73E-10	0.073	2.99E-03	0.05	1.91E-04	1.335	1.148	1.553	0
rs4766664	12-112925192-T-G	OAS1 intron	4.09E-10	0.072	3.40E-03	0.049	6.48E-05	1.367	1.174	1.591	32.39

B1 analysis: Hospitalized COVID19 vs non-hospitalized COVID-19
Analysis ID: r6-nStudies-23-nCases-14480-nControls-73191 (leave_23andme)
B2 analysis: Hospitalized COVID19+ vs. population controls
Analysis ID: r6-nStudies-43-nCases-24274-nControls-2061529 (leave_23andme)
Source: Covid-19 Host Genetics Initiative Browser, <https://app.covid19hg.org/variants>
COVNET: hospitalized vs non-hospitalized COVID-19, analysis is based on the same sample numbers for all markers

The authors should clarify if the effect sizes and allele frequencies were similar and expand on why there is inconsistent results between the two analyses.

Response: Figure S1 outlines our QC and analytical pipeline. Our analysis was done in specific genetically-defined ancestral groups with additional adjustments for 20 principal components (PCs) and relevant covariates. Since the HGI results are based on meta-analysis of various ancestries and variable numbers of studies, this complicates direct comparisons with our results. Based on available information, the direction of the effects is the same in the HGI and our analyses (Table S3).

2. The authors first fine-map the previously identified locus covering OAS1-3 using SuSiE, a method to finemap highly correlated data. This region is introgressed and in high LD, therefore it is not surprising that the credible set is unable to significantly narrow down the sites, resulting in one credible set of 76 variants and another of 12. This could be just due to sample size as the original analysis had a much larger sample size, therefore resulting in many more P-values above a certain threshold. This analysis is much smaller and therefore finds a smaller credible set because of relatively deflated p-values in the first place. Despite decreasing, it is still a large credible set and a large region including the original top variant.

Response: We agree that SuSie analysis in this region was not very informative and excluded it from the current version.

3. What is the ancestry of the study participants? Neanderthal introgression is known to have different geographical frequencies which can be recapitulated in population substructure. It would be helpful to know if there are differences in your study population compared to the earlier manuscripts to explain the difference in results.

Response: Figure S1 outlines our QC and analytical pipeline. In the updated manuscript, we show the association of the same introgressed ancestral/Neanderthal haplotype with protection from hospitalized COVID-19 in patients of both European and African ancestries. Because this haplotype does not exist in modern African populations of the 1000 Genomes Project but is found in African-American individuals (ASW in 1000 Genomes) and our COVID-19 patients, we suggest that it was introduced to individuals of African ancestry through historic admixture with non-African populations. We show frequencies of this and other haplotypes in different ancestral population groups in Table S4.

As explained in Figure S1, the ancestry of our patients was assigned by GRAF software and then additionally controlled for population substructure by generating 20 PCs for groups of patients included in the final analyses. Previous publications are based on meta-analyses of individuals of various ancestries, precluding direct comparisons with our results.

4. Throughout the manuscript, the focus is on comparison of p-values between analyses. Given the small

sample sizes, much more focus should be on the effect sizes and whether they are consistent across analyses, such as the all hospitalized versus non-hospitalized and the severe hospitalized versus non-hospitalized.

Response: We now present the results as ORs in Figure 1 and both ORs and p-values in Supplementary Tables.

5. I did not entirely follow the jump from the 76 variant credible set to the 4 missense variants in OAS1. Were there no eQTLs for surrounding genes in this credible set? Did these four variants have the highest posterior probabilities in the credible set?

Response: We removed the section on credible set analysis as it is not informative in this specific situation (limited sample set, many markers in high LD). We focused on four OAS1 markers: three of those (rs1131454, rs2660, rs4766664) were significantly associated with hospitalized vs non-hospitalized COVID-19 both in patients of European and African ancestries. We also added to this set rs10774671 as a previous COVID-19 susceptibility GWAS lead, although this SNP was significantly associated only in patients of European (OR=1.33, p=6.5E-04) but not in African ancestry (OR=1.23, p=0.079).

6. It is unclear how the two sections starting on page 9 line 205 to page 12 line 246 are related to SARS-CoV-2 infection at all. This is all focused on splicing variants with OAS1 isoforms, but none of this is related back to this specific virus.

Response: We were interested in understanding the anti-SARS-CoV-2 immune response regulated by this region. First, we demonstrated that OAS1 is important for the clearance of SARS-CoV-2 infection (Figure 2). Next, we focused on exploring how associated genetic variants regulate mRNA expression of OAS1. Then, by employing several methods (Figures 3 and 4), we demonstrated that genetic variants associated with COVID-19 severity in patients of European and African ancestries have direct functional effects on the expression of OAS1. Specifically, we demonstrated that the risk haplotype functionally causes downregulation of OAS1 expression, while the most protective haplotype, which happened to be ancestral/Neandertal, provides several layers of protection by different mechanisms resulting in higher levels of OAS1 expression.

7. In general I do not understand the framework of impaired spontaneous clearance of SARS-CoV-2, considering that it does not seem to exhibit chronic infections like other viruses (such as HCV and HIV). What is impaired spontaneous clearance? This is especially unclear as these are evaluated in mild cases where there seems to be no consequences for possibly taking longer to clear the virus.

Response: Although we used the term "spontaneous clearance" to differentiate natural from treatment-induced clearance, we removed it in the current version.

8. I would hesitate to come to any conclusions based on the trial data from 58 participants, which are further split into haplotypes and treatment arms. While there was marginal significance using a P<0.05 threshold at days 5 and 10 in the placebo arm, there was no difference on days 7 and 10, which seems odd. It would also be more appropriate to conduct a longitudinal analysis looking for linear trends, instead of multiple comparisons at irregular periods of time.

Response: We performed a mixed linear model of viral loads in this clinical trial. This analysis showed the differential effect of SNPs on viral load dependent on treatment arms (ANOVA, $p=0.02$). Based on this, we analyzed the data separately in the placebo and treatment arms and observed significant associations only in the placebo group. The strongest effect was for rs1131454, with the risk allele rs1131454-A associated with impaired SARS-CoV-2 clearance.

Although this is a relatively small clinical trial, we didn't have access to any other datasets with consent for genetic testing and longitudinal measurement of viral load starting immediately from diagnosis. We have nuanced the discussion in response to this key point and offer the findings as preliminary.

9. The last sentence of the manuscript is not supported by the evidence within the manuscript. It is unclear what deficient spontaneous clearance is with SARS-CoV-2 as this is never defined and all patients had a mild disease course. If individuals with these "risk" OAS1 haplotypes have a mild disease course, then why do they need treatment? How is the effect of interferon treatment related to these specific haplotypes? And why should these be further explored in clinical trials when their relevance has not been established by the evidence base in this manuscript. These conclusions would be much better supported by showing these effects in severe cases, since the associations in the first section (where the fine-mapping occurred) were comparing severe to mild cases.

Response: Unfortunately, we could not access datasets of patients with severe COVID-19 with consent for genetic testing and extensive longitudinal data on viral load. We do not assume that COVID-19 severity is determined entirely by the OAS1 genetic variants we studied. However, our results show that these genetic variants modulate expression levels of OAS1 protein and likely contribute to the efficiency of viral clearance. We hypothesize that slower viral clearance in the presence of OAS1 risk haplotype increases the risk of disease to progress to severe stage. Accelerating this clearance therapeutically at an early stage would be beneficial.

We demonstrated the relevance of treatment with interferons by their anti-SARS-CoV-2 potency *in vitro* and more efficient viral clearance in the clinical trial, suggesting that treatment overcomes the genetic deficiency in the immune response.

In other words, individuals with the protective OAS1 haplotype are more likely to more efficiently clear the infection on their own, and thus have a lower risk of requiring hospitalization. Treatment with interferons facilitates viral clearance in all individuals but could be particularly important for those who lack the protective haplotype.

Reviewer #2:

Remarks to the Author:

The key findings for me are:

- Splice variant rs10774671 - lead variant in the analysis of severe disease reported by the HGI - is associated with risk of hospitalization among infected individuals (all hospitalized [n=954] vs. not-hospitalized [n=601]). Figure 1a

Response: We show that rs10774671 is one of the functional variants in this region. Several variants associated with COVID-19 hospitalization among infected individuals of European and African ancestries comprise a protective ancestral/Neandertal haplotype, which we explored.

- When a lung cell line was infected with SARS-CoV-2, there were no differences in attenuation of viral load between OAS1 proteins carrying different missense variants that are in high LD with rs10774671.

Response: Because our OAS1 plasmids also include rs10774671 alleles that determine OAS1-p42 and p46 protein isoforms, our analysis concluded that once produced at similar amounts, OAS1 isoforms with three missense OAS1 variants and rs10774671 do not affect attenuation of viral load. This led us to explore the effects of several associated variants on expression levels of corresponding OAS1 isoforms.

The authors then present results from an additional set of functional experiments and analysis of a small clinical trial, but I find these less convincing/relevant to help understand the association between the OAS1 locus and severe COVID-19.

Line 143, please indicate here how association analyses were run (e.g. logistic regression in R, with age, sex, etc. included as covariates), and provide metrics of genomic inflation (i.e. lambda and LD-score regression intercept; based on genome-wide results).

Response: In Figure S1, we provide a flowchart outlining our QC and analyses. In Figure S2 and below, we provide genomic inflation factor values (λ) in population-specific datasets using 511,229 genome-wide array-genotyped markers. The values were: A). $\lambda=1.01$ in patients of European ancestry (n=2249) and B). $\lambda=1.00$ in patients of African ancestry (n=835).

We did not perform the LD score regression analysis. According to the LD Score software recommendations, this analysis requires at least 4000 samples, while our largest set of patients of European ancestry included 2249 samples. The analysis requires at least 200K markers, and we had only

115,368 markers shared between our dataset and the precalculated reference panel of LD scores in European populations of the 1000 Genomes Project provided by the software. In this regard, we intend to address this in a later publication when we have a sufficient number of cases to evaluate these key outcomes.

Line 146, please add sample size to each group (hospitalized and non-hospitalized).

Response: We provided this information in all relevant places in the text.

Posterior probabilities for individual variants from SuSIE are very low (Figure 1b), with a max of 0.2. For example, rs10774671 has a PIP very close to zero. So how confident are you that your fine mapping analysis is adequately identifying the likely causal variants at this locus?

Response: We removed the credible set analysis from the current version due to the limited sample set and high LD in the region. Instead, we performed comparative analyses of this region in patients of European and African ancestries.

If the signal (i.e. lead variant) at this locus is the same between your analysis of hospitalized (n=954) vs. not-hospitalized (n=601) and the A2 analysis of the HGI, then is it not more accurate to use the HGI results to fine-map this region, given the much larger sample size? Have you tried to fine map the HGI results? Does your analysis really improve (i.e. narrow) the credible set from the HGI?

Response: Please see response to Comment 1 of Reviewer 1.

Related to this, it is noteworthy that rs10774671 is not associated with hosp vs. not-hosp (B1 phenotype) in the HGI (P = 0.8), with almost 6K cases and 15K controls. Why do you think that this is the case?

Response: Please see response to Comment 1 of Reviewer 1.

Are there any missense variants in high LD with rs10774671? If so, please indicate in text. [I see now that this is mentioned in line 180 – I would move this up a few paragraphs.]

Response: Updated

Analysis of the severe hospitalized (n=566) vs. not-hospitalized (n=601) phenotype identified an additional set of 24 variants with $0.005 < P < 0.05$. As these variants are in modest LD with rs10774671, is it not possible that the weaker signal with these additional variants is fully explained by rs10774671? Comparing Figure 1a with 1c suggests that the signals in the region are very similar. The authors should repeat the association analysis of severe hosp vs. not hospitalized with rs10774671 included as a covariate (and make an additional locuszoom plot with those results). Looking at Figure 1e, it's possible that rs1131454 is driving the two significant associations shown in the bottom part of this figure, but both might disappear when conditioning on rs10774671. Please also add frequency of each haplotype to this figure.

Response: We provide this information in Figure S3 and Table S2. In the European ancestry set, conditioning on any of these variants strongly attenuated or eliminated the signal, suggesting that associations are entirely or nearly entirely attributable to any of these variants or their combination.

Line 164, “we selected four directly genotyped variants spanning a 4.2-kb region (rs10774671, rs1131476, rs2660, and rs4766664)”. If there are only two common haplotypes underlying this signal (which makes sense given that all the associated variants are in very high LD), why do we need multiple variants (i.e. a haplotype) to define the signal in this region? Why is this useful? And why did you select four variants, not more or not less? Why not just use rs10774671?

Response: Indeed, in patients of European ancestry, the association in this region is captured by rs10774671, as well as 80 other markers that belong to the same haplotypes. However, only 5 of the 80 markers shared between patients of European and African ancestries were associated with COVID-19 severity in individuals of African ancestry; notably, the splicing SNP rs10774671 was not included in this set ($p=0.079$). By analyses in both ancestries, we demonstrated that a specific haplotype, rather than alleles of individual variants, is associated with the outcome. Thus, focusing only on rs10774671 would be insufficient to resolve the genetic and functional associations in this region.

We selected the four variants due to their association in both ancestries and representation of the ancestral/Neandertal haplotypes.

Line 169, three (of the 24) variants identified in the severe hospitalized vs. not-hospitalized analysis were then added to the four variants included in the common risk haplotype. If the association with these 3 variants is explained by LD with rs10774671, I don't think this analysis is particularly informative.

Response: This section was removed from the current version

Line 173 to 174, "In smaller sets of patients of non-European ancestries (483 African, 166 admixed-Hispanic, and 103 East-Asian), none of these variants or their haplotypes showed significant associations". I suggest focusing the text on results for rs10774671, explicitly reporting the allele frequency and association results in the three different groups.

Response: Our updated analysis showed the association of the same alleles/haplotypes in patients of both European and African ancestries. We provide the results for individual markers and their haplotypes in patients of European and African ancestries.

As discussed above, focusing only on rs10774671 is insufficient to explain genetic and functional associations in this region. In this regard, our analyses in 835 patients of African ancestry are specifically informative. In this population, rs10774671 was associated with hospitalized vs. non-hospitalized disease only with $p=0.079$. In Europeans, the G and A alleles of rs10774671 correspond to the non-risk and risk haplotypes. In contrast, in patients of African ancestry, rs10774671-G is most protective in the context of the specific ancestral haplotype shared with Neandertals and common in Europeans. At the same time, the rs10774671-G was not sufficiently associated when included in another, common African-specific haplotype. Thus, the protection from COVID-19 hospitalization is conferred by a haplotype rather than by the rs10774671-G allele alone.

This is the first analysis of the OAS1 region, not just rs10774671, in individuals of African ancestry. Previously, it was discussed that the Neandertal haplotype introgression was beneficial because it increased the frequency of the functional rs10774671-G allele. Our results suggest a specific role for an ancestral haplotype that includes rs10774671-G, in which the contribution of rs10774671 is necessary but not sufficient to explain the effect of this haplotype, at least in severity of SARS-CoV-2 infection.

Also, it would be helpful to indicate power to detect a significant association with the European odds ratio in each of these smaller groups.

Response: We used the Genetic Association Study Power Calculator (csg.sph.umich.edu/abecasis/gas_power_calculator/).

We calculated the power to detect an association between hospitalized and non-hospitalized COVID-19 for rs1131454 and rs10774671, assuming an additive genetic model, significance level of $p=0.001$, a disease prevalence of 0.5, and an OR=1.2. In individuals of European ancestry, with a sample size of $n=2249$ (1035 non-hospitalized and 1214 hospitalized), the statistical power was estimated as 98.2% for rs1131454 (risk allele frequency (RAF)=0.57) and 96.6% for rs10774671 (RAF=0.64). In individuals of African ancestry, with a sample size of $n=835$ (324 non-hospitalized and 511 hospitalized), the statistical power was estimated as 87.8% for rs1131454 (RAF=0.25) and 91.5% for rs10774671 (RAF=0.42).

Line 180, "Our analyses suggested that in Europeans at least one effect variant for the risk of hospitalized disease is included in the 95% credible set of 76 OAS1 variants." This could be stated more clearly. I think what you mean is that the fine mapping analysis of your hosp vs. not hosp results identified a credible set of 76 variants, which is likely to include the (or at least one) causal variant underlying the association between this locus and the A2 phenotype reported by the HGI.

Response: This section is edited out in the current version

Line 182, "included four missense variants", helpful to indicate that these were missense variants for OAS1.

Response: Updated

Line 221 to 246. This section focuses predominantly on missense variant rs1131454, but this variant is not part of the credible set for the common risk signal in this region. Instead, it has weak associations with the hosp severe vs. not-hosp phenotype, which could potentially be explained by the modest LD with rs10774671. So not clear how relevant these functional data are to understand the common risk signal in this region.

Response: We focused on the signal for hospitalized vs. non-hospitalized COVID-19. As demonstrated by our analyses, the association for rs1131454 was detected both in patients of European and African ancestries and is either comparable or stronger than for rs10774671. The results of a meta-analysis of 3084 patients of European and African ancestries showed comparable results for these markers: OR=1.30, $p=1.87E-04$ for rs1131454, and OR=1.29, $p=1.48E-04$ for rs10774671.

We demonstrated that both these variants belong to the ancestral/Neandertal haplotype and functionally contribute to the molecular phenotype of this association, which is related to reduced *OAS1* expression leading to impaired viral clearance. The function of rs1131454 is an essential part of this molecular phenotype.

Line 288 to 303. Very small sample size for the genetic analysis in the clinical trial data (n=28 and n=30, respectively in placebo and treatment group). The weak association between the rs10774671/rs1131454 haplotype and viral load in the placebo group is not very convincing, it would not remain significant after correcting for multiple testing. I also think it would make more sense to just analyze rs10774671 per se, given my reservations with the association with rs1131454, as mentioned above. Not clear why rs2660 is included in this analysis (Figure 7d), as it does seem informative (high LD with rs10774671).

Response: We performed linear mixed model analysis for longitudinal measures of viral load. Since this analysis showed the differential effect of SNPs on viral load dependent on treatment arms (ANOVA, p=0.02), we analyzed the data separately in the placebo and treatment arms and observed significant associations only in the placebo group. Because our genetic analysis indicated the predominant associations due to haplotypes rather than individual markers, we used this approach for analysis of viral clearance as well. We analyzed variants rs1131454, rs10774671 and rs2660 that capture *OAS1* haplotypes both in patients of all ancestries.

Line 702. "Additionally, whole-genome sequencing data generated by contributing studies were used for a subset of samples." What proportion of samples had available whole-genome sequencing data and how were these data used?

Response: We provide this information in Figure S1. There were 238 WGS samples, all for patients of European ancestry (Italy). WGS-genotyped samples were processed together with array-genotyped samples through all the QC and analyses stages to determine ancestry, relatedness, PCs and then used for association analyses. Of those, 215 samples (9.5% of all European samples, n=2249) were retained for the final analysis. Using WGS as a covariate, we did not observe any differences in the association results in Europeans and mentioned this in the Materials and Methods. Below, we provide specific results for several markers of interest.

Without WGS as a covariate					With WGS as a covariate				
marker	OR	L95	U95	p-val	marker	OR	L95	U95	p-val
rs1131454	1.27	1.09	1.49	2.88E-03	rs1131454	1.26	1.08	1.48	3.95E-03
rs10774671	1.33	1.13	1.56	6.45E-04	rs10774671	1.32	1.12	1.56	7.89E-04
rs1131476	1.30	1.11	1.53	1.42E-03	rs1131476	1.30	1.11	1.54	1.56E-03
rs1051042	1.30	1.11	1.53	1.42E-03	rs1051042	1.30	1.11	1.54	1.56E-03
rs2660	1.30	1.11	1.53	1.42E-03	rs2660	1.30	1.11	1.54	1.56E-03
rs4766664	1.32	1.12	1.55	9.04E-04	rs4766664	1.31	1.12	1.55	1.14E-03

Line 703. "Sex, relatedness and population groups (European, African, admixed-Hispanic and East-Asian)

were determined based on genetic analyses for all patients." Please provide more detail. How exactly were samples assigned to continental ancestries?

Response: Figure S1 outlines our QC and analytical pipeline. Specifically, we assigned ancestral status using GRAF (Genetic Relationship And Fingerprinting) software. This software assumes that ancestry is contributed by three ancestral proportions (European [Pe], African [Pf], and Asian [Pa]). Based on genetic distances (GD1, GD2, GD3, and GD4) calculated from each subject to several reference populations, the ancestry of each individual is estimated using cutoff values based on these distance scores as shown in Tables below copied from <https://github.com/ncbi/graf>

Grouping subjects based on the ancestry proportions

PopID	Population	Cutoff standard
1	European	$P_e \geq 87\%$
2	African	$P_f \geq 95\%$
3	East Asian	$P_a \geq 95\%$
4	African American	$40\% \leq P_f < 95\%$ and $P_a < 13\%$
5	Hispanic 1	$P_f < 40\%$ and $P_e < 87\%$ and $P_a < 13\%$ and $P_f \geq P_a$
6,7,8	(Three populations)	$P_a < 95\%$ and $P_e < 87\%$ and $P_f < 13\%$ and $P_f < P_a$
9	Other	$P_a \geq 13\%$ and $P_f \geq 13\%$

Separating Asians and Hispanics using GD1 and GD4 scores

PopID	Population	Cutoff standard
7	Asian-Pacific Islander	$GD1 > 30 \times (GD4)^2 + 1.73$
8	South Asian	$GD4 > 5 \times (GD1 - 1.69)^2 + 0.042$
6	Hispanic 2	$GD4 < 0$ and PopID is not 7

Lines 708 to 714. Did you only impute chr 12? I think this is highly unlikely. Very important to have association results for the full genome, to be able to assess if associations statistics are well behaved (based on lambdas, etc.). In the unlikely event that imputed data is not available for the whole genome, then this should be performed with the array data.

Response: Because the project is still in development, we currently imputed only markers from the whole chromosome 12 to focus on this specific signal. In Figure S2 and below, we provide Lambdas (λ) in population-specific datasets using 511,229 genome-wide array-genotyped markers. The Lambdas were: A). 1.01 in patients of European ancestry (n=2249) and B). 1.00 in patients of African ancestry (n=835).

Based on these values, we believe our results in patients of European and African ancestries are well-behaved.

Reviewer #3:

Remarks to the Author:

The human genomic locus chr12q24.13 encoding 2',5'-oligoadenylate synthetases (OASs) 1-to-3 has been previously associated with severity of COVID-19. In this study of 1555 Covid-19 cases of European ancestry, a single haplotype of 76 OAS1 variants, included in a genomic fragment introgressed from Neandertals, was associated with risk of hospitalization of COVID-19 patients, and with spontaneous but not treatment induced SARS-CoV-2 clearance with pegIFN-lambda1. Two exonic variants examined affected splicing and nonsense-mediated decay (NMD) of OAS1 mRNAs. Data support that it is not transcriptional control that is responsible for higher levels of OAS1-p46, but rather control of mRNA stability through NMD. OAS1-p46-long exon 3 lacked premature termination codons, in contrast to other OAS1 transcripts. SiRNA knockdowns of NMD components convincingly demonstrate that the OAS1-p46 long isoform was resistant to NMD. Further results suggest that the OAS1 risk haplotypes with impaired spontaneous clearance of SARS-CoV-2 could be compensated by early treatment with IFN-lambda. In other words, pegIFN-lambda1 overcame the deficiency of OAS1 in the high-risk group.

This paper advances knowledge of how alternative splicing controls levels of OAS1 proteins. NMD mRNA decay reduces expression of the p42 OAS1 isoform, which is probably responsible for the relatively higher levels of the alternative spliced p46 OAS1 isoform. It is not transcription control, or differences in specific activities of p42 vs. p46 OAS1, but rather the higher levels of mRNA encoding p46 that sets protein levels, since the mRNA for this isoform is relatively stable.

The following point should be addressed in a revised manuscript. In Figure 2c, the effects of expressing the risk (OAS1-p42) and non-risk (OAS1-p46) isoforms on SARS-CoV-2 yields in A549-ACE2 cells were determined. The two OAS1 isoforms showed similar antiviral activity. In contrast, a recent study reported that the prenylated and endomembrane targeted OAS1-p46 isoform had greater antiviral

activity than the p42 OAS1 isoform (ref. 21, this reference should be updated as published in eLife: PMID: 34342578). However, the experiment in Figure 2c requires Western blots for p42 and p46 to monitor relative expression the different OAS1 isoforms (with antibody to the Flag epitope-tag), controlled for beta-actin levels. Are expressed levels of p42 and p46 similar, or, if different, do levels of these proteins correlate with the antiviral effects?

Response: We thank the reviewer for the perfect synopsis of our work and the important suggestion to demonstrate similar protein expression of all OAS1-Flag isoforms. By Western blotting, we confirmed that all OAS1-Flag protein isoforms were expressed at similar levels and added this information as an updated Figure 2d and in the text. We updated the references to previous preprints by the corresponding published papers.

Decision Letter, second revision:

8th March 2022

Dear Mila,

Your revised Article "Genetic regulation of OAS1 nonsense-mediated decay underlies association with COVID-19 hospitalization in patients of European and African ancestries" has been seen by the original referees. You will see from their comments below that, while they find the manuscript improved, they have a few ongoing concerns regarding aspects of the presentation and interpretation of the genetic association results. We remain interested in the possibility of publishing your study in Nature Genetics, but we would like to consider your response to these remaining concerns in the form of a further revision before we make a final decision on publication.

To guide the scope of the revisions, the editors discuss the referee reports in detail within the team, including with the chief editor, with a view to identifying key priorities that should be addressed in revision, and sometimes overruling referee requests that are deemed beyond the scope of the current study. In this case, we particularly ask that you provide more details regarding the association patterns observed among African ancestry study participants and that you revise the presentation and interpretation of the association analyses taking into account the referees' comments regarding significance thresholds, effect sizes, and sample sizes. We hope you will find this prioritized set of referee points to be useful when revising your study. Please do not hesitate to get in touch if you would like to discuss these issues further.

We therefore invite you to revise your manuscript taking into account all reviewer and editor comments. Please highlight all changes in the manuscript text file. At this stage we will need you to upload a copy of the manuscript in MS Word .docx or similar editable format.

*2) If you have not done so already please begin to revise your manuscript so that it conforms to our Article format instructions, available [here](http://www.nature.com/ng/authors/article_types/index.html). Refer also to any guidelines provided in this letter.

*3) Include a revised version of any required Reporting Summary:

[Redacted]

We hope to receive your revised manuscript within 4-8 weeks. If you cannot send it within this time, please let us know.

Sincerely,
Kyle

Kyle Vogan, PhD
Senior Editor
Nature Genetics
<https://orcid.org/0000-0001-9565-9665>

Referee expertise:

Referee #1: Genetics, infectious diseases

Referee #2: Genetics, inflammatory diseases

Referee #3: Innate immunity, viral infections

Reviewers' Comments:

Reviewer #1:

Remarks to the Author:

Thank you for the opportunity to review this resubmission. The authors have addressed many of my concerns, but I have a few more general comments and a few minor comments remaining.

- A general comment is that the authors use the terminologies of associations being weaker or stronger as determined by their P-value, not by the effect estimates. This can be problematic, given the differences in sample size. For example, page 6 lines 159-162 states that the African haplotype has a weaker association with an OR. However, one of the European haplotypes have the exact same effect size (OR=0.76), it's just that the p-value is smaller, probably due to the sample size lending more statistical power and precision in estimates. This language can muddy the interpretation, leading the authors to assume that other variants may be linked in different populations, instead of the possibility that their sample sizes just aren't comparable.

- Overall, it would be helpful to have some additional language to link the sections together. This can only be a sentence or two, but would help link them all together instead of the disjointed feel of the current structure. This will also help readers bring it all together at the end.

- The African ancestry samples appear to be largely African American or admixed individuals. It would be helpful to differentiate if the haplotype is from a European background in these individuals or on the African tracts for some of the conclusions. It is difficult to know if the African ancestry analyses are actually looking at haplotypes of African ancestry or of European tracts within admixed African-European populations.

- Given the haplotype analyses and the comparisons between European and African ancestry, it would be beneficial to know if the alleles/haplotypes of interest are found exclusively on European haplotypes as a Neanderthal introgression or if they are on African haplotypes. Local ancestry analyses may be useful for these analyses, but it is understandable if this is not possible due to the candidate region approach of the paper. This would be especially instrumental in explaining the very low LD between the lead variants and those used in various analyses as determined by r^2 .

- Please add sample size numbers for the comparisons made in the last section, specifically page 12, lines 284-291. It is hard to adequately gauge the evidence in such a small number without having the numbers in the main text. For example, how many had the AAA haplotype? This information should not only be in the supplement.

- Why wasn't the association between rs4767027, the intronic OAS1 variant, and outcome in your dataset? In previous datasets? It couldn't be imputed? It would be helpful to have that link for the section beginning on page 8, to link the previous genetic association work with the expression evidence in this section. (Also for the reader, it would be helpful to remind us that you show the

results of rs10774671, which is highly linked with this variant. Just to make the connection between it all.)

- How do you determine significantly associated? For example, for African ancestry, the variants associated with outcome all have p-values >0.01 . How do you account for multiple comparisons? While I agree that Bonferroni would be too conservative, did you use an LD-adjusted threshold?

- Add in minor allele frequencies within the study for Table S1 with stratified results.

- Page 5, lines 132-136 state that the association was weaker in African ancestry participants, but the effect sizes are largely the same (OR=0.92 vs 0.93). Some language differentiating between the effect sizes, which seem consistent, and the p-values, which are more likely due to power, would be helpful.

Reviewer #2:

Remarks to the Author:

While some of my questions were adequately addressed, the author's response to other questions were not particularly helpful or clear. It is also not helpful to simply point me to a response provided to another reviewer - in fact, in more than one instance I could not find an appropriate response to my specific questions elsewhere.

Despite this, the manuscript is well written and much improved. However, I remain unconvinced by some of the data that the authors use to support some of the conclusions of this manuscript.

For example, on lines 156 to 164, there's an informal comparison of the effect on risk of hospitalization between the GGGT haplotype (OR=0.46, $P=0.0029$) and the GGAG haplotype (OR=0.76, $P=0.059$) in individuals of African ancestry. Based on the weaker effect of the latter haplotype, the authors suggest that there are other genetic variants that modify the association and function of the protective Neandertal haplotype (tagged by rs10774671-G), and then this theme is carried forward to other sections of the paper. Are these two effect sizes really significantly different? The sample size used for these haplotype analyses is too small to provide this level of resolution.

Similarly, as I mentioned previously, the association between haplotypes and viral loss in the clinical trial data is based on such a small sample size that it is not clear to me if the association seen in the placebo group is a false-positive association (not corrected for multiple testing) that does not replicate in the treatment arm or, as the authors have interpreted, evidence that "pegIFN- λ 1 treatment overcame the deficit in OAS1 expression associated with the AAA haplotype".

Reviewer #3:

Remarks to the Author:

1. Lines 175-7: "To control for endogenous interferon responses to the plasmids, we transfected cells with either a control GFP plasmid or individual OAS1 plasmids and, after 48 hours, infected cells with SARS-CoV-2 for 24 hours (Figure 2b)". But according to figure 2b transfection was at 24 h and

infection at 74 h (50 hrs) and SARS-CoV-2 qRT-PCR was at 86 h (12 hpi, not 24 hpi). Either the text or the figure needs to be corrected.

2. In the revised manuscript, the lack of difference between p42 and p46 on viral RNA levels was not due to differences in expression as shown in the newly provided Western blot, figure 2d. The antiviral effect of p42 and p46 isoforms of OAS1 was equivalent (fig.2c). Authors should indicate that the antiviral effect of p42 or p46 versus GFP control was significant (this information is in Table S5, but should also be in figure 2c).

Author Rebuttal, second revision:

Response to referees

We thank the reviewers for their constructive comments that helped us to further refine our paper. Below, we provide a summary of the changes and point-by-point responses.

A summary of revisions we are providing:

1. Included data for OAS1-rs4767027 and incorporated this variant in all parts of the paper.
2. Based on this, we included a new Fig.S5 (shown below), which shows the structure of *OAS1* haplotypes in relation to ancestral status and association with COVID-19 severity.
3. Updated the emphasis of the paper, now focusing on the risk alleles/haplotype shared in COVID-19 patients of European and African ancestries. We posit that the emergence of OAS1 derived human-specific alleles rs1131454-A and rs10774671-A increased the risk of severe COVID-19 in humans, even though these alleles might have been beneficial under unknown/non-infectious conditions and became the major alleles in Europeans and Asians. Non-human species do not develop severe SARS-CoV-2 infection, which might be partially explained by the absence of these human-specific risk variants that decrease OAS1 expression.
4. Avoided referring to the strength of association based on p-values and referred to ORs instead
5. Removed comparisons of ORs that are not statistically significant
6. Added Effect Allele Frequencies (EAFs) in hospitalized, non-hospitalized and all patients to Table S1
7. Added LD adjusted p-values. New Fig. S4 (shown below) shows linkage disequilibrium blocks within the chr12q24.13 region in COVNET patients of European and African ancestries. The counts of ancestry-specific LD blocks - 4 LD blocks in European and 11 LD blocks in African lineages - were used to adjust p-values.
8. Updated Figure 2 to make the infection timeline clear and added p-values for comparison with empty vector (GFP).
9. Discussed preliminary results of Phase 3 clinical trial (TOGETHER trial, NCT04727424, 1936 outpatient COVID-19 patients treated with pegIFN- λ 1), which showed a significant reduction of COVID-19 related hospitalization and death compared to placebo.

New Figure S4:

Figure S4. Linkage disequilibrium blocks within the chr12q24.13 region in COVNET patients of European and African ancestries.

Analysis of the 113 Kb region (hg38:112,904,114-113,017,173, n=81 SNPs) at chr12q24.13 with Solid Spine method (Haploview 4.2) identified 4 linkage disequilibrium (LD, D') blocks in COVNET COVID-19 patients of European ancestry **a**) and 11 LD blocks in patients of African ancestry **b**) Dark red shading denotes $D' > 0.80$.

New Figure S5:

a

b

Figure S5. Structure of *OAS1* haplotypes in relation to ancestral status and association with COVID-19 severity.

a) Analysis of the *OAS1* haplotypes (14.13 Kb, hg38:112,911,065-112,925,192) comprised of 7 markers. The color-coding indicates the ancestral status of specific alleles – human (ancestral or derived), archaic humans (Neandertal or Denisova lineages), and COVID-19 severity status (Non-risk/Risk). Haplotype

frequencies are shown for hospitalized patients with COVID-19 of European and African ancestry from COVNET; NA-haplotype is not detected. Odds ratios (ORs) and 95% confidence intervals (95% CIs) are for comparison with the common Risk haplotype (also marked as ref); full details can be found in **Table S4**. The Non-risk haplotypes differ from the Risk haplotype by the alleles of rs1131454 and rs10774671. The COVID-19 risk is associated with human-specific derived alleles rs1131454-A and rs10774671-A. Additionally, the Risk haplotype includes a Denisova – type fragment of derived alleles spanning rs1131476 to rs4766664. Human polymorphisms rs1131454, rs1131476, rs1051042, and rs2660 were also explored and found monomorphic in genomic sequences of 29 chimpanzees: Pan troglodytes verus (n=24) and Pan troglodytes troglodytes (n=5). Source⁸: European Nucleotide Archive <https://www.ebi.ac.uk>, accession numbers FM163403.1–FM163432.1 **b**) OAS1 haplotypes and phylogenetic tree. COVID-19 risk alleles rs1131454-A and rs10774671-A are human-specific and derived.

Reviewers' Comments:

Reviewer #1:

Remarks to the Author:

Thank you for the opportunity to review this resubmission. The authors have addressed many of my concerns, but I have a few more general comments and a few minor comments remaining.

- A general comment is that the authors use the terminologies of associations being weaker or stronger as determined by their P-value, not by the effect estimates. This can be problematic, given the differences in sample size. For example, page 6 lines 159-162 states that the African haplotype has a weaker association with an OR. However, one of the European haplotypes have the exact same effect size (OR=0.76), it's just that the p-value is smaller, probably due to the sample size lending more statistical power and precision in estimates. This language can muddy the interpretation, leading the authors to assume that other variants may be linked in different populations, instead of the possibility that their sample sizes just aren't comparable.

Response: We agree with the reviewer and revised the text accordingly. Specifically:

- We have removed comparisons based on p-values alone and carefully compared ORs when appropriate with suitable nuance
- We state that ORs between the top 7 variants were comparable both in HGI and in our analyses

- Overall, it would be helpful to have some additional language to link the sections together. This can only be a sentence or two but would help link them all together instead of the disjointed feel of the current structure. This will also help readers bring it all together at the end.

Response: We are constrained by word limit but edited/added text at several places to improve the flow of the paper.

- The African ancestry samples appear to be largely African American or admixed individuals. It would be helpful to differentiate if the haplotype is from a European background in these individuals or on the African tracts for some of the conclusions. It is difficult to know if the

African ancestry analyses are actually looking at haplotypes of African ancestry or of European tracts within admixed African-European populations.

Response: Our analyses suggest that the *OAS1* haplotype protective from severe COVID-19 in both ancestries was likely acquired through admixture with European populations. We indicate this in several places – Table S1, Results, Discussion and new Figure S5 (shown above). We intentionally excluded a particularly small group of 48 individuals defined as of primarily African ancestry ($P_f > 95\%$) by GRAF software and analyzed 835 individuals defined as admixed African ancestry. Another protective haplotype (not statistically distinct from the shared haplotype, $p = 0.19$), which we describe in Table S4 and new Figure S5 is African-specific.

Discussion: Surprisingly, the ancestral haplotype that includes three missense *OAS1* variants is absent in African populations within the 1000 Genomes data set but present in individuals of African American ancestry (AWS in 1000 Genomes and our COVID-19 patients of African ancestry, **Table S4**), most likely due to admixture with non-African populations.

- Given the haplotype analyses and the comparisons between European and African ancestry, it would be beneficial to know if the alleles/haplotypes of interest are found exclusively on European haplotypes as a Neanderthal introgression or if they are on African haplotypes. Local ancestry analyses may be useful for these analyses, but it is understandable if this is not possible due to the candidate region approach of the paper. This would be especially instrumental in explaining the very low LD between the lead variants and those used in various analyses as determined by r^2 .

Response: By comparing African (YRI) and African-American (ASW) populations in the 1000 Genomes (Table S1), we showed that the common protective *OAS1* haplotype is European-derived, which is consistent with its Neandertal origin. Although this is also an ancestral haplotype, it is absent in the modern African populations. We added a new Figure S5 (shown above) that explores the ancestral and Non-Risk/Risk status of the *OAS1* haplotype. This analysis led us to conclude that the emergence of human-specific derived alleles rs1131454-A and rs10774671-A can differentiate the risk and non-risk haplotypes in both populations. In contrast, the non-risk alleles of both markers are human ancestral and also shared with both lineages of archaic humans (Neandertal and Denisova). Since all these lineages share the protective alleles of these markers, we avoided designating the protective properties solely to Neandertal lineage. Instead, we attributed the association to the human-specific gain of derived risk alleles. Thus, the risk alleles/haplotypes are shared by both ancestries, one non-risk haplotype is European-derived and is shared by both ancestries and another non-risk haplotype is African-specific.

The ancestry was analyzed with GRAF software, with the details presented in Figure S1. We focused our analysis on 835 individuals of African ancestry ($P_f = 40-95\%$), and excluded a small group of individuals of primarily African ancestry ($n = 48$, $P_f > 95\%$).

- Please add sample size numbers for the comparisons made in the last section, specifically page 12, lines 284-291. It is hard to adequately gauge the evidence in such a small number without

having the numbers in the main text. For example, how many had the AAA haplotype? This information should not only be in the supplement.

Response: We added the requested information to the main text.

- Why wasn't the association between rs4767027, the intronic OAS1 variant, and outcome in your dataset? In previous datasets? It couldn't be imputed? It would be helpful to have that link for the section beginning on page 8, to link the previous genetic association work with the expression evidence in this section. (Also for the reader, it would be helpful to remind us that you show the results of rs10774671, which is highly linked with this variant. Just to make the connection between it all.)

Response: Thank you for raising this issue. Initially, we did not include this variant due to its lower imputation score ($r^2=0.67$) in African ancestry (compared to other markers with $r^2>0.88$ and threshold of $r^2=0.8$). However, we now included the data for this SNP in Table S1, Figure 1c, new Figure S5, Results and Discussion.

In Europeans, this SNP is associated with COVID-19 severity similarly to other linked variants (including rs10774671) and showed an association in Africans (but due to lower imputation quality, this result should be taken with caution). This variant was specifically informative for understanding the ancestral status of this haplotype. We included a new Figure S5 (shown above), which summarizes this information based on the top 7 SNPs. Based on this, we adjusted our narrative to focus on risk alleles/haplotypes, the derived human-specific OAS1 alleles rs1131454-A and rs10774671-A.

- How do you determine significantly associated? For example, for African ancestry, the variants associated with outcome all have p-values >0.01 . How do you account for multiple comparisons? While I agree that Bonferroni would be too conservative, did you use an LD-adjusted threshold?

Response: Thank you for the suggestion, which we implemented and added as a new Figure S4 (shown above). As was suggested in the paper by Duggal et al (PMID:18976480), we used the Solid Spine method (Haploview 4.2), and identified 4 LD blocks in COVNET patients of European and 11 LD blocks in patients of African ancestry and used these values to adjust our p-values. We provide adjusted p-values in Table S1. Furthermore, we acknowledge the exploratory nature of our association analyses in this region but please note that the genetic association data together with the biological/functional data evidence provide strong evidence for the importance of this region. In this regard, it is also the only study that evaluated the entire region - and not a single selected variant (to avoid multiple testing) - in individuals of African ancestry.

- Add in minor allele frequencies within the study for Table S1 with stratified results.

Response: We added Effect Allele Frequencies (EAFs) in hospitalized, non-hospitalized, and the whole set of patients to Table S1.

- Page 5, lines 132-136 state that the association was weaker in African ancestry participants, but the effect sizes are largely the same (OR=0.92 vs 0.93). Some language differentiating between

the effect sizes, which seem consistent, and the p-values, which are more likely due to power, would be helpful.

Response: We avoided comparing p-values directly but instead have carefully commented on the ORs, which, as suggested, could be viewed as more biologically plausible.

Reviewer #2:

Remarks to the Author:

While some of my questions were adequately addressed, the author's response to other questions were not particularly helpful or clear. It is also not helpful to simply point me to a response provided to another reviewer - in fact, in more than one instance I could not find an appropriate response to my specific questions elsewhere.

Response: We apologize for this inconvenience and now provide responses to each question of the reviewers

Despite this, the manuscript is well written and much improved. However, I remain unconvinced by some of the data that the authors use to support some of the conclusions of this manuscript.

For example, on lines 156 to 164, there's an informal comparison of the effect on risk of hospitalization between the GGGT haplotype (OR=0.46, P=0.0029) and the GGAG haplotype (OR=0.76, P=0.059) in individuals of African ancestry. Based on the weaker effect of the latter haplotype, the authors suggest that there are other genetic variants that modify the association and function of the protective Neandertal haplotype (tagged by rs10774671-G), and then this theme is carried forward to other sections of the paper. Are these two effect sizes really significantly different? The sample size used for these haplotype analyses is too small to provide this level of resolution.

Response: Indeed, the difference between the GGGT haplotype (OR=0.46, P=0.0029) and the GGAG haplotype (OR=0.76, P=0.059) in individuals of African ancestry is not statistically significant (P=0.19), which we now acknowledge as:

P.7: In patients of African ancestry, the same variants formed four haplotypes. As in Europeans, the Neandertal **GGGT** haplotype was the main haplotype, which is protective from hospitalized disease (OR=0.46, P=2.39E-03), despite being less common than in Europeans (10.2% in non-hospitalized and 5.6% in hospitalized patients, **Table S4**). The rs10774671-G allele was also included in a common African-specific haplotype (**GGAG**, OR=0.76, P=0.059, **Figure 1c, Table S4**); the ORs of these haplotypes (0.46 vs 0.76) were not significantly different (P=0.19).

Similarly, as I mentioned previously, the association between haplotypes and viral loss in the clinical trial data is based on such a small sample size that it is not clear to me if the association seen in the placebo group is a false-positive association (not corrected for multiple testing) that does not replicate in the treatment arm or, as the authors have interpreted, evidence that "pegIFN- λ 1 treatment overcame the deficit in OAS1 expression associated with the AAA haplotype".

Response: We acknowledge the limitations of our exploratory analysis in this small but distinctive clinical trial. However, we would also like to point out our *in vitro* results that show that IFN treatment before and even after SARS-CoV-2 infection clears this virus. Our conclusions are based on the cumulative results of the clinical trial and the *in vitro* work.

We also discussed preliminary results of Phase 3 clinical trial (TOGETHER trial, NCT04727424, 1936 outpatient COVID-19 patients treated with pegIFN- λ 1), which showed a significant reduction of COVID-19 related hospitalization and death compared to placebo.

Unfortunately, we will not be able to analyze the genetic data of the participants of this trial.

Reviewer #3:

Remarks to the Author:

1. Lines 175-7: To control for endogenous interferon responses to the plasmids, we transfected cells with either a control GFP plasmid or individual OAS1 plasmids and, after 48 hours, infected cells with SARS-CoV-2 for 24 hours (Figure 2b). But according to figure 2b transfection was at 24 h and infection at 74 h (50 hrs) and SARS-CoV-2 qRT-PCR was at 86 h (12 hpi, not 24 hpi). Either the text or the figure needs to be corrected.

Response: Thank you for catching this inconsistency. We have updated Fig.2b to make it clearer and consistent with the text in the paper. Specifically, we marked the transfection time as "0 h" and the time of seeding as "-24 hrs", infection time as "48 hrs" and harvesting and qRT-PCR analysis time as "72 hrs".

2. In the revised manuscript, the lack of difference between p42 and p46 on viral RNA levels was not due to differences in expression as shown in the newly provided Western blot, figure 2d. The antiviral effect of p42 and p46 isoforms of OAS1 was equivalent (fig.2c). Authors should indicate that the antiviral effect of p42 or p46 versus GFP control was significant (this information is in Table S5, but should also be in figure 2c).

Response: Thank you for this suggestion. In Fig.2c, we added $P < 0.05$ to indicate the significance between antiviral effects of OAS1-p42/p46 isoforms versus GFP. In the legend of Fig.2c, we indicated that actual P -values are provided in Table S5.

Decision Letter, third revision:

Our ref: NG-A57954R2

19th April 2022

Dear Mila,

Thank you for submitting your revised manuscript "Genetic regulation of OAS1 nonsense-mediated decay underlies association with COVID-19 hospitalization in patients of European and African ancestries" (NG-A57954R2). In light of the changes made in response to the points raised at the previous round of review, we will be happy in principle to publish your study in Nature Genetics as an Article pending final revisions to comply with our editorial and formatting guidelines.

We are now performing detailed checks on your paper and we will send you a checklist detailing our editorial and formatting requirements soon. Please do not upload the final materials and make any revisions until you receive this additional information from us.

Thank you again for your interest in Nature Genetics. Please do not hesitate to contact me if you have any questions.

Sincerely,
Kyle

Kyle Vogan, PhD
Senior Editor
Nature Genetics
<https://orcid.org/0000-0001-9565-9665>

Decision Letter, final checks:

Our ref: NG-A57954R2

3rd May 2022

Dear Mila,

Thank you for your patience as we have prepared the guidelines for final submission of your Nature Genetics manuscript "Genetic regulation of OAS1 nonsense-mediated decay underlies association with COVID-19 hospitalization in patients of European and African ancestries" (NG-A57954R2). Please carefully follow the step-by-step instructions provided in the attached file and add a response in each row of the table to indicate the changes that you have made. Ensuring that each point is addressed will help to ensure that your revised manuscript can be swiftly handed over to our production team.

If you have not done so already, please alert us to any related manuscripts from your group that are under consideration or in press at other journals, or are being written up for submission to other

journals (see: <https://www.nature.com/nature-research/editorial-policies/plagiarism#policy-on-duplicate-publication> for details).

In recognition of the time and expertise our reviewers provide to our editorial process, we would like to formally acknowledge their contribution to the external peer review of your manuscript entitled "Genetic regulation of OAS1 nonsense-mediated decay underlies association with COVID-19 hospitalization in patients of European and African ancestries". For those reviewers who give their assent, we will be publishing their names alongside the published article.

Nature Genetics offers a Transparent Peer Review option for new original research manuscripts submitted after December 1st, 2020. As part of this initiative, we encourage our authors to support increased transparency into the peer review process by agreeing to have the reviewer comments, author rebuttal letters, and editorial decision letters published as a Supplementary item. When you submit your final files please clearly state in your cover letter whether or not you would like to participate in this initiative. Please note that failure to state your preference will result in delays in accepting your manuscript for publication.

Cover suggestions

As you prepare your final files we encourage you to consider whether you have any images or illustrations that may be appropriate for use on the cover of Nature Genetics.

We accept TIFF, JPEG, PNG or PSD file formats (a layered PSD file would be ideal), and the image should be at least 300 ppi resolution (preferably 600-1200 ppi), in CMYK color mode.

Please submit your suggestions, clearly labeled, along with your final files. We will be in touch if more information is needed.

Nature Genetics has now transitioned to a unified Rights Collection system which will allow our Author Services team to quickly and easily collect the rights and permissions required to publish your work. Approximately 10 days after your paper is formally accepted, you will receive an email in providing you with a link to complete the grant of rights. If your paper is eligible for Open Access, our Author Services team will also be in touch regarding any additional information that may be required to arrange payment for your article.

Please note that Nature Genetics is a Transformative Journal (TJ). Authors may publish their research with us through the traditional subscription access route or make their paper immediately open access through payment of an article-processing charge (APC). Authors will not be required to make a final decision about access to their article until it has been accepted. [Find out more about Transformative Journals](https://www.springernature.com/gp/open-research/transformative-journals)

Authors may need to take specific actions to achieve [compliance](https://www.springernature.com/gp/open-research/funding/policy-compliance-faqs) with funder and institutional open access mandates. If your research

is supported by a funder that requires immediate open access (e.g. according to [Plan S principles](https://www.springernature.com/gp/open-research/plan-s-compliance)), then you should select the gold OA route, and we will direct you to the compliant route where possible. For authors selecting the subscription publication route, the journal's standard licensing terms will need to be accepted, including [self-archiving policies](https://www.springernature.com/gp/open-research/policies/journal-policies). Those licensing terms will supersede any other terms that the author or any third party may assert apply to any version of the manuscript.

Please use the following link to upload your final submission files:

[Redacted]

Best wishes,
Kyle

Kyle Vogan, PhD
Senior Editor
Nature Genetics
<https://orcid.org/0000-0001-9565-9665>

Final Decision Letter:

In reply please quote: NG-A57954R3 Prokunina-Olsson

26th May 2022

Dear Mila,

I am delighted to say that your manuscript "Genetic regulation of OAS1 nonsense-mediated decay underlies association with COVID-19 hospitalization in patients of European and African ancestries" has been accepted for publication in an upcoming issue of Nature Genetics.

Your paper will be published online after we receive your corrections and will appear in print in the next available issue. You can find out your date of online publication by contacting the Nature Press Office (press@nature.com) after sending your e-proof corrections. Now is the time to inform your Public Relations or Press Office about your paper, as they might be interested in promoting its publication. This will allow them time to prepare an accurate and satisfactory press release. Include your manuscript tracking number (NG-A57954R3) and the name of the journal, which they will need when they contact our Press Office.

Before your paper is published online, we will be distributing a press release to news organizations worldwide, which may very well include details of your work. We are happy for your institution or funding agency to prepare its own press release, but it must mention the embargo date and Nature Genetics. Our Press Office may contact you closer to the time of publication, but if you or your Press Office have any enquiries in the meantime, please contact press@nature.com.

Please note that Nature Genetics is a Transformative Journal (TJ). Authors may publish their research with us through the traditional subscription access route or make their paper immediately open access through payment of an article-processing charge (APC). Authors will not be required to make a final decision about access to their article until it has been accepted. [Find out more about Transformative Journals](https://www.springernature.com/gp/open-research/transformative-journals)

Authors may need to take specific actions to achieve [compliance](https://www.springernature.com/gp/open-research/funding/policy-compliance-faqs) with funder and institutional open access mandates. If your research is supported by a funder that requires immediate open access (e.g. according to [Plan S principles](https://www.springernature.com/gp/open-research/plan-s-compliance)), then you should select the gold OA route, and we will direct you to the compliant route where possible. For authors selecting the subscription publication route, the journal's standard licensing terms will need to be accepted, including [self-archiving and license to publish](https://www.nature.com/nature-portfolio/editorial-policies/self-archiving-and-license-to-publish). Those licensing terms will supersede any other terms that the author or any third party may assert apply to any version of the manuscript.

Please note that Nature Portfolio offers an immediate open access option only for papers that were first submitted after 1 January 2021.

If you have not already done so, we invite you to upload the step-by-step protocols used in this manuscript to the Protocols Exchange, part of our on-line web resource, natureprotocols.com. If you complete the upload by the time you receive your manuscript proofs, we can insert links in your article that lead directly to the protocol details. Your protocol will be made freely available upon publication of your paper. By participating in natureprotocols.com, you are enabling researchers to more readily reproduce or adapt the methodology you use. [Natureprotocols.com](https://natureprotocols.com) is fully searchable, providing your protocols and paper with increased utility and visibility. Please submit your protocol to <https://protocolexchange.researchsquare.com/>. After entering your [nature.com](https://www.nature.com) username and password you will need to enter your manuscript number (NG-A57954R3). Further information can be found at <https://www.nature.com/nature-portfolio/editorial-policies/reporting-standards#protocols>

Sincerely,
Kyle

Kyle Vogan, PhD
Senior Editor
Nature Genetics
<https://orcid.org/0000-0001-9565-9665>

Click here if you would like to recommend Nature Genetics to your librarian
<http://www.nature.com/subscriptions/recommend.html#forms>

** Visit the Springer Nature Editorial and Publishing website at http://editorial-jobs.springernature.com?utm_source=ejp_NGen_email&utm_medium=ejp_NGen_email&utm_campaign=ejp_NGen for more information about our career opportunities. If you have any questions please click [here](mailto:editorial.publishing.jobs@springernature.com). **